# Top-down perceptual inference shaping the activity of early visual cortex

Ferenc Csikor [1] ✉, Balázs Meszéna [1], Katalin Ócsai [1,2] & Gergő Orbán [1] ✉

Deep discriminative models provide remarkable insights into hierarchical processing in the brain by predicting neural activity along the visual pathway. However, these models differ from biological systems in their computational and architectural properties. Unlike biological systems, they require teaching signals for supervised learning. Moreover, they rely on feed-forward processing of stimuli, which contrasts with the extensive top-down connections in the ventral pathway. Here, we address both issues by developing a hierarchical deep generative model and show that it predicts an extensive set of experimental results in the primary and secondary visual cortices (V1 and V2). We show that the widely documented sensitivity of V2 neurons to textures is a consequence of learning a hierarchical representation of natural images. Further, we show that top-down influences are inherent to hierarchical inference. Hierarchical inference explains neural signatures of top-down interactions and reveals how higher-level representation shapes low-level representations through modulation of response mean and noise correlations in V1.

Hierarchical processing of visual information is a fundamental property of the visual cortex. Recently, deep learning models have emerged as central tools to investigate the hierarchical processing of visual information[1]. These models rely on a feed-forward processing hierarchy, which enables them to effectively process natural images. A highly successful class of models applied to the visual system are goal-oriented image models, which postulate that the relevant training objective of visual cortical computations is to perform specific tasks, such as classification of inputs into discrete categories. Such goal-directed models have been immensely successful in predicting neuronal responses to natural images, such that progression of processing stages qualitatively matched those of the ventral stream[2], as well as in designing synthetic images that elicit specific response patterns in populations of visual cortical neurons[3,4]. However, images that were specifically designed to confuse the feed-forward model also significantly delayed neural responses, indicating that computations beyond feed-forward processing were recruited by the visual system[5].

Top-down interactions between the processing stages are ubiquitous in the visual cortex (Fig. 1) but lack a well-established role in goal-directed models of vision. Scrutinizing the computational principles underlying the goal directed models might provide normative arguments why biological vision recruits a more complex architecture and how top-down connections help perceptual processes. Goal-oriented models learn to compress stimulus information such that information relevant for the specific task is retained. For instance, if a model is trained for animal classification, it will excel in this task by recognizing a tiger irrespective of its posture. To achieve this, goal-directed models rely on a learning paradigm called supervised learning, which capitalizes on image and category pairs to train the hierarchical model. Recently, studies have highlighted that the specific task the goal-oriented model is trained on is actually affecting how well a model accounts for neural data[6]. In contrast with supervised learning of a task-specific neural representation, biological learning imposes two seemingly contradicting requirements. First, representations need to be learned without supervision, i.e., learning in the visual cortex should be performed without a signal what its output should be. Second, the visual system is expected to learn a model of natural images that is not adapted to perform one specific task, instead one that can flexibly perform arbitrary tasks, according to the needs of the current context of the animal. In other words, while under some

[1]Department of Computational Sciences, HUN-REN Wigner Research Centre for Physics, Budapest 1121, Hungary. [2]Department of Algebra and Geometry, Institute of Mathematics, Budapest University of Technology and Economics, Budapest 1111, Hungary. ✉e-mail: csikor.ferenc@wigner.hun-ren.hu; orban.gergo@wigner.hun-ren.hu

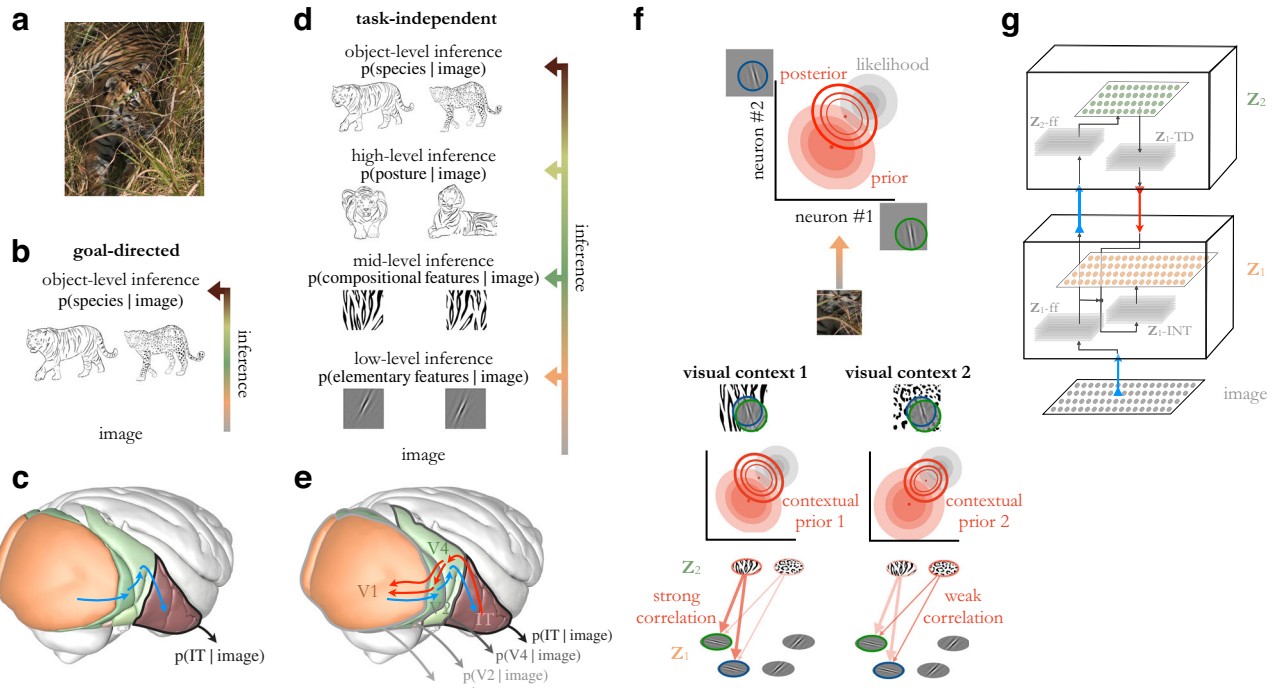

**Fig. 1 | Hierarchical inference in task-independent models. a** Example natural image. **b** Inference in a goal-directed model, which aims to identify categories of images. **c** Illustration of feed-forward processing in the ventral stream of the visual cortex. **d** Inference in a hierarchical task-independent model, which permits reverse engineering the contribution of a hierarchical set of features to the observed stimulus. **e** Illustration of top-down (*red arrows*) influences supplementing feed-forward processing (*blue arrows*) along the hierarchy of the ventral stream[10]. **f** Computational role of top-down connections. *Top*: Response intensities of a pair of $Z_1$ neurons (axes) are determined by the features they represent (insets on axes). Interpretation of the image (posterior distribution, *empty contours*) is the combination of the prior (*filled red ellipses*) and the evidence carried by the image (likelihood, *gray ellipses*). Mean response of the neuron (measured across trials, or across a long span of time) is the posterior mean (*dot*), its correlation is contributing to the noise correlation. *Middle*: Change of contextual priors upon changes in stimulus statistics. *Bottom*: Contribution of $Z_2$ neurons to contextual priors in $Z_1$. *Saturated colors* indicate stronger activation. **g** Neuronal circuitry of TDVAE that learns a hierarchy of features $Z_1$ (*orange*) and $Z_2$ (*green*) corresponding to V1 and V2 in the ventral stream. Features are represented in a layer of neurons (*planes with orange and green disks that correspond to individual neurons*). Layers of neurons (*gray planes*) transform feed-forward ($Z_1$-ff, $Z_2$-ff, *blue arrows*) and top-down information ($Z_1$-TD, *red arrow*) to ensure precise inference. Integration of feed-forward and top-down information is achieved at $Z_1$-INT. Image credit: (**a**, **f**) iStock Photos/Getty Images; (**b**, **d**) Users Appaloosa, Patrick Giraud, and VikiUNITED of Wikimedia Commons, user Nikhil Patle of pexels; (**c**, **e**) Imaging data provided by the Duke Center for In Vivo Microscopy NIH/NIBIB (P41 EB015897)[116,117].

circumstances it is sufficient to infer if we are facing a tiger or a zebra, in other circumstances it is imperative to precisely judge which movements are plausible based on the current posture of the animal. However, optimization for only one set of task variables (distinguishing different species) does not provide guarantees for efficient computations relying on other variables (evaluating the posture). These requirements motivate a task-independent framework for hierarchical processing in the visual cortex[7].

We propose that the ingredient that can address the dual challenge of learning is provided by top-down interactions. We argue that top-down signals have a central role in a class of task-independent learning systems, which has recently achieved considerable success in machine learning applications, deep generative models. Conceptually, top-down interactions establish expectations to support inference[8]: representing and computing with expectations is a key topic in theories of perception[9,10] and was shown to have both neuronal and behavioral correlates[8,11,12]. Top-down interactions ensure that instead of performing inference at the top of the hierarchy by a simple feed-forward pass of information, inference can be performed at intermediate layers too, which is supported by contextual information fed back from higher levels of hierarchy. Formally, top-down interactions establish a contextual prior which is integrated with evidence, carried by the feed-forward channel, to perform hierarchical inference.

In this study we investigate how a task-independent hierarchical model is acquired through adaptation to natural image statistics. Specifically, we investigate the hierarchical representations learned in

a deep generative model from two major directions. First, we seek to characterize learned representations to see if it reflects the properties of the early and mid-level visual cortices, V1 and V2 in particular. Second, we seek to understand if top-down effects found in V1 are aligned with the way high-level representations interact with low-level representations by establishing contextual priors. In order to explore the defining characteristics of this model, we study texture representations, which is motivated by three factors. First, textures are highly relevant features of natural images and critically contribute to higher-level visual processes such as segmentation and object recognition[13,14]. Second, texture representations have been established in mid-level visual cortices, and in particular in V2, across multiple species, ranging from humans[15], through non-human primates[16–18], to rodents[19], suggesting that textures are not only relevant from a machine learning point of view but from a neuroscience point of view as well. Third, while some features that the V2 is sensitive to can be identified with linear computations, textures can only be learned through non-linear computations, a critical component of the successes of deep learning models.

Based on fundamental principles of hierarchical inference, we extend the feed-forward deep learning framework to perform task-independent hierarchical inference (Fig. 1). The adopted Variational Autoencoder (VAE) framework allows unsupervised learning of a nonlinear model of natural images. We develop the hierarchical TDVAE (Top-Down VAE) model, which we train end-to-end on natural image patches. First, to establish the feasibility of the model we contrast the

representation learned by the hierarchical TDVAE with the responses of neurons in the early visual cortex of macaques. Next, we investigate the learned properties of top-down connections through a series of experiments that explore two aspects: contextual influences of V2 on V1 responses, and effects of V2 representations on V1 response properties. The proposed framework relies on a feed-forward V1 – V2 pathway and a top-down pathway that avoids direct feedback to the feed-forward components, but integrates feed-forward and top-down components in a distinct population. Through the development of a Variational Autoencoder model, our work provides a normative interpretation of the extensive top-down network present in the visual cortical hierarchy and demonstrates signatures of top-down interactions in neural response patterns.

## Results

To investigate the contribution of top-down connections to computations in a normative framework, we introduce a task-independent hierarchical model of natural images. In a hierarchical task-independent model inference aims to establish the presence or absence of a hierarchy of features of different complexities (Fig. 1d). This is phrased as learning a joint distribution over objects and features of different complexities: $p(\text{features}_L, \text{features}_M, \text{features}_H, \text{object} \mid \text{image})$. In contrast, a goal-oriented model is trained to make inference only about high-level features, object category in our case (Fig. 1b). Here, $p(\text{object} \mid \text{image})$ is calculated such that object category information is provided during the training to learn a flexible and non-linear mapping from images to object categories[1,2,20]. In contrast with learning to infer the topmost features of the hierarchy, task-general learning is a more fine-grained form of inference that enables performing not only one particular task, such as viewpoint invariant recognition of objects, but also learning to infer the different aspects of different observations of the same visual component. Given that a task-independent learning is taking place, no category labels are provided at any level of the computational hierarchy. In both model categories, activity of neurons in the hierarchy of the ventral stream are identified with the activities of model neurons at different processing stages but while in goal-directed models intermediate states merely serve the inference of the features at the top level (Fig. 1c), in task-general models neurons at different stages of the ventral stream represent different stages in hierarchical inference (Fig. 1e). In the coming sections, we first introduce the computational principles of hierarchical inference in a task-independent model of natural images; second, we introduce algorithmic details of inference, which provides insights about the circuitry underlying inference; third, the neural circuitry is introduced.

### Computational principles of hierarchical inference

We focus on inference on two levels of features: $p(\text{features}_L, \text{features}_M \mid \text{image})$, which we will refer to as $\mathbf{Z}_1$ and $\mathbf{Z}_2$ for clarity. Thus, the values in vector $\mathbf{Z}_1$ denote the activity of a population of low-level model neurons, while values in $\mathbf{Z}_2$ denote the activities of model neurons at the higher layer of the computational hierarchy. Image-evoked activity in $\mathbf{Z}_1$ is determined by solving an inference problem:

$$p(\mathbf{Z}_1 \mid \text{image}) = \int d\mathbf{Z}_2 \; p(\mathbf{Z}_1, \mathbf{Z}_2 \mid \text{image}). \tag{1}$$

Here, $p(\mathbf{Z}_1 \mid \text{image})$, the posterior distribution, assigns probabilities to different response intensities of the neurons of the $\mathbf{Z}_1$ population for a stimulus such that the most likely response intensity corresponds to the most probable value of the feature in the image. Partitioning the joint distribution $p(\mathbf{Z}_1, \mathbf{Z}_2 \mid \text{image})$ provides insights into the dependencies. The feed-forward partitioning solely relies on feed-forward progression of information (see also Methods): $p(\mathbf{Z}_1 \mid \text{image}) \cdot p(\mathbf{Z}_2 \mid \mathbf{Z}_1)$. In contrast, the top-down partitioning also relies on top-down

interactions between processing stages: $p(\mathbf{Z}_1 \mid \mathbf{Z}_2, \text{image}) \cdot p(\mathbf{Z}_2 \mid \text{image})$. These formulations provide mathematically equivalent formulations to the inference problem and as such either of these could be used to learn a hierarchical task-independent model. However, inference is only approximated and one of the most effective approximations, variational inference, terminates equivalence[21]. Motivated by the top-down interactions characteristic of the anatomy of the visual cortex[22], we focus on hierarchical inference that recruits top-down interactions besides feed-forward processing (Fig. 1e). In our study we will seek to establish links between model neurons in $\mathbf{Z}_1$ and $\mathbf{Z}_2$ with neurons in V1 and V2 regions of the visual cortex. Specifically, we identify $\mathbf{Z}_1$ responses with layer 2/3 responses in V1 (see later in this section). Although the circuitry for inference is more complex than a single layer, for the sake of simplicity, we refer to the $\mathbf{Z}_1$ neurons as V1 throughout the paper. In order to assess potential differences of the two partitionings of the joint distribution, we also investigate the feed-forward parametrization.

A key insight of the top-down component in hierarchical inference becomes evident when the interpretation of an input image becomes uncertain as a consequence of poor viewing conditions, occlusion, or noise (Fig. 1a). Under uncertainty, an observer needs to rely on their prior[23,24] (Fig. 1f, top), which carries information about learned regularities in the environment. Importantly, regularities change from environment to environment, implying that an observer knowledgeable about the environment shall rely on local environment-dependent priors, termed contextual priors (Fig. 1f, middle). According to hierarchical inference, those are the $\mathbf{Z}_2$ neurons that can provide contextual priors for $\mathbf{Z}_1$ neurons through top-down connections, as patterns in top-down connections reflect the statistics of the local context. This provides the flexibility to alter the contextual information of $\mathbf{Z}_1$ neurons through different activation patterns in $\mathbf{Z}_2$ (Fig. 1f, bottom).

### Algorithmic solution to hierarchical inference

In order to train a network to perform hierarchical inference in a task-independent model of images, we turn to the Variational Autoencoder (VAE) formalism[25,26]. VAEs learn a pair of models: the recognition model supports inference, i.e. it describes the probability that a particular feature combination underlies an observed image; and the generative model, which summarizes the 'mechanics' of the environment, i.e. how the combination of features produces observations (see Methods for details, Supplementary Fig. 1). Relying on advances in hierarchical versions of VAEs[27,28], we developed Top-Down VAE, or TDVAE for short (Fig. 1g). We constructed the TDVAE model based on two principles. First, the computational framework is determined by a highly principled approximation in probabilistic inference but apart from the well-motivated restrictions (optimizing a lower bound to the true objective, performing amortized variational inference, deterministic propagation of signals across layers of the hierarchy), the TDVAE was left as general as possible such that its properties were determined solely by natural image statistics. Second, we capitalized on neuroscience intuitions about V1 and V2 to constrain the architecture for learning about low- and mid-level features, $\mathbf{Z}_1$ and $\mathbf{Z}_2$. Along this line, we assumed sparseness in the generative model[29] (Methods). Further, we assumed a linear relationship between the features that $\mathbf{Z}_1$ neurons are sensitive to and the pixels of natural images[29,30]. At the level of $\mathbf{Z}_2$, the generative model allowed flexible nonlinear computations such that more complex feature sets can be learned. Next, as the recognition model ultimately defines the neural circuitry for hierarchical inference, we relied on anatomical insights (Fig. 1g). The top-down formulation requires computing two components: $p(\mathbf{Z}_1 \mid \mathbf{Z}_2, \text{image})$, and $p(\mathbf{Z}_2 \mid \text{image})$. In a VAE formalism, these components are calculated by neural networks. We used the same feed-forward pathway to process image information (Fig. 1g), implemented by the neural network that is denoted by $\mathbf{Z}_1$-ff, which pathway forked into a pathway reaching

$Z_1$ and another reaching $Z_2$, the latter implemented by the neural network denoted by $Z_2$-ff. This joint feed-forward component ensures that no direct connection to $Z_2$ is assumed. The top-down information (Fig. 1g), implemented by the neural network that is denoted by $Z_2$-TD, is integrated with the feed-forward pathway in a neural network denoted by $Z_1$-INT before reaching $Z_1$ (Fig. 1g, see also Methods), reminiscent of the integration of the likelihood and prior for Bayesian inference. The joint feed-forward pathway is analogous to parameter sharing, a widely used approach in machine learning[28,31]. In order to demonstrate that the shared feed-forward pathway does not limit the performance of the TDVAE model, we have also implemented the alternative model with distinct feed-forward pathways (see later).

The full computational graph of the TDVAE describes the way the input is processed through subsequent layers to infer $Z_1$ and $Z_2$, as well as the way the generative component contributes to the reconstruction of the input, the precision of which ultimately drives learning (Supplementary Fig. 1). In this paper we focus on the analysis of the recognition model, that is, activities of model neurons in $Z_1$ and $Z_2$ are used to predict patterns in cortical neural activities (Fig. 1g).

$Z_1$ and $Z_2$ activations in TDVAE are assumed to correspond to activations of V1 and V2 neurons up to a linear combination of the variables (see Methods for details), standard in earlier approaches[2]. The probabilistic nature of inference (Eq. (1)) necessitates that we define how probability distributions are represented. In a non-probabilistic approach the neural response would be identified with the maximum of the distribution. In neural data, this maximum a posteriori response can be identified with the trial-averaged response and variance is interpreted as mere noise in the circuitry. For a probabilistic interpretation of neural responses, we adopt the sampling hypothesis[32]. The sampling hypothesis takes a stochastic approximation to represent a probability distribution. According to this, neural activity at any given time is a stochastic sample taken from the probability distribution and the sequence of samples constitutes the distribution, with mean of the neural activity corresponding to the most probable interpretation of the stimulus (see also Methods). Thus, mean of $p(Z_1 | image)$ and $p(Z_2 | image)$, referred to as posteriors, corresponds to the mean response of the V1 and V2 neurons. Spike count collected in a time window corresponds to a finite number of samples from the posterior therefore it is expected to show considerable variance around the mean posterior, which can be reduced through averaging across trials. At the level of V1, a single sample can be identified with an activity from a 20 ms time window[33]. Width of the posteriors is reflected in response variance, while correlations in the posterior are reflected in noise correlations[34,35]. Importantly, noise correlations can be produced by various sources[36], and thus our interpretation corresponds to a form of noise correlation that results from high-level perceptual variables. In this paper response variances are not investigated in detail but we refer the reader to earlier work[37,38]. In practice, we collect samples from the variational posteriors of $Z_1$ and $Z_2$, which is then compared to responses of V1 and V2 neurons in electrophysiology experiments. The number of these samples is set to approximate the specific experimental conditions TDVAE responses are tested on (see Methods). Activity of model neurons are not constrained to be positive and therefore are identified with membrane potentials, similar to earlier approaches[37] (see Methods for details). The presented results can be translated to firing rates by transforming membrane potentials with a threshold-linear transformation[37,39]. We motivate reporting results on the untransformed activities of $Z_1$ and $Z_2$ in order to demonstrate that the results of an end-to-end natural image trained VAE can be directly applied to predict neural data.

## Neural circuitry for hierarchical inference

We investigate activity of $Z_1$ and $Z_2$ of the recognition model. As these deliver the result of inference, read by downstream structures, we relate their activities to layer 2/3 of V1 and V2. This assumption is well aligned with the fact that layer 2/3 integrates feed-forward and top-down information[40]. While we do not expect that the neural circuitry is fully determined by the computational architecture proposed here, it is a useful exercise to consider potential mapping between the two. The joint feed-forward component of the recognition model (Fig. 1g, $Z_1$-ff) can be considered as layer 4 neurons in V1, receiving relatively unfiltered input from LGN[40], and sending projections within V1 as well as feed-forward input to V2[41]. The top-down information initially reaches V1 at $Z_1$-INT (Fig. 1g) before converging on $Z_1$. $Z_1$-INT is in a similar position as layer 5 neurons of V1, known to receive top-down input[42,43]. Note that the computational graph does not rely on direct forward projections from $Z_1$ towards $Z_2$, thus avoiding dense recurrent information flow. Extensive layer 2/3 forward connections that are characteristic of V1[40,41] are not exploited by the recognition model and thus can serve further elaboration of the posterior.

To investigate the representations emerging as a result of adaptation to natural image statistics[23,24], we trained the TDVAE model end-to-end on natural image patches (Methods). To demonstrate the invariance of our results on the specific choice of the generative and recognition models, we trained an array of models with different parametrizations, including the feed-forward partitioning of the posterior (Eq. (1)) and an untrained TDVAE model (see Methods). The goodness of these alternative models was established through calculating the value of the optimization objective (ELBO, Fig. 2a, Supplementary Table 3, Methods) and the model having the highest ELBO was used in the main text of the study, while alternatives are presented as Supplementary Information. Inference in the model was validated by contrasting the match between inference in the recognition and in the generative models (see Methods for details and Supplementary Fig. 1). We tested a model alternative, in which parameter sharing was not present in the two components of the recognition model. We performed the comparison on a smaller model that used 20 × 20-pixel image patches instead of the default 40 × 40-pixel patches. Simulations did not identify major performance or qualitative differences (− 400, distinct pathways: − 407; see also Supplementary Table 3; see also Methods). Apart from these architectural choices, TDVAE has no free parameters that are fitted to neural data: all parameters are determined by natural image statistics.

## Hierarchical representation of natural images

First, we investigate the representations that emerge in TDVAE. At the level of $Z_1$, we found that TDVAE robustly learns a complete dictionary of Gabor-like filters (Fig. 2b, Supplementary Fig. 3, Supplementary Table 3). The receptive fields of individual model neurons are localized, oriented, and bandpass (Supplementary Fig. 4), qualitatively matching those obtained by training a VAE lacking a $Z_2$ layer (Supplementary Fig. 5). This result confirms earlier studies using single layer linear generative models featuring sparsity constraint on neural activations[29,30]. Thus, learning a hierarchical representation complete with a $Z_2$ layer on top of $Z_1$ left the qualitative features of the $Z_1$ representation intact in the TDVAE model.

Unlike the compact receptive fields of $Z_1$ model neurons, the representation at the $Z_2$ layer has no noticeable linear structure (Fig. 2b STA). The second-order, nonlinear receptive fields and the orientation tuning curves of $Z_2$ units, however, reveal orientation and wavelength selectivity (Fig. 2b STC and ori., Methods), offering a first glimpse into the representation learned by the $Z_2$ layer.

Inspired by extensive evidence supporting that V2 neurons are sensitive to texture-like structures[15,19,44], we used synthetic texture patches to further explore the properties of the $Z_2$ representation that TDVAE learned on natural images[45]. We selected 15 natural textures characterized by dominant wavelengths compatible with the image patch size used in the study (Fig. 2c, Methods). We matched low-level texture family statistics (Methods), thus making texture families linearly indistinguishable in the pixel space (Fig. 2d, Methods). Since

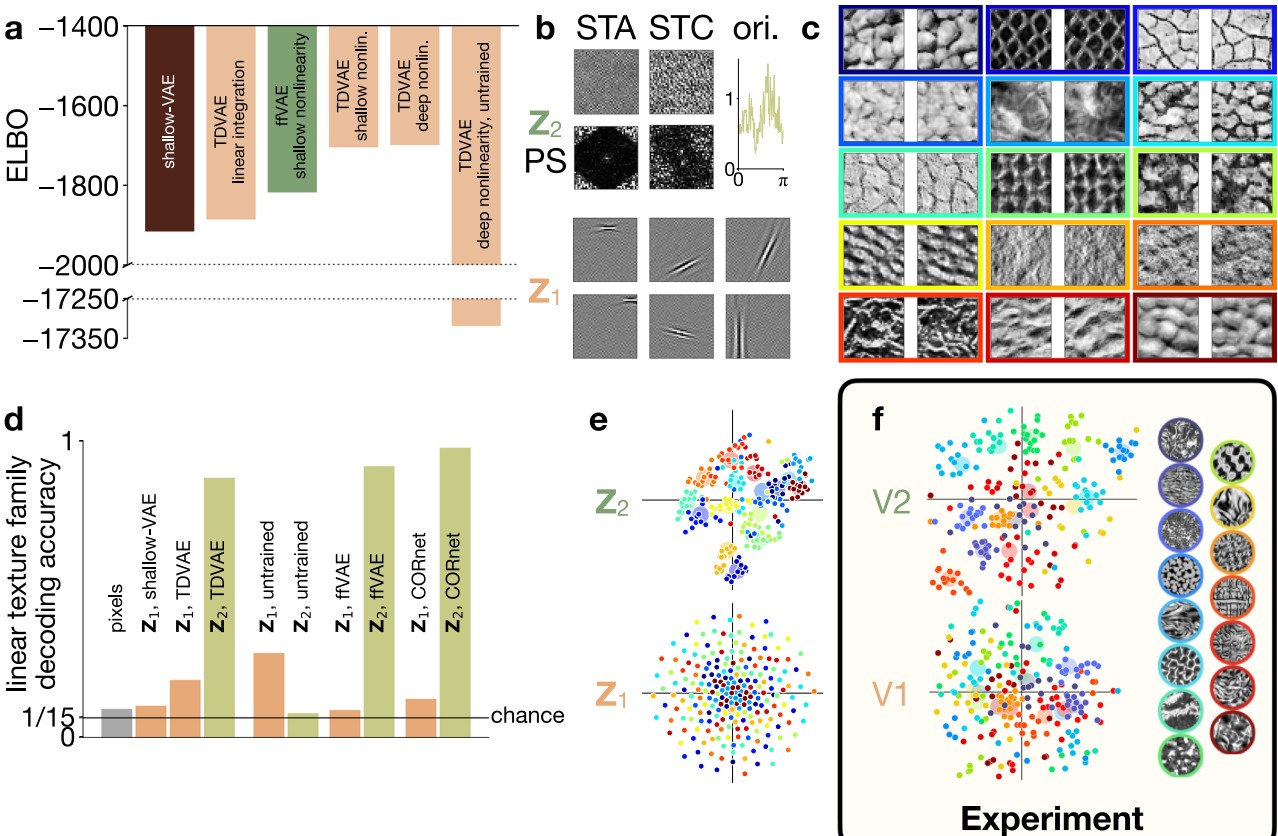

**Fig. 2 | Hierarchical representation in $Z_1$ and $Z_2$ learned by TDVAE. a** Loss function of alternative models. *Shallow-VAE*: non-hierarchical VAE constrained to $Z_1$, yielding a linear model of V1; *ffVAE*: hierarchical VAE with feed-forward processing; *TDVAE*: variants of hierarchical VAEs with top-down processing. *Linear integration*: feed-forward and top-down pathways are linearly combined to reach $Z_1$; *shallow nonlinearity*: generative model between $Z_2$ and $Z_1$ relies on single-layer MLP (Supplementary Table 2); *deep nonlinearity*: as above but with two-layer MLP. *Untrained*: Model with random parameters. **b** *Top*: The first-order and an example second-order receptive field, as well as the orientation tuning (*STA, STC*, and *ori.*, respectively, see Methods) of a selected active $Z_2$ neuron, together with power spectra (*PS*). *Bottom*: Example first-order receptive fields of six $Z_1$ units. **c** Examples from the 15 texture families (*colors*) used in the paper. **d** Texture family decoding accuracies from different layers of the hierarchy. Means of $n = 5$ fits are shown; s.d. < 0.01 everywhere. *Black line*: chance performance. *Pixels*: direct decoding from images. Labels as in (**a**). *CORnet*: goal-directed feed-forward model. **e** Two-dimensional visualization (t-SNE) of mean responses of $Z_1$ and $Z_2$ neurons to randomly sampled texture images (*dots*). *Colors* as on panel (**c**). *Disks*: means across samples from a family. **f** Same as (**e**) but in V1 and V2 recordings from macaques. Reproduced with permission from[17](PNAS). Source data are provided as a Source Data file.

model $Z_1$ neurons learn a linear mapping from the neural space to the pixel space, we expect that texture information cannot be read out with a linear decoder from $Z_1$ response intensities either. Decoders that queried the texture family that a particular image was sampled from but ignored the identity of the image showed low performance on model $Z_1$ neurons, which was still distinct from chance ($0.1943 \pm 0.0012$, error denotes standard deviation over five repeats of the fit, Fig. 2d '$Z_1$, TDVAE'). The non-hierarchical version of VAE that completely lacked a $Z_2$ was slightly outperformed by TDVAE (compare $0.1073 \pm 0.0020$ with $0.1943 \pm 0.0012$ for '$Z_1$, shallow-VAE' and '$Z_1$, TDVAE', respectively, Fig. 2d).

Nonlinear components in the generative model endow the $Z_2$ with building a representation that is capable of distinguishing texture families such that textures can be effectively represented in this layer. Texture was reliably decodable from the response intensities of $Z_2$ neurons of TDVAE ($0.8761 \pm 0.0001$, '$Z_2$, TDVAE', Fig. 2d). This texture representation was very robust as it was qualitatively similar for all investigated hierarchical architectures (Supplementary Table 4). The texture representation discovered by TDVAE was low dimensional, irrespective of the architectural details of the model (Supplementary Table 4). This finding is surprising but recent experiments have confirmed that even though textures are represented by large neuron populations in the V2 of macaques, the effective dimensionality of this

representation is very limited[19]. Intriguingly, the TDVAE architecture best fitting natural images only featured texture-selective dimensions, yielding a very compact representation exclusively focusing on textures (Supplementary Tables 3, 4). Responses of the $Z_2$ population displayed clustering by the texture family (Fig. 2e), similar to the representations identified in V2 neuron populations in macaque (Fig. 2f). In contrast, no such clustering could be identified in $Z_1$ (Fig. 2e), reminiscent of V1 neuron populations of the same macaque recordings (Fig. 2f).

We introduced three alternative models as controls to our TDVAE model. First, we investigated a model that had identical architecture to the TDVAE but was not adapted to natural images. Second, we tested the robustness of the learned texture representation against the form of partitioning of the posterior distribution. Third, representation learned by a goal directed feed-forward model was tested. The untrained model showed slightly higher decoding performance on texture families in $Z_1$ (compare $0.2850 \pm 0.0078$ with $0.1943 \pm 0.0012$ for '$Z_1$, untrained' and '$Z_1$, TDVAE', respectively, Fig. 2d) but, in contrast with TDVAE, markedly lower performance in $Z_2$ than the TDVAE (compare $0.0820 \pm 0.0076$ with $0.8761 \pm 0.0001$ for '$Z_2$, untrained' and '$Z_2$, TDVAE', respectively, Fig. 2d). The contra-intuitive higher $Z_1$ result is explained by the fact that decoding is performed from a large dimensional space (matching the dimensionality of active $Z_1$

dimensions) and the non-linear recognition model has the potential to distribute responses of $Z_1$ such that it carries texture family information. As it is the generative model that assumes linear relationship between $Z_1$ and pixel activations, the nonlinear recognition model 'unlearns' texture decodability as the generative and recognition models are jointly adapted to natural images. In contrast with TDVAE, no clustering was found in either $Z_1$ or $Z_2$ of the untrained model (Supplementary Fig. 6a).

The variant of TDVAE using feed-forward partitioning of the joint posterior distribution instead of top-down partitioning, ffVAE (*feed-forward VAE*), showed limited texture family decodability from $Z_1$ (0.0930 ± 0.0003, '$Z_1$, ffVAE') and high texture family decodability from $Z_2$ (0.9150 ± 0.0002, '$Z_2$, ffVAE', Fig. 2d), similar to TDVAE. This underlines the robustness of the learned texture representation against the exact implementation of the recognition model.

We have contrasted the hierarchical representation learned by TDVAE with that of a goal-directed model. We used a standard feed-forward implementation available in the literature, the CORnet model[46]. The CORnet model was trained on ImageNet and we designed analyses analogous to those performed with TDVAE (see Methods). In particular, we sought to identify layers across the processing hierarchy that bear similarities with $Z_1$ and $Z_2$ of TDVAE. Early layers with localized and orientation sensitive filters could be identified in this goal-directed model. Texture family decoding was not possible from the responses of this layer (0.1303 ± 0.0005, '$Z_1$, CORnet', Fig. 2d), but layers immediately above this layer were characterized by a representation from which texture family decoding was effective (0.9770 ± 0.0003, '$Z_2$, CORnet', Fig. 2d).

We investigated the representation learned by TDVAE further in order to contrast it with more specific properties of the representation found in V1 and V2. We first investigated the learned invariances of TDVAE. Mean responses of individual $Z_1$ neurons displayed a high level of variability both across instances of texture images belonging to the same texture family and those belonging to different texture families (Fig. 3a). This tendency was similar to the sensitivities displayed by V1 neurons in the macaque visual cortex (Fig. 3b). In contrast, $Z_2$ neurons displayed a higher level of invariance across texture images belonging to the same texture family (Fig. 3a), again reflecting the properties of macaque V2 neurons (Fig. 3b). To quantify this observation, across-population statistics was calculated by dissecting across-sample and across-family variances in mean responses both in $Z_1$ and $Z_2$ neurons of TDVAE using a nested ANOVA method[17]. Note, that in order to make our analysis more general by permitting an arbitrary linear mapping between model $Z_2$ neurons and biological neurons, we used a random rotation of $Z_2$ to perform the analysis (Methods). The relative magnitude of response variance across families versus across samples was significantly higher in $Z_2$ than in $Z_1$ (Fig. 3c), corroborating experimental findings (Fig. 3d).

Higher invariance of responses to different stimuli belonging to the same texture family in $Z_2$ than in $Z_1$ can also be captured by a linear decoding analysis (Fig. 3e). Efficiency of discrimination between images in the same texture family is weaker in $Z_2$ than that of $Z_1$ (Fig. 3e). Such increasing invariance to lower-level features along the visual hierarchy is a hallmark of gradual compression and can be identified at the population level in the primate visual cortex as well (Fig. 3f). In contrast with within-family stimulus identity decoding, decoding of texture family is more efficient from $Z_2$ than from $Z_1$ in TDVAE (Fig. 3e), a finding confirmed by macaque recordings (Fig. 3f).

Texture families can be defined through a set of pairwise statistics over the co-activations of linear filters represented in $Z_1$/V1[45]. To demonstrate that the learned $Z_2$ representation reflects these quintessential features of a proper texture representation, the TDVAE model can be tested with stimuli in which these high-level statistics are selectively manipulated while keeping the low-level

statistics intact (Methods). To achieve this, we used phase scrambling of images[16]. As texture sensitive neurons are assumed to be sensitive to high-level statistics, their removal is expected to reduce $Z_2$ activations. For this, a modulation index was calculated for each neuron, which was defined as the difference between sample- and presentation-averaged absolute responses to undisturbed texture images and their phase scrambled counterparts normalized by their sum. Texture-sensitive units in $Z_2$ showed higher modulation than active units in $Z_1$ (Fig. 3g), in line with findings in macaque recordings (Fig. 3h).

In summary, $Z_1$ and $Z_2$ representations of TDVAE display key features of V1 and V2 representations of the macaque visual cortex. In particular, an elaborate texture representation is a salient feature of hierarchical representations of natural images.

## Top-down contributions to hierarchical inference

In a task-independent model inference can be performed on all levels of the hierarchy (Fig. 1d). According to equation (1), top-down influences affect the inference of $Z_1$ activities through establishing a contextual prior for $Z_1$ (Fig. 1f). Importantly, contrary to simple Bayesian computations, this prior is not an invariant component, instead it depends on the higher-level interpretation of the scene. In the following sections we seek to identify signatures of this contextual prior in $Z_1$ response statistics: first on mean responses, second on response correlations.

Top-down influences can be expected when local interpretation of images depends on the surroundings. Establishing a wider context is directly supported by the growing receptive field sizes in higher hierarchical layers of TDVAE. To study top-down influences in detail, we rely on an implementation of TDVAE, which is trained on larger (50-pixel, see Methods) natural image patches than our standard model, which was trained on 40-pixel images. While the larger patch size requires more careful training, it offers a more precise evaluation of contextual effects. A particularly strong form of contextual effect can be established by selectively blocking direct stimulus effects from designated $Z_1$ neurons and therefore top-down effects can be studied in isolation by investigating the emerging $Z_1$ activation. This 'illusory' activity in $Z_1$ neurons resonates with the interpretation of illusions as the contribution of priors to perception under uncertainty[47]. We constructed illusory contour stimuli by creating Kanizsa square stimuli (Fig. 4a) such that illusory contour segments were aligned to individual $Z_1$ neurons and measured mean responses (Fig. 4b). We compared Kanizsa-elicited responses to both unobstructed images of squares and to control stimuli that were built from identical elements to the Kanizsa stimuli but in configurations not congruent with the percept of a square. Illusory edge responses were similar to unobstructed square stimulus-evoked responses, albeit with a lower gain for the illusory edge (Fig. 4b), similar to responses of V1 neurons of macaques[48] (Fig. 4c). Confirming expectations, responses to Kanizsa-incongruent stimulus elements were muted (Fig. 4b), again reproducing experimental findings (Fig. 4c). This tendency was consistent across the model neurons investigated using stimuli that were tailored to their response properties (see Methods, Fig. 4d).

Using the TDVAE model, we designed a stricter control. Overlap between the receptive field of $Z_1$ neurons with the elements of the Kanizsa stimulus could contribute to model responses. To isolate top-down effects, we calculated linear responses of the neurons to the Kanizsa stimuli (Fig. 4e). We found significant boosting of the linear response in TDVAE for the example neuron. By restricting the set of analyzed neurons to those having (1) only a small overlap between the 'Illusory' stimulus and the receptive field and (2) all four response peaks in Fig. 4e close to each other (a proxy of regular peak shapes), consistent boosting was observed ($n = 9$; significant boosting: $n = 9$, one-sided one-sample $t$-tests, $P < 0.05$; population statistics: Fig. 4g, Methods). Incongruent Kanizsa stimuli produced

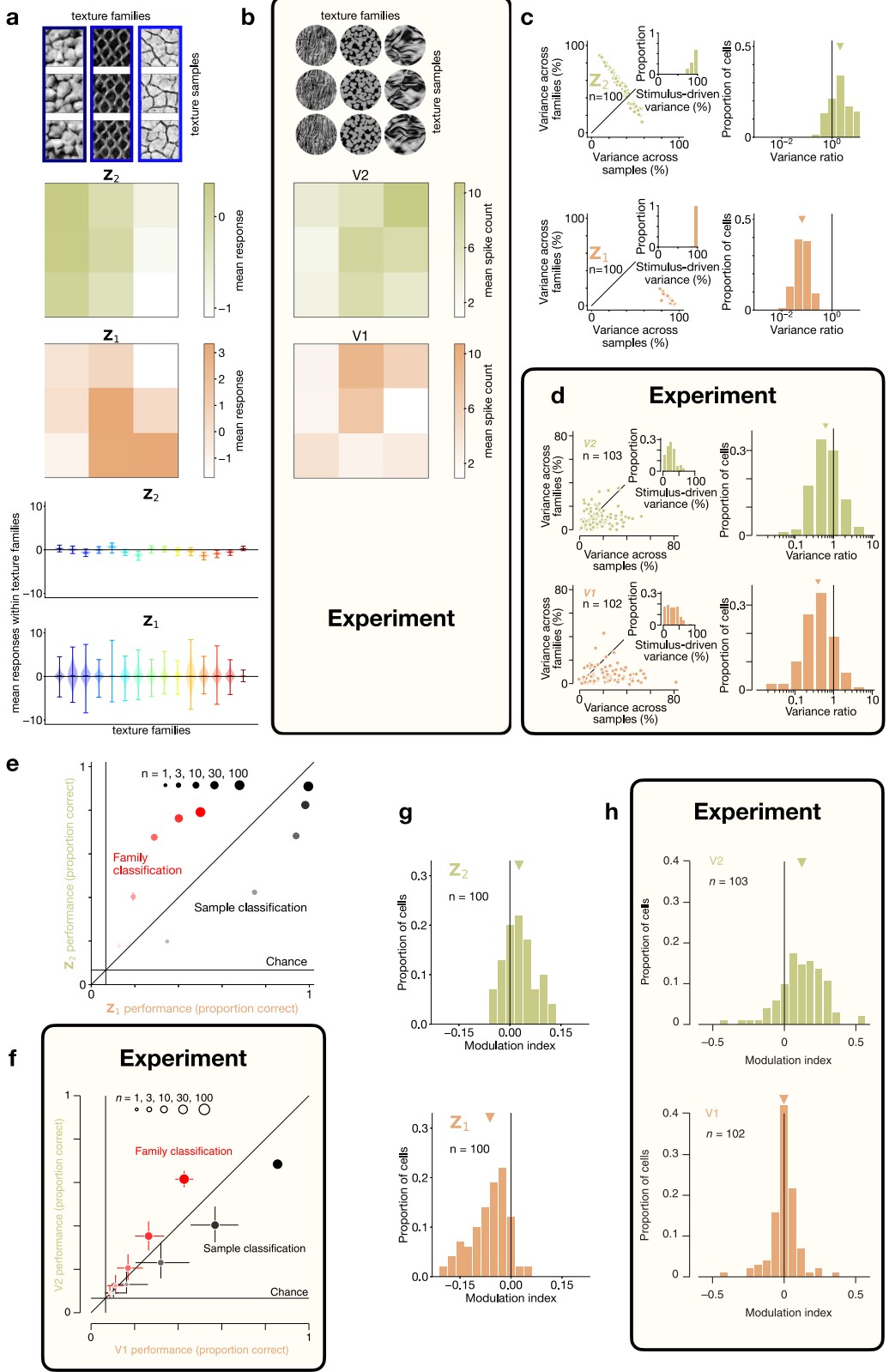

limited and inconsistent modulation in the same set of $Z_1$ neurons ($n = 9$; significant boosting: $n = 5$, significant suppression: $n = 3$, one-sided one-sample $t$-tests, $P < 0.05$; population statistics: Supplementary Fig. 6b). Confirming the contributions of top-down connections to illusory activity, modulation of $Z_1$ responses was significantly weaker in a model that lacked $Z_2$ than in the TDVAE model (Fig. 4g).

To explicitly investigate the contribution of $Z_2$ to illusory responses in $Z_1$, we manipulated $Z_2$ in two different ways. First, to simulate severing feed-forward connections to $Z_2$, we removed stimulus information by sampling the prior of $Z_2$ instead of the posterior when presenting a Kanizsa stimulus. As predicted, severing feed-forward connections results in TDVAE responses matching linear responses (Fig. 4g). Second, as a proxy to severing feedback

**Fig. 3 | Signatures of learning a texture representation. a** Mean responses of one example neuron from $Z_2$ and one example neuron from $Z_1$ (rows two and three, respectively) to three example images from three texture families each (*top*). *Bottom*: Variability of mean responses of the same two neurons across 4000 images in each of the 15 texture families (*colors*; center lines: means, whiskers: extrema). **b** Same as (**a**) *top* in V1 and V2 recordings from macaque monkeys. **c** *Left*: Partitioning single-unit response variances into variance across images from different families, variance across images within families and a residual component (across stimulus repetitions) with nested ANOVA both in $Z_2$ and $Z_1$ ($n = 100$ neurons, *dots*). *Insets*: Distribution of the sum of the first two components of variance. *Right*: Distribution of the ratio of the first two components. Geometric mean of variance ratios (*triangles*) is significantly smaller in $Z_1$ (0.064) than in $Z_2$ (2.097) (one-sided independent two-sample t-test in the log domain, $t(\mathrm{df} = 198) = -32.7$, $p = 6.5 \times 10^{-82}$, $d'_e = 4.7$, 95% confidence interval = $[-\infty, -3.3]$). Here and later in the paper, $d'_e$ denotes the average sd discriminability index. **d** Same as (**c**) in 102 V1 and 103 V2

units from a pool of 13 macaque monkeys. **e** Decoding texture family (*red*) and within-family stimulus identity (*black*) from the same 100 $Z_1$ and $Z_2$ units (*horizontal axis* and *vertical axis*, respectively) as in (**c**). *Dot sizes*: decoding on randomly selected subpopulations ($n = 1, 3, 10, 30, 100$). *Solid line*: Chance performance. Dot centers: means; error bars: 95% confidence intervals of bootstrapping over included neurons and partitionings. **f** Same as (**e**) from single unit V1 and V2 macaque recordings. **g** Phase scrambling-induced modulation of the magnitude of mean responses to texture images in the same 100 $Z_2$ (*top*) and $Z_1$ (*bottom*) neurons as in (**c**). Mean modulation index (*triangles*) is significantly smaller in $Z_1$ (−0.062) than in $Z_2$ (0.027) (one-sided independent two-sample t-test, $t(\mathrm{df} = 198) = -12.4$, $p = 8.6 \times 10^{-27}$, $d'_e = 1.77$, 95% confidence interval = $[-\infty, -0.077]$). **h** Same as (**g**) from V1 and V2 recordings in macaques. Experimental data panels are reproduced with permission from PNAS (**b**–**f**[17]) and SNCSC (**h**[16]). Source data are provided as a Source Data file.

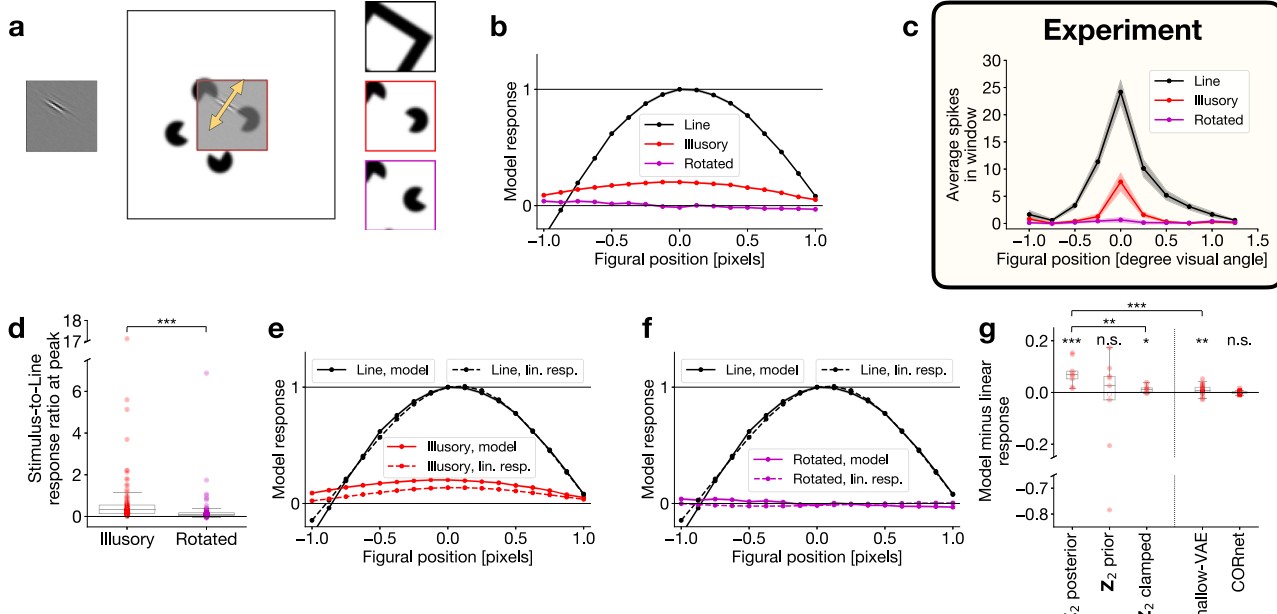

**Fig. 4 | Contribution of top-down computations to illusory contour responses. a** Illustration of the illusory contour experiment. *Left*: receptive field of an example model $Z_1$ neuron. *Middle*: receptive field-aligned Kanizsa square stimulus. Red border indicates the actual boundary of the stimulus. Arrow: direction of shifts of the stimulus relative to the neural receptive field. *Right*: real square (`Line'), Kanizsa square (`Illusory') and incongruent (`Rotated') stimuli. **b** Mean responses of the $Z_1$ neuron in (**a**) to 500 presentations of the three stimuli as a function of stimulus shift. Shaded regions (invisibly small): s.e.m. **c** Same as (**b**) for a selected unit in the V1 of a macaque monkey. Number of trials is unknown. Reproduced from[48] with permission (PNAS, Copyright (2001) National Academy of Sciences, U.S.A.). **d** Ratio of mean $Z_1$ responses to `Illusory' and `Line' stimuli and to `Rotated' and `Line' stimuli, respectively, at the `Line' response peak, for the largest stimulus sizes that fit into the patch, for the analyzed $Z_1$ population ($n = 114$ = (57 central, localized, medium wavelength $Z_1$ filters) × (upright and upside-down stimulus orientations), Methods; center line, median; box limits, upper and lower quartiles; whiskers, 1.5 × interquartile range). Mean peak ratios: 0.70 (`Illusory'), 0.23 (`Rotated'), the latter being significantly less than the former (one-sided paired two-sample t-test, $t(\mathrm{df} = 113) = 4.3$, $p = 1.6 \times 10^{-5}$, $d'_e = 0.39$, 95% confidence interval = $[0.29, \infty]$). *$p < 0.05$, **$p < 0.01$, ***$p < 0.001$; n.s., $p \geq 0.05$ in this and all subsequent Figures. **e** Same as (**b**) for the `Line' and `Illusory' stimuli, together with linear responses of the $Z_1$ neuron (dashed lines). Magnitude of the linear response to the `Line' stimulus was scaled to the mean response peak to the `Line' stimulus. Shaded regions (invisibly small): s.e.m. **f** Same as (**e**) for the `Line' and `Rotated' stimuli.

**g** Differences between the mean and linear responses to the `Illusory' stimulus (center line, median; box limits, upper and lower quartiles; whiskers, 1.5 × interquartile range). *Left*: Same restricted $Z_1$ population of TDVAE in three different conditions ($n = 9$ each; see text, Methods): intact inference in $Z_1$ (mean model-linear response difference (0.076) is positive: one-sided one-sample t-test against 0: $t(\mathrm{df} = 8) = 4.7$, $p = 0.00076$, 95% confidence interval = $[0.046, \infty]$), inference without stimulus-specific information at $Z_2$ (mean model-linear response difference (−0.068) is not significant: two-sided one-sample t-test against 0: $t(\mathrm{df} = 8) = -0.71$, $p = 0.50$, 95% confidence interval = $[-0.29, 0.15]$), inference with $Z_2$ clamped to zero (mean model-linear response difference (0.014) is significant: one-sided one-sample t-test against 0: $t(\mathrm{df} = 8) = 2.9$, $p = 0.011$, 95% confidence interval = $[0.0049, \infty]$, but smaller than TDVAE with intact inference: one-sided paired two-sample t-test: $t(\mathrm{df} = 8) = -4.2$, $p = 0.0015$, $d'_e = 2.07$, 95% confidence interval = $[-\infty, -0.034]$). *Right*: Illusory responses in two control models: shallow-VAE ($n = 29$; mean model-linear response difference (0.011) is positive: one-sided one-sample t-test against 0: $t(\mathrm{df} = 28) = 3.2$, $p = 0.0016$, 95% confidence interval = $[0.0050, \infty]$, but smaller than TDVAE with intact inference: one-sided independent two-sample t-test: $t(\mathrm{df} = 36) = -6.1$, $p = 2.2 \times 10^{-7}$, $d'_e = 2.06$, 95% confidence interval = $[-\infty, -0.047]$), and feed-forward goal-directed model ($n = 40$; mean model-linear response difference (−0.00030) is not significant: two-sided one-sample t-test against 0: $t(\mathrm{df} = 39) = -0.37$, $p = 0.71$, 95% confidence interval = $[-0.0019, 0.0013]$). *Filled circle*: significant boosting or suppression. *Hollow circle*: no significant effect. *Square*: deterministic model. Source data are provided as a Source Data file.

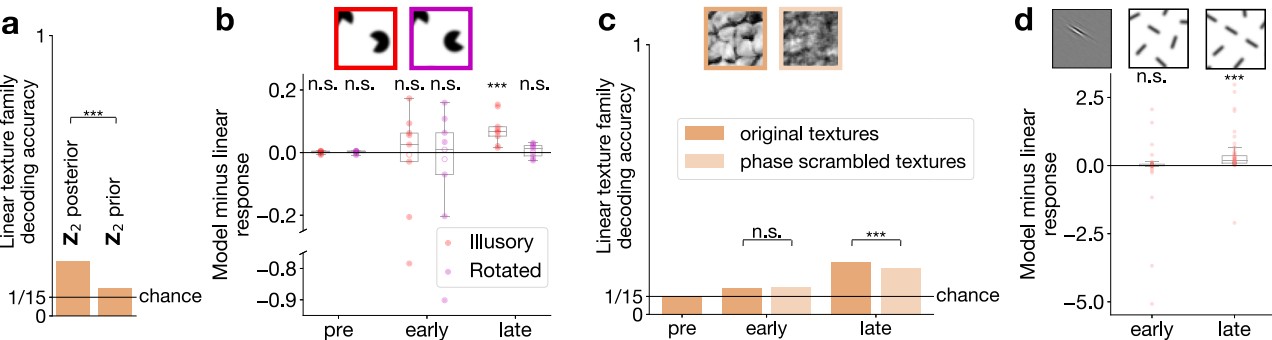

**Fig. 5 | Top-down influences shaping the representation in $Z_1$. a** Mean performance of texture family decoder from mean responses of $Z_1$ with intact inference (0.1943) is higher than without stimulus-specific information at $Z_2$ (0.0986) ($n = 5$ fits from different random seeds; one-sided independent two-sample $t$-test: $t(\mathrm{df} = 8) = 70$, $p = 9.6 \times 10^{-13}$, $d'_e = 52.4$, 95% confidence interval = [0.093, ∞]). Bar heights: means; s.d. < 0.01 everywhere. **b** Qualitative time course of illusory contour responses in the same restricted $Z_1$ population in TDVAE as in Fig. 4g ($n = 9$ $Z_1$ units everywhere; two-sided one-sample $t$-test against 0 everywhere except late Illusory; pre Illusory: mean: 0.0011, $t(\mathrm{df} = 8) = 0.72$, $p = 0.49$, 95% confidence interval = [−0.0025, 0.0047]; pre Rotated: mean: 0.00077, $t(\mathrm{df} = 8) = 0.47$, $p = 0.65$, 95% confidence interval = [−0.0030, 0.0045]; early Illusory: mean: −0.068, $t(\mathrm{df} = 8) = −0.71$, $p = 0.50$, 95% confidence interval = [−0.29, 0.15]; early Rotated: mean: −0.091, $t(\mathrm{df} = 8) = −0.85$, $p = 0.42$, 95% confidence interval = [−0.34, 0.16]; late Illusory: mean: 0.076, one-sided one-sample $t$-test against 0: $t(\mathrm{df} = 8) = 4.7$, $p = 0.00076$, 95% confidence interval = [0.046, ∞]; late Rotated: mean: 0.0065, $t(\mathrm{df} = 8) = 0.92$, $p = 0.39$, 95% confidence interval = [−0.0098, 0.023]). Center line, median; box limits, upper and lower quartiles; whiskers, $1.5 \times$ interquartile range. *Inset*: example stimuli. **c** Qualitative time course of texture family decoding performance from intact and phase-scrambled versions of textures ($n = 5$ fits from different random seeds; mean accuracies ± stds: pre: 0.0680 ± 0.0007, early:

0.0986 ± 0.0025, early scrambled: 0.1007 ± 0.0012, late: 0.1943 ± 0.0012, late scrambled: 0.1709 ± 0.0020; two-sided independent two-sample $t$-test between early and early scrambled: $t(\mathrm{df} = 8) = −1.6$, $p = 0.15$, $d'_e = 1.19$, 95% confidence interval = [−0.0053, 0.00098]; one-sided independent two-sample $t$-test between late and late scrambled: $t(\mathrm{df} = 8) = 20.3$, $p = 1.8 \times 10^{-8}$, $d'_e = 14.8$, 95% confidence interval = [0.021, ∞]). Bar heights: means; s.d. < 0.01 everywhere. *Inset*: example stimuli. **d** Qualitative time course of contour completion experiment. *Top*: Illustration of stimuli. Optimally oriented, identical line segment is shown in the receptive field of a $Z_1$ neuron (*left*) under the condition that the neighboring visual field is filled with randomly oriented segments (*center*) or the neighboring segments continue the segment that covers the receptive field (*right*). *Bottom*: Qualitative time course of the response intensity difference between the contour completion and random conditions in the $Z_1$ population ($n = 57$ central, localized, medium wavelength $Z_1$ filters in TDVAE in both cases; early: mean effect size: −0.080, two-sided one-sample $t$-test against 0: $t(\mathrm{df} = 56) = −0.65$, $p = 0.52$, 95% confidence interval = [−0.33, 0.17]; late: mean effect size: 0.36, one-sided one-sample $t$-test against 0: $t(\mathrm{df} = 56) = 3.9$, $p = 0.00014$, 95% confidence interval = [0.21, ∞]). Center line, median; box limits, upper and lower quartiles; whiskers, $1.5 \times$ interquartile range. Source data are provided as a Source Data file.

connections, we simulated illusory contour responses in $Z_1$ by clamping $Z_2$ responses to zero. This also showed modulations similar to the shallow-VAE rather then those of TDVAE (Fig. 4g). To show that illusory contour responses are a consequence of learning in the task-general model, we repeated the experiment with the goal-directed feed-forward model. No illusory contour responses were observed in this goal-directed model (Fig. 4g).

In non-human primates, extensive evidence supports that stimulus which resonates with the sensitivities of higher-order visual areas produce top-down effects in V1[49,50]. A simple analysis of texture-level information in $Z_1$ provides a strong indication that statistics represented at $Z_2$ flows back to $Z_1$ through top-down connections: we performed texture decoding both with intact $Z_2$ and with removing stimulus information from it. This analysis confirmed that the small but significant decodability of texture-family information in $Z_1$ was a consequence of top-down feedback from $Z_2$ (Fig. 5a). Direct ablation of V2 inputs is currently not widespread in neuroscience research, instead experimental studies use the temporal evolution of V1 signals to distinguish feed-forward and top-down components. We model this effect by assuming that without direct stimulus contribution the cortex samples the prior[12,32,51]. Thus, pre-stimulus responses were calculated by sampling the prior of both $Z_1$ and $Z_2$, while at early-stimulus responses we assumed that stimulus information reaches $Z_1$ but $Z_2$ responses still rely on the prior. Only late responses showed illusory activity in $Z_1$ (Fig. 5b), consistent with late-emergence of illusory contour responses in macaques[48]. Similar late-emerging top-down effects were identified when high-level statistics of stimuli was manipulated through phase scrambling (Fig. 5c) and when large-scale integration of visual elements was disrupted by manipulating contours (Fig. 5d, Supplementary Fig. 6c, d). These results are confirming results obtained from electrophysiology recordings from V1 of macaques when presenting stimuli matching those used in our experiments[49,50].

## Response correlations emerging through contextual priors

Illusory contour, contour integration, and phase scrambling experiments demonstrate how contextual priors affect V1 neurons through features represented higher in the cortical hierarchy, V2 in our specific case. These contextual priors carry information about the regularities of the surroundings of a neuron's receptive field, such as the tendency of another neuron to be jointly active in the local environment. These regularities in the local environment can change from image to image and thus the changing contextual prior dictates different co-activation patterns between neurons (Fig. 6a, Supplementary Fig. 7a). Such stimulus-specific co-activation patterns are captured in correlations between neuronal responses to individual images and thus noise correlations. Indeed, across-stimulus dissimilarity of noise correlations is larger than that expected from estimating noise correlations from a finite sample (Fig. 6b). Noise correlations provide us with a tool to identify and characterize top-down influences as contextual priors contribute to these. Importantly, if regularities in a set of images are similar, the contextual prior and thus the noise correlations are not changing, therefore texture family-specific activity in V2 is expected to induce texture family-specific noise correlations in V1 (Fig. 6c).

We measured the texture-family specificity of noise correlations in the TDVAE model by constructing linear decoders that we applied to noise correlation matrices of a small $Z_1$ neuron population (Fig. 6d). In order to demonstrate the contribution of top-down interactions, we capitalize on a specific property of TDVAE. At a given stimulus and a specific level of $Z_2$ response, the variational posterior is chosen to be a Laplace distribution, and specifically a variant that is free of correlations. However, as $Z_2$ response is not a given number but a distribution over possible values as defined by the posterior, variations in $Z_2$ might induce covariations in $Z_1$. Consequently, in TDVAE, it is the top-down influence that is exclusively responsible for the covariability of $Z_1$ neurons. To disentangle top-down correlations from private

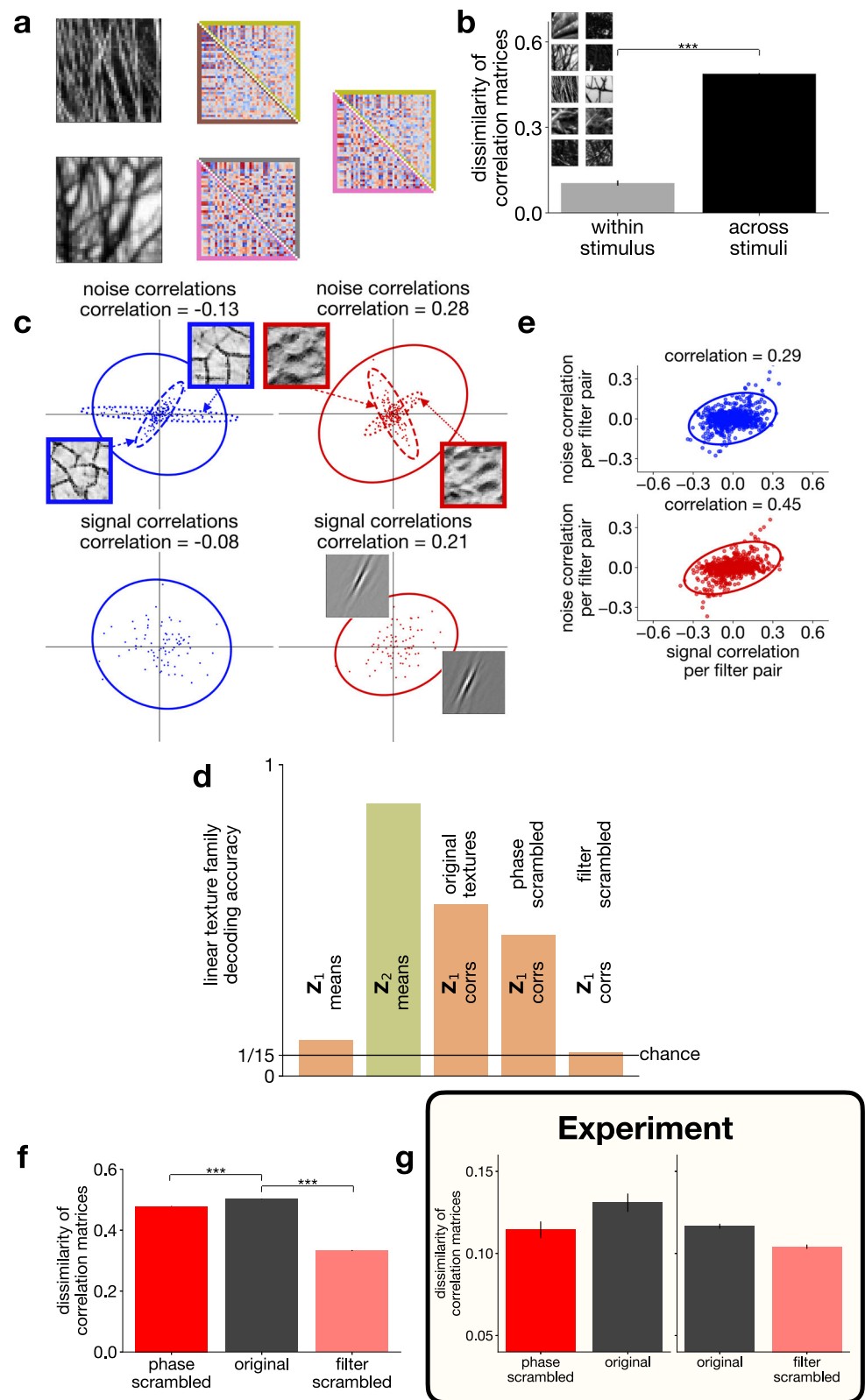

variability, we use the changes in the mean of the posterior of $Z_1$, which is resulting from sampling $Z_2$, to assess correlations. Additional private variability reduces the magnitude of noise correlations but it does not affect the overall trends reported in this section. As $Z_2$ response means show linear decodability of texture family information, linear decoding of texture family from $Z_1$ correlations was also highly effective (Fig. 6d). In addition, stimulus manipulations weakening the texture

representation in $Z_2$ like phase and filter scrambling also reduce the texture family decodability from $Z_1$ correlations (Fig. 6d). In contrast to $Z_1$ response correlations, $Z_1$ response means are worse suited to linear decoding of texture family information (Fig. 6d).

The contextual prior reflects the regularities of the environment that are specific to the particular context. For texture families, the tendency of a pair of neurons to be jointly active for different texture

**Fig. 6 | Contribution of top-down influences to response correlations in $Z_1$. a** $Z_1$ noise correlations for $n = 40$ neurons with central, localized, medium wavelength receptive fields for two example natural images (*left*). *Center*: Effect of sampling noise. Comparison of noise correlations from two independent sets of trials ($n = 80$ each, *lower* and *upper* triangles, respectively). *Right*: across-image noise correlation differences. Data taken from the central column (*border colors* matching those in the middle column). **b** Within- and across-image dissimilarity of noise correlations (from $n = 80$ trials each), averaged over 10 images (*inset*). Mean dissimilarity of noise correlations within images (0.104) is smaller than across images (0.488) (one-sided independent two-sample $t$-test, $t(\text{df} = 188) = -38.5$, $p = 1.9 \times 10^{-91}$, $d'_e = 13.1$, 95% confidence interval = $[-\infty, -0.37]$). Data are presented as mean values ± s.e.m. **c** *Top*: Response covariance of two $Z_1$ neurons to individual images. *Dashed* and *dotted covariance ellipses* correspond to the two *inset images*. *Solid ellipses*: average covariance ellipses for $n = 80$ images per texture family. *Left* and *right* columns: two example texture families. *Text labels*: per-family mean noise correlation coefficients. *Bottom*: Trial-averaged mean responses of the same $Z_1$ units to $n = 80$ images (dots) from the same texture family as top. *Covariance ellipses, text labels*: signal correlations calculated from these responses. **d** Texture family decoding performance from $Z_1$ ($n = 100$) and $Z_2$ ($n = 6$) mean responses (as in Fig. 2d) and $Z_1$ noise

correlations (from the same $n = 100$ $Z_1$ units as for $Z_1$ means). Decoders were constructed for texture images, as well as phase and filter scrambled versions of these. Bar heights: means for $n = 5$ random seeds; s.d. < 0.01 everywhere. *Black line*: chance performance. **e** Relationship between noise and signal correlations for a population of $Z_1$ ($n = 40$ $Z_1$ units) neuron pairs (*dots*) for two texture families. Colors and data as in panel (**c**). **f** Across-stimulus dissimilarity of noise correlations (20 natural images, four noise correlation matrices per image, each calculated from 80 trials). Mean dissimilarity for intact images (0.503) is larger than for phase scrambled images (0.479) (one-sided paired two-sample $t$-test, $t(\text{df} = 3039) = 25.4$, $p = 3.2 \times 10^{-129}$, $d'_e = 0.54$, 95% confidence interval = $[0.023, \infty]$), and also larger than for filter scrambled versions (mean for filter scrambled images: 0.333, one-sided paired two-sample $t$-test, $t(\text{df} = 3039) = 87.4$, $p < 1 \times 10^{-100}$, $d'_e = 2.6$, 95% confidence interval = $[0.17, \infty]$). Bar heights: means; error bars: s.e.m. **g** Same as (**f**) in V1 multiunit recordings from two macaque monkeys. The two comparisons were calculated from data recorded in distinct experimental sessions. Number of noise correlation dissimilarity metric data points: *phase scrambled* vs. *original*: $n = 645$; *filter scrambled* vs. *original*: $n = 7469$. Data are presented as mean values ± s.e.m. Regenerated from the data of[34] with permission from the authors (M. Bányai, personal communication 2025). Source data are provided as a Source Data file.

---

images, referred to as signal correlation, is expected to be captured by the contextual prior of a well trained model. Indeed, Karklin and Lewicki argued that the correlations between mean filter responses is characteristic to texture families[52], and we also found that such texture family-specific signal correlations were present between an example pair of neurons (Fig. 6c). Thus, the contextual prior is related to two different forms of joint response statistics: the signal correlations, measuring correlations between mean responses across images, and the noise correlations, measuring correlations between neurons following multiple presentations of the same image. Therefore, we measured for every texture family the relationship between these two quantities for a large number of $Z_1$ neuron pairs. We found strong dependence of noise correlations on signal correlations in the responses of pairs of $Z_1$ neurons (Fig. 6e, Supplementary Fig. 7b).

We test the consequences of the proposal that noise correlations are shaped by contextual priors in a modified setting. We argued that stimulus-specific noise correlations arise due to top-down feedback that carries information about high-level inferences. Consequently, removal of high-level structure is expected to result in reduced stimulus-specificity of noise correlations in $Z_1$. Both our earlier analyses (Figs. 4, 5) and electrophysiology data[49] support the idea that texture-statistics related information is delivered to $Z_1$ through top-down feedback from $Z_2$. The former analysis concerned top-down induced changes in response mean in $Z_1$ neurons and here we extend this to top-down influences on noise correlations. To test this, we used the same stimulus manipulations that we used for exploring the properties of $Z_1$ and $Z_2$ representations: phase and filter scrambling of natural stimuli (Methods). We calculated the dissimilarity of noise correlation matrices for natural images and for phase- and filter-scrambled versions of the same natural images (Methods). We found reduced dissimilarity for phase scrambled versions of natural images and also for filter scrambled versions (Fig. 6f). Similar reduction in the dissimilarity of noise correlations for phase- and filter-scrambled versions of natural images was found in the V1 of macaques[34] (Fig. 6g). Note, that larger magnitude of dissimilarity, i.e. higher stimulus specificity of noise correlation, in the model is a direct consequence of focusing on quantifying correlated variability and removing private variability from analyses. Sampling the posterior of $Z_1$ reduces the measured stimulus specificity of noise correlations (Supplementary Fig. 7c, d), making the magnitude comparable to the experimentally obtained values. The exact level of reduction depends on the details of how sampling is performed, as well as experiment details, including trial length. Given the 400-ms time window of the experiments in[34], we have explored a number of variants in sampling (Supplementary Fig. 7c,d). While systematic changes were introduced by the relative

number of samples from $Z_1$ and $Z_2$, relative differences of correlation dissimilarities across stimulus types were consistent. In summary, similar stimulus statistics dependence of noise correlations in $Z_1$ of TDVAE and V1 of macaques provide support that contextual priors emerge as a consequence of hierarchical inference.

## Discussion

In this paper we extended deep learning models of the ventral stream by developing a deep generative model. The developed TDVAE model departs from the feed-forward architectures of earlier deep learning accounts as it naturally accommodates top-down interactions: top-down connections serve the implementation of contextual priors for hierarchical inference. Importantly, to investigate how neural representations emerge and shape computations, our normative model was end-to-end trained on natural images alone and neural representations could be studied in a setting where no free parameters were available. The hierarchical organization of TDVAE showed strong alignment with the early hierarchy of the ventral stream of the visual cortex: linear features of $Z_1$ reflected simple cell characteristics of V1, while the texture representation in $Z_2$ closely matched texture sensitivities of V2 neurons in primates and rodents. The match between the hierarchy of representations made it possible to investigate top-down computations. Our deep generative model was shown to reproduce key experimental signatures of top-down computations. Importantly, studying the computations in TDVAE enabled us to gauge how feedforward and top-down contributions interact in V1, predicting changes both in response intensities of individual neurons and in stimulus-dependent changes in noise correlations. Beyond reproducing a wide range of experimental findings, we demonstrate that the framework opens a new window on exploring the representation, computations, and the circuitry of the early visual system.

Generative models have long been implicated in perceptual processes as these establish an essential framework for learning in an unsupervised manner. Conceptually, generative models summarize our knowledge about the world and describe the way observations are produced as a result of the interactions between (a hierarchical set of) features. Perception is assumed to invert the generative process by inferring the features that underlie observations[9,24]. However, modeling tools to investigate neural processes have been limited. Significant contributions to understanding the computations in the visual cortex were made with non-hierarchical, linear models[29], manually designed hierarchical extensions[53,54] and linear hierarchical models[55,56]. Non-hierarchical extension of the linear models were shown to account for extra-classical receptive field effects and response variability[37,38,53,57]. Further, a mild hierarchical extension that explicitly modeled

dependencies between features represented by receptive fields in natural images[58], when applied to V1, could account for complex cell properties[52]. Here we propose to overcome the obstacle of training models that are both hierarchical and flexible enough to accommodate the complexity of natural images by recruiting the recently developed Variational Autoencoders (VAEs) from machine learning[25,26] and extending these towards hierarchical computations[27,28]. Non-hierarchical versions of VAE have previously been applied to various stages of the ventral stream, including early visual cortex[30,59] and inferotemporal cortex[60]. Interestingly, a hierarchical VAE proved useful for modeling computations along the dorsal stream[61] and could establish relationships to responses in MT but top-down computations were not addressed.

The proposed TDVAE architecture relies on a limited set of assumptions while remaining as faithful as possible to the principles of probabilistic inference. In order to stick with the guarantees of VAEs, we did not rely on extensions of the framework that encourage disentanglement[60,62], regularization methods that might distort the learned representations (see Methods), or methods that help limiting the number of learned parameters (convolution). TDVAE implements an approximate form of probabilistic inference, which we related to neural response statistics through the sampling hypothesis. A range of studies delivered support that stochastic sampling accounts for patterns in response variability in V1 and beyond[7,12,37,38,63–67]. In our study we obtained samples from the variational posterior of the recognition model and performed this sampling with a fixed neural architecture, yielding the so-called amortized inference. Alternatively, inference can be performed by directly sampling the learned generative model. We performed such analysis to validate the recognition model but relied on sampling the recognition model throughout the paper. We motivate the chosen approach by three arguments. First, while the two methods yield the same posterior in a well-calibrated model, the variational posterior provides a useful tool for disentangling the contribution of top-down influences to both response means and response correlations. This feature was particularly effective when interpreting stimulus-dependent modulations of noise correlations. Second, by keeping our approach as close as possible to standard machine learning approaches, the framework can be naturally extended in further directions. Third, we found the circuitry implied by the recognition model to be a good inspiration for interpreting layer-by-layer computations. In our study, TDVAE was directly used to model neural responses. Earlier studies using generative models of natural images have distinguished real-valued responses that were used to model membrane potentials and firing-rate transformed outputs to model spiking activity[37,38]. A Poisson-distributed version of VAE is an appealing variant to model spiking activity[59] but modeling stimulus-dependent changes in noise correlations needs careful evaluation of private variability of model neurons as this affects quantitative fits to neural data[68].

How neural circuits implement these computations is a critical question. The presented model provides a computational/algorithmic level account of the processes taking place in the early visual cortex. The approach is computational in the sense that it provides a normative account of how the connections from V2 to V1 need to be organized in order to perform hierarchical inference. It is algorithmic in the sense that sampling is assumed to perform inference. The choice of Variational Autoencoders to approximate inference is not considered to be essential but we propose it to be a convenient platform for the investigation of complex computations[59,60]. However, the approach hinders some circuit-level motivations, which can be addressed in an implementation-level model[67,69]. Such an approach has demonstrated that the combination of amortized inference and sampling to perform inference can provide insights into the recurrent interactions within the lateral structure of V1[67,70] but a version that could perform hierarchical computations has not been explored so far.

The deep generative modeling approach presented here complements the more traditional works on generative models of vision that directly rely on statistical insights to construct the neuronal circuitry[57,71]. While the multi-layer perceptron networks that implement the computational components of the recognition and generative models in TDVAE are endowed with the flexibility necessary for high-complexity models, these prevent obtaining direct mechanistic insights that more traditional approaches offer[71]. These mechanistic insights might help to establish inductive biases for deep generative models for more efficient learning of natural image statistics[72]. Note that the information that reaches $Z_1$ through the top-down influences of $Z_2$ are not implemented in a recurrent loop, instead, the feed-forward pathway that is shared between $Z_1$ and $Z_2$ is relaying information to $Z_2$ before feeding it to $Z_1$. This single-step top-down pass avoids highly recurrent refinement of the responses between V1 and V2. Experiments relying on time delays between neural responses in V1 and V2 are compatible with our account but more precise experimental data can identify more elaborate recurrent computations.

Top-down interactions have previously been implicated in an alternative computational framework to our deep generative models, predictive coding[73,74]. The two frameworks have strong parallels: both are unsupervised, rely on a generative model, and are hierarchical. The two frameworks can be distinguished based on the goal top-down connections serve. While in our account, top-down interactions contribute to rich inferences that can serve flexible task execution[10,21], in the predictive coding framework top-down connections serve an efficient representation of the stimuli[75]. In our framework inference is inherently probabilistic. Probabilistic inference augments classical inference frameworks by representing the uncertainty associated with the interpretation of the incoming stimuli[12,76]. Theoretical arguments provide strong support for probabilistic inference and have also gained strong empirical support from behavioral studies[11,77]. More importantly, the representation of uncertainty has been identified in neuronal responses in V1[76] and such probabilistic computations have explained a wide range of phenomena[37,38,63,67]. Representation of uncertainty is not part of the existing formulations of predictive coding models (inference is performed at the maximum a posteriori level[78]), albeit proposals have been made to extend the original framework of predictive coding in this direction[79]. In practice, our results on stimulus-dependent correlations are tightly linked to a probabilistic representation and are therefore beyond the scope of predictive coding accounts.

In our hierarchical model, the computational graph of the recognition model provides insights into the potential circuitry serving the computations (Fig. 1). Feed-forward information reaching $Z_2$ passed through a relay that precedes the $Z_1$ layer. Neurons implementing this relay may correspond to layer 4 neurons in V1. Feedback from $Z_2$ can be interpreted as projections to the infragranular layers of V1, which is supported both by anatomical data[43] and the progression of texture information across laminae[49]. Recent experiments have highlighted that so-called feedback receptive fields in layer 2/3 are inherited through top-down interactions, and these are specific to layer 2/3 but are absent from layer 4[80,81]. In line with this, we found that $Z_1$ could inherit specific information from $Z_2$ as texture sensitivity was affected by the integrity of top-down feedback. Layer 2/3 of V1 is well positioned to integrate feed-forward and top-down information, consequently we propose to identify $Z_1$ with this layer. As texture was shown to be represented in V2, and evidence supports that texture-related information reaches V1 with delay[49], this result also confirms that the feedback from $Z_2$ to $Z_1$ in TDVAE corresponds to across-area rather than within-area computations. It is important to note that feed-forward information flow towards $Z_2$ is not implemented through $Z_1$. While feed-forward projections from V1 have been identified from layer 4 neurons[41], layer 2/3 provides a strong source of feed-forward information flow[40]. In our paper we did not consider how the V1 code is

read by downstream areas for computational purposes, in which layer 2/3 projections certainly have a central role. Interestingly, such a feed-forward component would contribute to a more direct recurrent loop between V1 and V2, which could enrich computations. Note, that the neural networks that implemented the components of TDVAE ($\mathbf{Z}_1$-ff, $\mathbf{Z}_2$-ff, $\mathbf{Z}_1$-INT, $\mathbf{Z}_1$-TD) were considered as general computation elements. Therefore we made no attempts to link constituent neurons to the neurons of the ventral stream but this can be an inviting subject of future research.

While our model did not include explicit feedback loops, these are ubiquitous in the cortex. Such recurrent interactions have been investigated through generative models of natural images in the context of modeling collective modulations of neuronal responses by stimulus contrast, which were shown to contribute to nonlinear response properties of V1 neurons[38,53,57]. Interestingly, this neurally well-motivated generative model has recently been shown to extend and improve the variational autoencoder framework[72], which can also be integrated in the future with the hierarchical VAE framework discussed here. Another role of recurrence has been proposed for performing fast and accurate inference in generative models of natural images by neural networks[67]. Recurrent computations have gained traction in deep discriminative models too and were shown to surpass purely feed-forward models in prediction of neural activity[82]. More specifically, recurrence was shown to contribute to such predictions when stimuli were perceptually challenging[5]. This result is in line with our argument that recurrent activity helps under perceptual uncertainty. Indeed, in human MEG recordings, contribution of recurrent computations was shown to be important when processing occlusion, another source of perceptual uncertainty[83]. This provides an insight that beyond the uncertainty arising through the hierarchical structure of natural images, uncertainty arising from the interactions of different visual elements might rely on recurrent instead of top-down computations. Generative models that combine hierarchical as well as lateral interactions have been proposed in machine learning[84], which can provide inspirations for further extensions.

We emphasized the role of top-down interactions in inferring perceptual variables through establishing contextual priors. Importantly, priors that are not dependent on a context can be efficiently implemented without the need of top-down connections[85–87]. The advantage of top-down induced priors is their flexibility as these can depend on the wider context and such can depend on the stimulus itself. In contrast with the implication of top-down connections in perceptual inference, a wide array of studies consider attentional modulation as an important factor contributing to top-down interactions[88]. Reconciling these alternative interpretations is a critical challenge. We argue that considering task-relevant variables besides perceptual variables can be interpreted in the same inference framework[66]. Indeed, top-down delivered information in the early visual cortex was shown to be specific to the task being performed in humans[89] and also has signatures in mice[90]. Also, it has been argued that attentional effects can rely on a similar computational scheme, in particular hierarchical inference, to that proposed here for perceptual inference[64,91]. Interestingly, the top-down modulation of noise correlation patterns has also been demonstrated under changing task conditions[92]. The question how perceptual and task-related variables are integrated in the circuitry that establishes top-down interactions remains a largely open question but recent studies indicate that they may share the same communication pathways[93].

We argued that the emergence of illusions can distinguish goal-directed models from task-independent models that perform probabilistic inference through establishing contextual priors. Indeed, a variety of perceptual illusions have been reported to be at odds with goal-directed models[94]. Interestingly, modeling works, which had a focus on feedback but not on the cortical representation, highlighted an additional perceptual effect related to the Kanizsa illusion that

relied on higher-level shape information[95,96]. This is in line with reports that cortical areas beyond V2 are central to the Kanizsa effect[97]. Indeed, higher-level features that are represented beyond V2 could contribute to top-down boosting of illusory responses, but the critical contribution of texture-representing region became evident through the evidence that causal intervention in the lateromedial region of mice eliminated illusory responses[98]. Hierarchical Bayesian models have been proposed to account for illusory contour responses[99,100], which implicate recurrent connections from higher visual areas. While these hand-crafted models can capture the signatures of the Kanizsa experiment, it is an end-to-end trained model that can account for the interplay between the learned representations of a neural network and the computations taking place in them. While illusions are key signatures of the contextual priors that emerge through top-down interactions, these interactions have wider ranging effects. Recently, modulation of V1 neurons by the visual context of their receptive field has received support from a hierarchical inference account[54]. Disentangling the contributions of feed-forward and feedback components is thus a key challenge of systems neuroscience but novel tools promise to break this impasse by providing a quantitative testbed for theories[101].

In our study we found a robust texture representation in a layer homologous to V2. In our approach we did not encourage learning texture-like representations since the model was trained on natural images. We argue that multiple factors contributes to the properties of the representations emerging at different levels of the hierarchy. In the TDVAE, low-level representation was constrained by the inductive bias applied to the generative model. Texture statistics are well characterized by a spectrum of second-order statistics[45]. Therefore a limited computational complexity model can capture textures. Thus, we expect that a generative model with a deeper hierarchy but with a generative layer that is still constrained to second-order statistics will learn similar texture representation. Higher levels of the hierarchy that can recruit further nonlinearities will be able to capture more nonlinear features. We found similar representations in variants that featured a more complex generative model. We argue that this might highlight the contribution of another factor: image size. The statistics relevant to textures have a characteristic scale. Intuitively, more complex features might have statistical signatures spanning even larger spatial scales. We speculate that the limited patch size actually affected the features TDVAE developed sensitivity to. Taken together, our results demonstrate a very robust texture representation that emerges for natural images in a nonlinear generative model. The constraints implicit in our implementation of TDVAE contributed to this robustness, and provide the necessary insights to build generative models with deeper hierarchies that capture more complex features and higher-level processing in the visual cortex. Several machine learning studies indicate that developing multiple levels of hierarchy is a viable route with VAEs[27,28]. The two-layer TDVAE can be extended to higher levels of the hierarchy. Distinguishing the computations at different layers of the hierarchy can be achieved by inductive biases (similar to the linear limitation we implemented in one component of the generative model) or explicit architectural constraints (such as limiting the statistics by limiting the visual field a neuron has access to[102]). The nature of the inductive biases shaping higher level representations is subject to further research.

## Methods

### Generative models

We study Markovian two-layer hierarchical latent variable generative models that learn the joint distribution of observations and latent variables in the form

$$p_\theta(\text{image}, \mathbf{Z}_1, \mathbf{Z}_2) = p_\theta(\text{image} \mid \mathbf{Z}_1) \cdot p_\theta(\mathbf{Z}_1 \mid \mathbf{Z}_2) \cdot p_\theta(\mathbf{Z}_2) \qquad (2)$$

where image denotes the pixel intensities of the modeled images, $\mathbf{Z}_1$ and $\mathbf{Z}_2$ denote stochastic model layers modeling biological neurons in visual cortical areas V1 and V2, respectively, and $\theta$ denotes the parameters of the generative model. To make inference tractable, we train Variational Autoencoders (VAEs)[25,103] which supplement the above generative model with a separate recognition model $q_\Phi(\mathbf{Z}_1, \mathbf{Z}_2 \mid \text{image})$ with parameters $\Phi$ that establishes a variational approximation of the true posterior distribution $p_\theta(\mathbf{Z}_1, \mathbf{Z}_2 \mid \text{image})$.

In general, the tractable Evidence Lower Bound (ELBO) for a two-layer VAE can be written as

$$\mathcal{F}(\text{image}, \theta, \Phi) = \mathbb{E}_{q_\Phi(\mathbf{Z}_1, \mathbf{Z}_2 \mid \text{image})}\left[\log p_\theta(\text{image} \mid \mathbf{Z}_1)\right] - \mathrm{KL}\left[q_\Phi(\mathbf{Z}_1, \mathbf{Z}_2 \mid \text{image}) \,||\, p_\theta(\mathbf{Z}_1, \mathbf{Z}_2)\right] \quad (3)$$

where $\mathrm{KL}[\cdot \,||\, \cdot]$ denotes the Kullback–Leibler divergence of two probability distributions.

In our main model, TDVAE, we factorize the variational posterior in the following way:

$$q_\Phi^{\mathrm{TDVAE}}(\mathbf{Z}_1, \mathbf{Z}_2 \mid \text{image}) = q_\Phi(\mathbf{Z}_1 \mid \text{image}, \mathbf{Z}_2) \cdot q_\Phi(\mathbf{Z}_2 \mid \text{image}). \quad (4)$$

This leads to the recognition model architecture depicted in Fig. 1f and the following objective function (ELBO):

$$\mathcal{F}_{\mathrm{TDVAE}}(\text{image}, \theta, \Phi) = \mathbb{E}_{q_\Phi(\mathbf{Z}_1 \mid \text{image}, \mathbf{Z}_2) \cdot q_\Phi(\mathbf{Z}_2 \mid \text{image})}\left[\log p_\theta(\text{image} \mid \mathbf{Z}_1)\right] \\ - \mathbb{E}_{q_\Phi(\mathbf{Z}_2 \mid \text{image})}\left[\mathrm{KL}[q_\Phi(\mathbf{Z}_1 \mid \text{image}, \mathbf{Z}_2) \,||\, p_\theta(\mathbf{Z}_1 \mid \mathbf{Z}_2)]\right] \quad (5) \\ - \mathrm{KL}\left[q_\Phi(\mathbf{Z}_2 \mid \text{image}) \,||\, p_\theta(\mathbf{Z}_2)\right].$$

The resulting computational graph is depicted in Supplementary Fig. 1. Note that the generative model component was used for training but our analyses exclusively focused on the recognition model. This choice is motivated by the fact that the recognition model is responsible for performing inference in the model, that is the recognition model delivers the (approximate) posteriors. Those are the posteriors that V1 and V2 neurons are assumed to represent during perceptual computations.

The computational graph shows that the $\mathbf{Z}_1$ layer is reached through $\mathbf{Z}_1$-ff, $\mathbf{Z}_2$-ff, $\mathbf{Z}_1$-TD, and $\mathbf{Z}_1$-INT by crossing the $\mathbf{Z}_2$ layer. As such this is a top-down architecture. While parts of this circuitry, notably $\mathbf{Z}_1$-ff, involve neurons that might reside in V1, the architecture is not recurrent as those are not the same neurons that participate in the feed forward and feedback information flows.

In our alternative model, ffVAE, the top-down interaction is not manifested in the factorization of the variational posterior:

$$q_\Phi^{\mathrm{ffVAE}}(\mathbf{Z}_1, \mathbf{Z}_2 \mid \text{image}) = q_\Phi(\mathbf{Z}_2 \mid \mathbf{Z}_1) \cdot q_\Phi(\mathbf{Z}_1 \mid \text{image}). \quad (6)$$

This results in the following objective function:

$$\mathcal{F}_{\mathrm{ffVAE}}(\text{image}, \theta, \Phi) = \mathbb{E}_{q_\Phi(\mathbf{Z}_2 \mid \mathbf{Z}_1) \cdot q_\Phi(\mathbf{Z}_1 \mid \text{image})}\left[\log p_\theta(\text{image} \mid \mathbf{Z}_1) \right. \\ \left. + \log p_\theta(\mathbf{Z}_1 \mid \mathbf{Z}_2)\right] \\ + \mathrm{H}\left[q_\Phi(\mathbf{Z}_1 \mid \text{image})\right] \quad (7) \\ - \mathbb{E}_{q_\Phi(\mathbf{Z}_1 \mid \text{image})}\left[\mathrm{KL}[q_\Phi(\mathbf{Z}_2 \mid \mathbf{Z}_1) \,||\, p_\theta(\mathbf{Z}_2)]\right]$$

where $\mathrm{H}[\cdot]$ denotes the entropy of a probability distribution.

For baseline we also trained a non-hierarchical VAE which we call shallow-VAE and has the objective function

$$\mathcal{F}_{\mathrm{shallow-VAE}}(\text{image}, \theta, \Phi) = \mathbb{E}_{q_\Phi(\mathbf{Z}_1 \mid \text{image})}\left[\log p_\theta(\text{image} \mid \mathbf{Z}_1)\right] \\ - \mathrm{KL}\left[q_\Phi(\mathbf{Z}_1 \mid \text{image}) \,||\, p_\theta(\mathbf{Z}_1)\right]. \quad (8)$$

For clarity, in the following we will omit $\theta$ and $\Phi$ from the notation of the generative and recognition models, respectively.

## Architectural details

Inspired by models of V1 activity[29], we chose to implement the $\mathbf{Z}_1$ layer of latent variables as an overcomplete layer $(\dim(\mathbf{Z}_1) = 1800 > \dim(\text{image}) = 40^2 = 1600)$ with a sparse (Laplace distribution) prior that is in a linear generative relationship to observations: $p(\text{image} \mid \mathbf{Z}_1) = \mathcal{N}(\text{image}; \mathbf{A} \cdot \mathbf{Z}_1 + \mathbf{b}, \sigma_{\mathrm{obs}}^2 \cdot \mathbf{I})$ where $\sigma_{\mathrm{obs}}$ denotes the standard deviation of the independent Gaussian observation noise for which we used $\sigma_{\mathrm{obs}} = 0.4$ in each presented model. While the dimensionality of $\mathbf{Z}_1$ permitted learning an overcomplete representation, in practice the dimension of the learned representation (1256) was complete, such that the number of actively participating model neurons was equal to the number of independent dimensions (after the whitening of the images). In addition, we chose $\dim(\mathbf{Z}_2) = 250$ for the hierarchical models TDVAE and ffVAE. While this may seem like an artificially induced compression constraint compared to the 1800 units in $\mathbf{Z}_1$, you will see in Supplementary Table 3 that 250 dimensions were enough to accommodate the entirety of the learned representations in $\mathbf{Z}_2$.

The components of the generative and recognition models were parameterized through neural networks with the softplus activation function. We used fully connected networks (multilayer perceptrons, MLPs) instead of convolutional networks (CNNs) to avoid the indirect effects of CNNs on emerging representations. Supplementary Table 1 highlights the number of hidden units for each part of the computational graph in the shallow−VAE and the ffVAE models.

The computational graph of the recognition model for the TDVAE model (depicted in Supplementary Fig. 1) is nontrivial and contains further inductive biases. There are four MLPs defined for the recognition model as depicted in Fig. 1g. The first neural network $\mathbf{Z}_1$-ff maps the pixel space to a layer $\mathbf{L}_x$ that is shared between the computations of $q(\mathbf{Z}_2 \mid \text{image})$ and $q(\mathbf{Z}_1 \mid \text{image}, \mathbf{Z}_2)$. From $\mathbf{L}_x$, a second MLP ($\mathbf{Z}_2$-ff) computes the means and variances of the $q(\mathbf{Z}_2 \mid \text{image})$ distribution. The third MLP, $\mathbf{Z}_1$-TD transforms $\mathbf{Z}_2$ into a layer $\mathbf{L}_z$. We fuse the information from image and $\mathbf{Z}_2$ by concatenating $\mathbf{L}_x$ and $\mathbf{L}_z$ and applying an MLP $\mathbf{Z}_1$-INT to the concatenated layer to obtain the means and variances of $q(\mathbf{Z}_1 \mid \text{image}, \mathbf{Z}_2)$. The number of hidden layers and hidden units used in each MLP to calculate the means and standard deviations of the conditional generative and variational posterior distributions is shown in Supplementary Table 2 for the different TDVAE models referenced in the paper.

As detailed in Figs. 4, 5 and the accompanying text, $40 \times 40$-pixel image patches proved to be too small for studying certain top-down effects with the TDVAE model family. For this reason, we created two larger, $50 \times 50$-pixel models called shallow-VAE-50px and TDVAE-50px that are proportionally upscaled versions of the $40 \times 40$-pixel shallow-VAE and TDVAE models, respectively (see Supplementary Tables 1, 2, 3, and 4). The qualitative properties of the upscaled TDVAE models match the $40 \times 40$-pixel models, so we omit them from the discussion below. In the paper, we use the $50 \times 50$-pixel models only for the illusory contour and contour completion experiments; all other results were reached with the $40 \times 40$-pixel models.

Finally, we assessed the potential effect of the shared $\mathbf{Z}_1$-ff mapping by comparing a standard $20 \times 20$-pixel TDVAE model with shared encoding called TDVAE-20px and a corresponding 'TDVAE-20px non-shared' model (for model parameters, see Supplementary Table 2). In the latter, $\mathbf{Z}_2$-ff took its input from the input image instead of the output of $\mathbf{Z}_1$-ff, and its MLP layers were prepended with an MLP with the same dimensions as $\mathbf{Z}_1$-ff (for these notations, see Fig. 1g).

## Inference in the hierarchical generative model

The recognition model directly provides the essential tools to perform inference in the model. The amortized variational inference inherent in the VAE framework yields a normal-distributed approximation of the true posterior distribution $p_\theta(\mathbf{Z}_1, \mathbf{Z}_2 \mid \text{image}) \approx q_\Phi(\mathbf{Z}_1, \mathbf{Z}_2 \mid \text{image})$. To relate the inference of the model to neural responses in V1 and V2,

marginal distributions are required. Marginal posterior for $\mathbf{Z}_2$ is directly provided by our choice of partitioning, as the TDVAE learns the function $q_\Phi(\mathbf{Z}_2 \mid \text{image})$. For the marginal posterior of $\mathbf{Z}_1$, however, only $q_\Phi(\mathbf{Z}_1 \mid \mathbf{Z}_2, \text{image})$ is available, necessitating marginalization over $\mathbf{Z}_2$:

$$q_\Phi(\mathbf{Z}_1 \mid \text{image}) = \int d\mathbf{Z}_2 \, q_\Phi(\mathbf{Z}_1 \mid \mathbf{Z}_2, \text{image}) \, q_\Phi(\mathbf{Z}_2 \mid \text{image}) \quad (9)$$

Instead of performing the integral, we obtain a Monte Carlo approximation by sampling $q_\Phi(\mathbf{Z}_2 \mid \text{image})$. Consequently, the marginal posterior for $\mathbf{Z}_1$ combines variational inference with sampling.

### Validation of the learned deep generative model

Optimizing the ELBO promises tightening approximation of the true posterior with the variational posterior, $q_\Phi(\mathbf{Z}_1, \mathbf{Z}_2 \mid \text{image})$. As it is critical to assess if this approximation is sufficiently tight, we have validated the model by calculating the moments of the posterior obtained defined by the generative model, $p_\theta(\mathbf{Z}_1, \mathbf{Z}_2 \mid \text{image})$, and that provided by the recognition model, $q_\Phi(\mathbf{Z}_1, \mathbf{Z}_2 \mid \text{image})$. We used self-normalized importance sampling (SNIS) to calculate the means of $\mathbf{Z}_1$ and $\mathbf{Z}_2$ of $p_\theta(\mathbf{Z}_1, \mathbf{Z}_2 \mid \text{image})$ and correlations between $\mathbf{Z}_1$'s. SNIS provides a convenient opportunity to calculate these by simply obtaining samples from the joint normal distribution of $q_\Phi(\mathbf{Z}_1, \mathbf{Z}_2 \mid \text{image})$ and then reweighing these samples by the fraction of the generative and recognition posteriors.

We performed SNIS for images from the 15 texture families, using 80 or three instances from each family for the means and correlations, respectively. Means and correlations were calculated by collecting a large number of samples ($4 \times 10^3$ for means and $38.4 \times 10^6$ for correlations) discarding the sample with the largest weight. SNIS-estimated moments were contrasted with the moments estimated from the recognition model (Supplementary Fig. 2).

### Relating inference in the generative model to neural activity

Features learned by the model are identified with the response properties of individual latent variables, termed model neurons, of the recognition model (Eq. (4)). Intensity of response strength of the model is identified with the intensity of the neuronal responses. As the generative modeling framework yields a probabilistic representation, we need to adopt a representational scheme to represent these probabilities. We adopt the sampling hypothesis[10,32,63], in which neural activity in any given moment in time represents a sample from this probability distribution and the sequence of samples yields a histogram of potential values from the distribution. The histogram is an approximate representation of the probability distribution, which can converge to the exact distribution as more samples are collected. This means that neural activity yields progressively better representation of the distribution as time passes. In biological neurons, an approximation of the time in which new samples are obtained can be estimated through the time constant of the response autocorrelation, which is approximately 20 ms[33] in the V1. Thus, in a trial of 500 ms 25 samples can be obtained, and a firing rate in this time window informs the experimenter about the mean of the samples. Trial averages of firing rates provide a more accurate estimate of the mean of the posterior distribution, while response variance carries information about the width of the posterior, i.e. the uncertainty of the estimate.

In contrast with firing rates, $\mathbf{Z}_1$ and $\mathbf{Z}_2$ can take positive as well as negative values. Optionally, the values of $\mathbf{Z}_1$ and $\mathbf{Z}_2$ can be identified with membrane potentials and thus the recorded firing rates can be considered as membrane potentials fed through a firing rate nonlinearity. In line with earlier works with VAEs[60], we chose to report the values of the latent variables instead of such firing-rate transformed versions in order to increase the transparency of the modeling approach. Earlier works have demonstrated that both membrane

potential and firing rate predictions of a generative model of natural images are in line with neural data on systematic changes in response mean and response variance[37], and therefore we focus on nontransformed values of model neuron activities. The firing rate transformation introduces a nonlinearity in responses, which can have substantial consequences on reading the information encoded by neurons[34,39,104]. While these affect the results quantitatively, the qualitative trends that our paper is focussing on remain unaffected by such change.

Noise correlations in neural responses are identified in the model with correlations in $q(\mathbf{Z}_1 \mid \text{image})$. In the specific VAE framework the variational posterior of $\mathbf{Z}_1$, $q(\mathbf{Z}_1 \mid \text{image}, \mathbf{Z}_2)$ is not correlated, thus the source of correlation is marginalization over $\mathbf{Z}_2$: variance in $q(\mathbf{Z}_2 \mid \text{image})$ results in correlated changes in $\mathbf{Z}_1$. In practice, instead of fully integrating $\mathbf{Z}_2$, we sample its variational posterior $q(\mathbf{Z}_2 \mid \text{image})$ and perform Monte Carlo integral:

$$q(\mathbf{Z}_1 \mid \text{image}) \approx \frac{1}{N} \sum q(\mathbf{Z}_1 \mid \mathbf{Z}_2, \text{image}), \quad \mathbf{Z}_2 \sim q(\mathbf{Z}_2 \mid \text{image}). \quad (10)$$

Limiting the source of correlations to result from variance in $\mathbf{Z}_2$ might seem as a limitation of the model, as the true posterior of $\mathbf{Z}_1$ might feature other statistical structure too. We argue that this setting actually provides clarity in interpreting how noise correlations emerge in the model and thus how top-down effects shape noise correlations.

### Model training details

We trained all our models with the Adam optimizer on natural image patches with a minibatch size of 128. We found that while the learned $\mathbf{Z}_1$ representation was robust against the tested regularization techniques (weight decay, gradient clipping[105], gradient skipping[27]), the learned $\mathbf{Z}_2$ representation was sensitive to them. To eliminate such regularization artifacts, we turned off weight decay and increased gradient clipping and skipping thresholds to have an activation frequency below $10^{-6}$.

For training our hierarchical VAE models TDVAE and ffVAE we applied $\beta$ annealing, which is a standard technique for training VAEs. This means that in their ELBO losses (5) and (7) we introduced a scaling factor $\beta$ in front of the last term. We started training with $\beta = 0.1$ for 1000 epochs and then linearly increased $\beta$ to 1 over the course of 1800 epochs and then continued training with the original ELBO loss. No $\beta$ annealing was used for the shallow−VAE model.

We found that our two-layer models TDVAE and ffVAE achieved significantly better ELBO values if we changed the learning rate parameter $\alpha$ whenever the validation loss reached a minimum. The following learning rate schedules were found to lead to the highest ELBO values (these are the models reported throughout this paper):

**shallow-VAE** $\alpha = 3 \cdot 10^{-5}$ for 12,000 epochs

**TDVAE linear integration** $\alpha = 10^{-5}$ for 40,500 epochs, then $2 \cdot 10^{-5}$ for 2100 epochs, and $2.85 \cdot 10^{-5}$ for 630 epochs

**ffVAE shallow nonlinearity** $\alpha = 10^{-5}$ for 7400 epochs, then $2 \cdot 10^{-5}$ for 600 epochs, $3 \cdot 10^{-5}$ for 320 epochs, and $10^{-6}$ for 10 epochs

**TDVAE shallow nonlinearity** $\alpha = 5 \cdot 10^{-5}$ for 4500 epochs, then $2.5 \cdot 10^{-5}$ for 9000 epochs

**TDVAE deep nonlinearity** $\alpha = 10^{-5}$ for 205,700 epochs, then $2 \cdot 10^{-5}$ for 2100 epochs, $3 \cdot 10^{-5}$ for 5700 epochs, and $5 \cdot 10^{-5}$ for 900 epochs

**TDVAE deep nonlinearity repeated** $\alpha = 10^{-5}$ for 10,800 epochs, then $2 \cdot 10^{-5}$ for 4500 epochs, $3 \cdot 10^{-5}$ for 3300 epochs, and $4 \cdot 10^{-5}$ for 3000 epochs

**shallow-VAE-50px** $\alpha = 1.5 \cdot 10^{-5}$ for 72,000 epochs

**TDVAE-50px shallow nonlinearity** $\alpha = 10^{-5}$ for 100,500 epochs, then $3 \cdot 10^{-5}$ for 800 epochs. In this case, changing the learning rate alone could not compress the $\mathbf{Z}_2$ representation to include only texture family encoding units. We promoted compression by increasing $\beta$ to 8.5 and then lowering it back to 1.

**TDVAE-20px** $\alpha = 10^{-4}$ for 6000 epochs, then $2 \cdot 10^{-4}$ for 1050 epochs, and $3 \cdot 10^{-4}$ for 300 epochs

**TDVAE-20px non-shared** $\alpha = 10^{-4}$ for 6000 epochs, then $2 \cdot 10^{-4}$ for 1050 epochs, $3 \cdot 10^{-4}$ for 750 epochs, and $3.4 \cdot 10^{-4}$ for 450 epochs

Validation ELBOs at the end of training and the number of active units in the learned representations in $\mathbf{Z}_1$ and $\mathbf{Z}_2$ are presented in Supplementary Table 3. Of all presented $40 \times 40$-pixel models, TDVAE with a deep nonlinearity in the higher level of its generative model $p_\theta(\mathbf{Z}_1 | \mathbf{Z}_2)$ reached the highest ELBO value, therefore it is the primary interest of this paper. For all models, $\mathbf{Z}_1$ learned a complete linear basis of the training data (see below on the effective dimensionality of the natural training data) while the nonlinear representations learned by $\mathbf{Z}_2$ had different dimensionalities for different models.

In the paper, we present one instance of each model type, except our main model 'TDVAE deep nonlinearity' the training of which we repeated starting from a different random seed. We found that the validation ELBO values (original: −1699, repeated: −1699) are well separated from those of our other $40 \times 40$-pixel models (next largest validation ELBO: −1705) and that the dimensions used by the learned representations (active $\mathbf{Z}_1$ units: original: 1256, repeated: 1256; active $\mathbf{Z}_2$ units: original: 6, repeated: 5) are consistent (see Supplementary Table 3). These results signal that our main model 'TDVAE deep nonlinearity' is well defined.

The models were trained on an Nvidia GeForce RTX 3080 Ti GPU.

## Discriminative model benchmark

When comparing our VAEs to a discriminative model we picked CORnet-Z[46] as a benchmark. This is a purely feedforward model which was optimized for object recognition. It has various blocks which were shown to have high representational similarity to V1, V2, V4, and IT. We used only the 'V1' and 'V2' blocks. A block in the model consists of two layers, a 'conv' and an 'output' layer, where the latter is calculated with pooling operations from the former. We found with reverse correlation experiments that the 'conv' layer has features reminiscent of Gábor filters (similar to our models' lower layer) while the output layer has more complex receptive field structure. Therefore we used the 'conv' layers to compare CORnet feature values to our model. The model expects $224 \times 224$-pixel RGB images as input. In our analysis we only fed non-whitened images to the model where only the blue channel had non-trivial pixel values.

## Natural image datasets

All presented models were trained on a 'training' natural image dataset compiled by the authors in a way similar to[30]. We took 320,000 random $40 \times 40$-pixel crops, 160,000 random $50 \times 50$-pixel crops, or 640,000 random $20 \times 20$-pixel crops from the first 1000 images of the van Hateren natural image dataset[106], matched their grand total pixel intensity histogram to the standard normal distribution, and applied ZCA whitening while keeping only $25\pi\%$ of the largest-eigenvalue PCA components. This made the effective dataset dimension 1256, 1963, or 314 instead of the number of pixels in an image patch, 1600, 2500, or 400, respectively. This whitening procedure is motivated by its resemblance to the spatial-frequency response characteristic of retinal ganglion cells[30,107], and also helps gradient based learning by uniforming variances along the main axes of variation. A statistically independent 'validation' set of 64,000 whitened natural images was created in the same way and used to monitor and report validation losses during model training (Fig. 2a). Another statistically independent 'test' set of 64,000 whitened natural images was also created in an identical way and was used throughout the paper for model evaluation.

## Texture image datasets

Our texture image datasets, created solely for model evaluation, were synthesized from 15 texture family seed images with the algorithm in[45].

Computational constraints limited the training image patch sizes to $40 \times 40$ or $50 \times 50$ pixels; therefore, we selected only starting images for which the dominant wavelength of synthesized textures fits into a $40 \times 40$ patch. We then randomly cropped a balanced set of $40 \times 40$-pixel or $50 \times 50$-pixel texture patches from the synthesized texture images, matched the grand total pixel histogram to the standard normal distribution for each texture family separately, and applied the whitening procedure used for natural images to the whole texture dataset. Figure 2c shows sample patches (before whitening) from the selected 15 texture families. Using this procedure, we created a balanced 'train' texture dataset of 600,000 patches in total and a statistically independent balanced 'test' texture dataset of 60,000 patches in total and used these for the texture family decoding experiments in Figs. 2d and 6d. In the rest of the paper, only the latter, 'test' texture dataset was used for model evaluation. (For more details, see[21].) For our CORnet discriminative benchmark we used non-whitened $224 \times 224$-pixel patches created in a similar way.

## Phase scrambling

Phase scrambling (Figs. 3g and 6f) was performed on the cropped $40 \times 40$ images (either from the van Hateren dataset or the synthesized texture image dataset) in the way described in[16] (and previously in[45]): we took the Fourier transform of the image, randomized the phase of each Fourier component, and took the real part of the inverse Fourier transform of the result. This manipulation did not change the spectral properties of the input images. After this we applied histogram matching and ZCA whitening as with the non-phase scrambled images.

## Filter scrambling

Filter scrambling (Fig. 6f) was performed in a way that is equivalent to low-level-synthetic (LL-synthetic) image generation in[34], to which it is compared in Fig. 6g. First, we calculated the $\mathbf{Z}_1$ posterior mean of the image under the shallow-VAE model, randomly permuted unit activations separately for active and non-active units, and generated the mean image from the result with the shallow-VAE model. This manipulation did not preserve the spectral properties of the input images. Finally, histogram matching and ZCA whitening were applied as above.

## Comparison texture image dataset

Figures 2e, 3c, e, and g compare TDVAE model predictions to experiments in[16] and[17]. These two papers are built on the same electrophysiology data set that we tried to mimic as described here. The papers present spike counts in a 100 ms time window for 102 individual V1 and 103 individual V2 neurons found in 13 macaque monkeys as a response to the same texture stimulus sequence with 15 texture families, 15 samples plus 15 phase scrambled samples from each texture family, and 20 presentations per image. In our comparison dataset, we randomly sampled 100 active $\mathbf{Z}_1$ dimensions as a model of the 102 V1 neurons in the experiment. The TDVAE model only has 6 active dimensions in $\mathbf{Z}_2$ which is less than the 103 experimental V2 neurons. As a substitute, we generated 100 random rotations of the 6D space and took the first components of the rotated 6D active $\mathbf{Z}_2$ space as a model of 100 V2 neurons. (Note that this procedure is consistent with the rotation-invariant unit normal prior in the $\mathbf{Z}_2$ layer of the TDVAE model.) To avoid correlated responses which are clearly not present in the experimental dataset, we sampled the stimulus-conditional model posteriors separately for all 100 $\mathbf{Z}_1$ and 100 $\mathbf{Z}_2$ model neurons, respectively. For stimuli, we used test texture images from the 15 texture families used throughout this paper, 15 samples plus 15 phase scrambled samples from each texture family, and 20 presentations per image. We named this dataset the 'comparison' dataset. To mimic the 100 ms time window in the experiment, we sampled the model posteriors in the following way. For $\mathbf{Z}_1$ units, we took one sample from the $\mathbf{Z}_2$ posterior and used that single $\mathbf{Z}_2$ sample in the $\mathbf{Z}_1$ posterior from which we averaged five samples. For $\mathbf{Z}_2$ units, we simply took a sample from the $\mathbf{Z}_2$ posterior.

## Linear and non-linear characterization of model units

Receptive fields of $Z_1$ and $Z_2$ units (Figs. 2b, 4a, 5d, 6c, Supplementary Figs. 3, 4, 5) were calculated with a method customary in visual neuroscience, reverse correlation estimated by spike-triggered averaging (STA,[108]), using 128,000 white noise images where each individual pixel intensity value was drawn from an independent Gaussian distribution with its position and scale matching the grand total mean and standard deviation of the pixels in the 'test' natural image dataset. In the case of the CORnet model, receptive fields of 64 independent features (corresponding to the depth of the convolutional map) were calculated. The receptive fields of the remaining features are translated versions of the computed ones due to the convolutional nature of the latent space. Projective fields (Supplementary Figs. 3, 5) were calculated with latent traversal along each model unit in $Z_1$ and $Z_2$ by taking two points symmetrically to 0 that differed only along the given unit dimension, generating a large number ($Z_1$: 128, $Z_2$: 2560) of mean images from these two points and taking the difference between the mean images for the two points. The second-order receptive fields of a $Z_2$ unit (Fig. 2b) were calculated with the spike-triggered covariance (STC) method after the mean has been projected out[108]. In this paper we do not aim for a detailed analysis of the TDVAE model in terms of the LNP (Linear-Nonlinear-Poisson) model underlying STA and STC analysis, we only demonstrate the applicability of the LNP model by plotting a representative eigenvector of the STC matrix. We had to use $10^8$ white noise images to get visible structure. The orientation tuning of a $Z_1$ or $Z_2$ unit was calculated from the posterior mean responses of the unit to sinusoidal gratings with different orientation, phase, wavelength, and contrast values. Response amplitude was defined as the difference between the maximum and minimum responses while sweeping over all phase values at fixed stimulus orientation, wavelength, and contrast. Figure 2b shows the response amplitude of the selected $Z_2$ unit at different stimulus orientations with stimulus wavelength 1/8 of the patch edge and stimulus contrast equal to that of the training set.

## Identifying active model units

In a given model, we classified $Z_1$ and $Z_2$ model units into active and inactive categories based on four criteria (Supplementary Fig. 3). (1) Variance of the mean response to the images in the 'test' natural image dataset was found to be orders of magnitude larger for active units than for inactive units. (2) Mean of the response variance (uncertainty) over the same image dataset also creates two clean clusters for active and inactive units. (3) Mean squared pixel intensities of receptive fields are clearly larger for active units than for inactive units. (4) Mean squared pixel intensities of projective fields are also larger for active units than for inactive units. We found that these four criteria consistently cluster units into the same active and inactive sets (for TDVAE, see Supplementary Fig. 3). The number of active $Z_1$ and $Z_2$ units is denoted in Supplementary Table 3 for all discussed models. In the paper we only consider active model units.

## $Z_1$ receptive fields in single-layer and hierarchical models

Learned $Z_1$ receptive fields are localized, oriented and bandpass in both the hierarchical TDVAE model and the single-layer shallow-VAE model (Supplementary Figs. 4, 5). We calculated two quantities for each active $Z_1$ unit for a more detailed comparison: filter center (estimated as the center of mass of the pointwise squared receptive field) and dominant wave vector (calculated from the maximum position of the squared magnitude of the Fourier transform of the receptive field). To increase the precision of these estimates, we used projective fields instead of receptive fields as they are very similar but less noisy (Supplementary Fig. 5). We found that the empirical distribution of filter centers and dominant wave vectors are very similar in the TDVAE and shallow-VAE models, making them equivalent complete bases in the space of natural training images, at least in terms of first-order probability densities.

## Multiunit texture family decoding

Linear decoding accuracies of texture family in Figs. 2d, 5a, c, and 6d were calculated with multinomial logistic regression. Each bar reports the mean of test classification accuracies calculated from $n = 5$ fitting sessions started from different random seeds. Standard deviation values are not plotted since they are all below 0.01 and would be invisible. In each case, the dependent variable was texture family and independent variables were derived from the 'train' and 'test' texture datasets as follows. Figure 2d: *Pixels*: independent variables were image pixel intensities. $Z_1$ *and* $Z_2$ *of all VAE models*: independent variables were the mean responses of all active units in the denoted layer of the denoted model to 'train' and 'test' texture images; for the untrained model, we used the same number of randomly selected units. *CORnet-*$Z_1$: independent variables were CORnet-Z V1 'conv' layer units in response to non-whitened $224 \times 224$-pixel texture images. To make comparison with our generative models, only model units which correspond to the central $40 \times 40$-pixel part of the image were considered ($20 \times 20 \times 64$ units). *CORnet-*$Z_2$: independent variables were $10 \times 10 \times 128$ CORnet-Z V2 'conv' layer units. Figure 5a,c: mean responses of all active $Z_1$ units in TDVAE. Figure 6d: $Z_1$ *means:* mean responses of 100 randomly selected active $Z_1$ units. $Z_2$ *means:* mean responses of the six active units. $Z_1$ *corrs* (*original textures*, *phase scrambled*, *filter scrambled*): independent variables were noise correlation coefficients between the mean responses of the same 100 randomly selected active $Z_1$ units as for $Z_1$ means, calculated from 16,384 presentations of every 10th image in the original, phase scrambled and filter scrambled texture image datasets, respectively. Overfitting was prevented with weight decay regularization where needed.

## Robust texture representation in the $Z_2$ model layer

The multiunit texture family decoding test was repeated for mean $Z_1$ and $Z_2$ responses in all models presented in the paper. As it is shown in Supplementary Table 4, decoding accuracies from $Z_1$ were around chance level $1/15 = 0.067$ for all models (we have 15 texture families) and decoding accuracies from $Z_2$ were close to perfect performance 1 for all hierarchical models, implying no texture representation in $Z_1$ and a robust texture representation in $Z_2$ for all studied models. Repeating the same analysis for subsets of active $Z_2$ units showed that texture family is only decodable from some of the $Z_2$ dimensions and not from others in TDVAE models (last two columns of Supplementary Table 4), revealing a low-dimensional (4–7 dimensions) learned texture representation in TDVAE models. In contrast, we found no axis-alignment in the $Z_2$ texture representation in the ffVAE model as texture family could be more or less decoded from all active $Z_2$ units (Supplementary Table 4). Since our primary model, TDVAE with deep nonlinearity, contains no non-texture-family-coding active $Z_2$ units, we do not analyze such units in the present paper. We also found that even in the discriminative model CORnet, decoding accuracy from the V1 layer was just above chance and decoding accuracy from the V2 layer was high.

## t-SNE visualization

In Fig. 2e we compare the low dimensional visualization of model responses in the TDVAE model with the low dimensional visualization of neural responses in[17] (Fig. 2f). To this end, we used the 'comparison' dataset defined above and the t-SNE visualization method utilized in[17]. t-distributed stochastic neighbor embedding (t-SNE)[109] attempts to minimize the divergence between the distributions of neighbor probability in the original high dimensional data space and the visualized low dimensional space. Similar to[17], we input a set of 225 data vectors (15 texture families, 15 images per family) to the t-SNE algorithm, each of which collected the mean responses of 100 neurons in a model layer to a stimulus. Like[17], we normalized the data so that, for each model unit, responses to the 225 images had a mean of 0 and standard deviation of 1, and executed the t-SNE analysis with a perplexity value of 30.

## ANOVA

We also repeated the ANOVA study in[17] in the context of the TDVAE model (Fig. 3c, d). The utilized nested ANOVA method partitioned the total variance of $Z_1$ and $Z_2$ model unit responses to the 'comparison' dataset into portions arising across families, across samples within a family, and across repetitions of the same stimulus. The ANOVA generates an F-statistic that captures the ratio of variances between each hierarchical level. For all model neurons we found the F-statistic to be significant for ratios of variance across repetitions and across samples as well as for ratios of variance across samples and across families. Following[17] we performed further analysis using the ratio between partitioned variance components. To obtain the variance ratio, we divided the percent variance across families by the percent variance across samples.

## Classification decoding

We also reproduced the decoding study in[17] with the TDVAE model (Fig. 3e,f; differences from[17] in parentheses). We used Gaussian decoders (experiment: Poisson decoders) to classify model responses to the 'comparison' dataset into 15 sample categories per family and, independently, into 15 texture family categories. On each iteration, we randomly selected a number of units from our model neurons in the $Z_1$ or $Z_2$ model layer. To compute performance in the sample classification task, we estimated the mean and variance of the response of each selected neuron for each of the 15 samples within each family by computing the sample mean and sample variance over four randomly selected repetitions out of the 20 repetitions in the dataset (experiment used 10 repetitions but that made the task trivial for the model). For the held-out 16 repetitions of each sample, we computed which of the 15 samples was most likely to have produced the population response, assuming independent normal distributions for the model units under the estimated mean and variance. We computed the average performance (% correct) over all samples and families, and repeated this process 100 times to get a performance for each population size (experiment used 10,000 here but estimated errors for the model were already small at 100 repetitions). To compute performance in the family classification task, we estimated the mean and variance of the response for each family over 8 randomly selected samples from the 15 different samples and for all repetitions. For each of the repetitions of the held-out seven samples, we computed which of the five families was most likely to have produced the population response. We computed the average performance over all repetitions and repeated this process 100 times to get a performance for each population size (experiment used 10,000 here but estimated errors for the model were already small at 100 repetitions). We computed performance measures for both tasks using population sizes of 1, 3, 10, 30, and 100 neurons.

## Phase scrambling induced response modulation in $Z_1$ and $Z_2$

We studied how reducing high-level stimulus structure affects model responses by reproducing the electrophysiology experiment in[16] (see Fig. 3g,h). First, as the experiment queries changes in response magnitudes, we took the absolute value of all model responses to undistorted and phase scrambled texture images in the 'comparison' dataset. Following[16], we then calculated per-neuron per-family mean response magnitudes by averaging these absolute responses over per-family samples and stimulus presentations. Next we calculated modulation indices for each 'comparison' $Z_1$ and $Z_2$ unit and texture family by dividing the difference between the above mean response magnitudes to undistorted and phase scrambled textures with their sum. As the final step, we averaged these per-neuron modulation indices over the 15 texture families and plotted separate histograms for $Z_1$ and $Z_2$ units (Fig. 3g).

## Filter selection for top-down experiments

While studying top-down effects with the TDVAE model, we found that the results were sensitive to certain limitations of the model such as pixel aliasing and finite patch size. To get rid of the ensuing artifacts, we restricted our top-down experiments to $Z_1$ dimensions with central, localized, medium wavelength generative fields, which we selected algorithmically with the following criteria: (1) centrality: center of mass of the pointwise square of the generative field falls in the central $(40\%)^2$ of the image patch, (2) locality: size $\sigma$ of the filter, calculated by fitting a cylindrical Gaussian profile with scale $\sigma$ to the pointwise absolute value of the $Z_1$ generative field by requiring the two to have the same moment of inertia, falls between 18% and 33% of the patch edge length, (3) medium-wavelength: the dominant wavelength, calculated from the maximum position of the squared magnitude of the 2D Fourier transform of the generative field, falls between 17% and 28% of the patch edge length. These criteria found 40 ($40 \times 40$-pixel TDVAE), 57 ($50 \times 50$-pixel TDVAE), and 53 ($50 \times 50$-pixel shallow−VAE) central, localized, medium wavelength $Z_1$ filters, respectively, which we used exclusively in Figs. 4, 5, and 6 as well as Supplementary Figs. 6b–d and 7, except for the texture family decoding experiments in Figs. 5a, c and Fig. 6d. For the discriminative CORnet model we started out from the 64 $Z_1$-like units that had their $7 \times 7$-pixel receptive fields in the center of the $224 \times 224$-pixel input images. For these, all illusory-like stimuli fitted into the $224 \times 224$-pixel input image area. However, the receptive fields of some of these 64 $Z_1$-like units were not localized and oriented, preventing the alignment of the illusory stimuli to them. To algorithmically select only localized and oriented filters, we leveraged a key mathematical property of Gabor wavelets: their uncertainty, defined as the product of their variances in the position and wave number domains, reaches the theoretical minimum of $\frac{1}{2}$. Based on this criterion, we selected the 20 units with the lowest-uncertainty receptive fields.

## Illusory contour stimuli

Stimuli used for illusory contour experiments (Figs. 4a–g and 5b, Supplementary Fig. 6b) were based on stimuli used in the experimental paper[48] (see Fig. 4c) that were aligned to the 57 ($50 \times 50$-pixel TDVAE) or 53 ($50 \times 50$-pixel shallow−VAE) central, localized, medium wavelength $Z_1$ filters (see above). Upright and upside-down stimuli were both used, hence the $n = 114$ $Z_1$ units in Fig. 4d. We applied a second, principled filter selection step that left only $Z_1$ units with significantly amplified illusory TDVAE model responses compared to the linear response. The selection criteria were the following: (1) fit into patch: the line segment connecting the two "Pac-Man" centers in the 'Illusory' stimulus aligned to the receptive field (based on the above definitions of center of mass and dominant wave vector), extended by 11.1% in both directions, fits into the patch, (2) overlap between the 'Illusory' stimulus and the generative field (defined as the sum of the squared generative field pixel values on the figure positions of the 'Illusory' stimulus divided by the same sum but masked with the 'Line' stimulus instead) is <0.1, (3) all four response peaks in Fig. 4e are closer than 0.25 pixels to each other (this is a proxy of regular peak shapes). This selection step found $n = 9$ $Z_1$ units for $50 \times 50$-pixel TDVAE, $n = 29$ $Z_1$ units for $50 \times 50$-pixel shallow−VAE, and $n = 0$ $Z_1$ units for $40 \times 40$-pixel TDVAE. In technical terms, generative-field-fitted 'Line', 'Illusory', and 'Rotated' stimulus sets were first generated in black (-1) and white (1) with $16 \times 16$ grid supersampling, then downsampled in a bilinear way, filtered with Gaussian blur, Z scored, and whitened with the PCA basis obtained from natural image whitening above. (Whitening with the PCA basis of natural images instead of the illusory contour PCA basis was chosen because of the small effective dimensionality of the illusory contour stimulus set.) Illusory stimulus parameters used in Figs. 4 and 5b: Pac-Man radius: 9 pixels; Gaussian blur standard deviation: 0.9 pixels; step size of stimulus displacement: 0.125 pixels.

For the discriminative CORnet model we used the following stimulus parameters: Pac-Man radius: 3 pixels; Gaussian blur standard deviation: 1.0 pixels; step size of stimulus displacement: 0.5 pixels. Note that, since CORnet was trained on raw, not-whitened images, we did not apply whitening to the illusory stimuli in this case.

## Model responses to illusory stimuli

One important goal of modeling illusory contour responses in Figs. 4 and 5b was to faithfully compare them with Fig. 3a in the experimental paper[48] (reproduced as Fig. 4c). This Figure in[48] presents mean spike counts in a 50 ms time window, a response averaging procedure we modeled in the following way. Experimentally, auto-correlations of membrane potentials for static stimuli typically decay on a ~20 ms timescale in the V1 cortical area[33]. Therefore, in the context of Figs. 4 and 5b, we modeled 50 ms response averaging in[48] by averaging three independent samples from $Z_1$ model response distributions. Autocorrelation times in higher cortical areas are longer[110–113], which we modeled by taking the three $Z_1$ samples using the very same sample from $Z_2$ model response distributions. For consistency, all model responses to illusory stimuli in Figs. 4 and 5b, as well as Supplementary Fig. 6 were calculated in this way. Ref. 48 does not specify the number of trials each stimulus was presented to the animal. We performed 500 trials for each illusory stimulus in our model experiments which proved high enough that most selected model units showed a significant illusory effect (boosting or suppression). On the other hand, CORnet is a deterministic model, therefore, we could only calculate one model response value for each stimulus–model unit pair. This means that while we were able to calculate differences between model and linear responses, we could not make statements on the significance of their differences.

## Contour completion stimuli

Stimuli used for the contour completion experiment (Fig. 5d, Supplementary Figs. 6c, d) were based on the stimuli used in the experimental paper[50] with the exception that the lattice constant (distance between neighboring bar centers) was reduced by 10% to help the stimuli fit into a $50 \times 50$-pixel patch. As in[50], the central bars of the stimuli were aligned to the 57 central, localized, medium wavelength $Z_1$ filters found in the $50 \times 50$-pixel TDVAE model (see above for the filter selection procedure). Like the illusory contour stimuli, contour completion stimuli were first generated in black (-1) and white (1) with $16 \times 16$ grid supersampling, then downsampled in a bilinear way, filtered with Gaussian blur, Z scored, and whitened with the PCA basis obtained from natural image whitening above. (Whitening with the PCA basis of natural images instead of the contour completion PCA basis was chosen because of the small effective dimensionality of the contour completion stimulus set.) Contour completion stimulus parameters used in Fig. 5d were the following: bar length: 11 pixels; Gaussian blur standard deviation: 0.9 pixels. Supplementary Fig. 6c uses a bar length of 13 pixels instead of 11 pixels, and Supplementary Fig. 6d a Gaussian blur standard deviation of 1 pixel instead of 0.9 pixels. A contrast value (defined as the standard deviation of pixel intensities in the whitened stimuli) of 0.8 was used everywhere.

## Analysis of model responses to contour completion stimuli

The goal of our contour completion experiment was to prove the existence of top-down boosting in the responses of central, localized, medium wavelength $Z_1$ units of the $50 \times 50$-pixel TDVAE model to filter-aligned contour completion stimuli. Therefore, we used an analysis similar to our illusory contour experiment above, as follows. For each $Z_1$ filter, each stimulus type ('random segments' and 'three parallel segments', see Fig. 5d), and each stimulus parameter set (bar length, Gaussian sigma, contrast) we generated 100 stimuli with different random orientations of the freely rotating bars. We selected an experimental time window of 50 ms and used the same one-$Z_2$-sample

mean-of-three-$Z_1$-samples model sampling procedure as with the illusory contour experiment. Using this procedure, we conducted $N = 30$ trials for each stimulus (like in[50]). For each stimulus and $Z_1$ unit, we normalized the trial-averaged three-parallel-segments model response with the trial-averaged random-orientations model response, and subtracted from it the three-parallel-segments linear response normalized with the random-orientations linear response. As the final step, we selected the values of bar length, Gaussian blur standard deviation, and contrast, and for each $Z_1$ unit we plotted the mean of these differences over the 100 random stimulus orientations described above, each $Z_1$ unit giving one dot in Fig. 5d, and Supplementary Fig. 6c, d, respectively.

## Noise and signal correlations

We calculated noise correlation matrices by recording the mean responses of the 40 central, localized, medium wavelength $Z_1$ filters in the $40 \times 40$-pixel TDVAE model to the repeated presentation of the same stimulus and then calculating the Pearson product-moment correlation coefficients between each distinct pair of $Z_1$ units over stimulus presentations. We calculated the signal correlation matrix of a texture family by selecting a number of samples from the family, recording the means of the mean responses of the 40 central, localized, medium wavelength $Z_1$ filters in the $40 \times 40$-pixel TDVAE model to the repeated presentation of each sample, and calculating the Pearson product-moment correlation coefficients between each distinct pair of $Z_1$ units over the selected samples. Following[34], we defined the dissimilarity of two correlation matrices as the mean absolute difference between their elements. To mimic the 400 ms recording window used in[34], we repeated the above experiments with three more model sampling procedures: mean of 20 $Z_1$ samples from one $Z_2$ sample; mean of $2 \times 10$ $Z_1$ samples; and mean of $4 \times 5$ $Z_1$ samples (Supplementary Fig. 7c, d). On a final note, we only used natural images for calculating correlations that had a contrast (standard deviation of pixel intensity values) larger than the observation noise value 0.4 used when training our VAE models.

## Statistics

To test the significance of our results, we used one- and two-sample, one- and two-sided, independent- and paired-sample versions of Student's t-test, depending on the specific individual questions we were asking about the data. We documented the details of the individual statistical tests together with their results throughout the paper.

## Reporting summary

Further information on research design is available in the Nature Portfolio Reporting Summary linked to this article.

# Data availability

All experimental data reported here have been collected and analyzed by others, previously published, and published Figures were reproduced with permission. Learned parameters of the models presented in the paper as well as all test stimuli have been deposited in the Zenodo database and have been made publicly available[115]. Source data are provided with this paper.

# Code availability

Python code of the TDVAE and shallow–VAE models is available as Supplementary Software 1. Python code for all models and all analyses and Figures in the paper has been deposited in the Zenodo database and has been made publicly available together with learned parameters and test stimuli[115]. The study used Python 3.8.5 with the libraries dm-sonnet 1.35, NumPy 1.17.3, scikit-image 0.18.2, scikit-learn 0.24.1, SciPy 1.7.0, statsmodels 0.12.2, TensorFlow 1.15.5 and TensorFlow Probability 0.8.0.

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

## Acknowledgements

The authors thank Mihály Bányai and Dávid G. Nagy for early discussions about the model. G.O. was supported by the European Union project RRF-2.3.1-21-2022-00004 within the framework of the Artificial Intelligence National Laboratory in Hungary and the National Research, Development and Innovation Office under grant agreement ADVANCED 150361. We thank the HUN-REN Cloud for providing compute support for the study (see ref.[114]; https://science-cloud.hu/).

## Author contributions

F.C., B.M., and G.O. conceptualized the study, F.C. performed data curation, F.C. and B.M. wrote simulation software, F.C. performed the simulations, F.C., B.M., and K.Ó. performed the analyses, F.C. and G.O. were designing the visualization, F.C., G.O., and B.M. wrote the manuscript.

## Competing interests

The authors declare no competing interests.
