## [Transparent Peer Review file · Nature Communications]

Top-down perceptual inference shaping the activity of early visual cortex

Corresponding Author: Mr Ferenc Csikor

Version 0:

Reviewer comments:

Reviewer #1

(Remarks to the Author)

The paper attempts to explain certain phenomena in cortical areas V1 and V2 via a "top-down VAE (variational auto encoder)" model. In contrast to the normal VAE that has a fixed prior, the prior here in V1 is modulated by a higher area V2. The authors show that when the model is trained on 'natural images' (dataset not specified), a number of phenomena emerge such as the ability to classify textures and completion of Kaniza figures, which bear resemblance to response properties of V2 neurons.

Overall I don't think we gain much insight about V1 or V2 from this paper. It strikes me as a hacked together VAE with some intriguing emergent properties, but how exactly these arise mechanistically from the top-down flow in the model is entirely unclear. Moreover, there are now numerous models of texture classification and of kaniza figure completion - what does this model achieve that other models do not? Does it perform better? Does it give a better account of V1 and V2? These things are not clear.

I am sympathetic to the authors point of view regarding top-down feedback - it is obviously an essential aspect of inference in visual cortex - but the missing things we would like to know are, what do V2 neurons compute from the outputs of V1? and what signals is V2 sending back to V1, and how exactly do these modulate V1 neurons? In other words, how mechanistically do the feedforward and feedback flows work? The problem with the VAE approach is that it is highly opaque with regard to these issues. The encoder and decoder are composed of multilayer perceptrons, and the paper gives little justification for the number of neurons in each layer and how these might map on to neural substrates in V1 or V2. This also makes it problematic to show what V2 learns, as illustrated in Figure 2b. By contrast, other models such as Hosoya & Hyvarinen (2015), which is not cited, do allow one to visualize how the learned V2 neurons aggregate signals from V1.

The authors should also consider the work of Karklin & Lewicki (2003) who formed a two-layer sparse coding/ICA model. It bears similarities to their approach, as it is also a model where the second layer modulates the prior of the first layer by adjusting its variance.

Finally, I found the paper long-winded and hard to read - extremely long and complicated figure captions, complicated figures, and most of the interesting and important bits of the model are shoe-horned into the methods section at the end, or even into the 'extended data' section. If had to summarize this model to someone else, to describe what the authors did, it would be extremely hard to explain.

(Remarks on code availability)

Reviewer #2

(Remarks to the Author)

This manuscript addresses a long-standing challenge in models of the visual cortex: explaining the computational role of top-down feedback and capturing the resulting modulations of neural activity. The authors set the problem up by contrasting discriminative feedforward models against generative models. The first class of models has been extremely successful in capturing response properties of neurons in the visual cortex, but require supervision for training and generally lack feedback at inference time. Generative models on the other hand are trained without supervision, to build general-purpose representations, but applications to cortical data remain limited to simplistic generative models, and often do not model feedback signals explicitly in multistage systems. The authors propose to use more powerful deep generative models, specifically VAE, that are common in machine learning but have not been used widely to model visual cortex. They formulate variants of a two-layer (deep?) VAE as a model of areas V1 and V2, including one that explicitly incorporates feedback as part of the inference (top-down VAE, TDVAE). With this model, and assuming a neural code based on probabilistic sampling, they show that the V2 layer learns a representation of textures that reproduces properties measured previously in V2, and that the effects of feedback in the V1 layer reproduce two prior empirical observations, i.e. illusory contour responses and modulation of noise correlations (published by this lab in 2019).

This manuscript builds on past work pioneered by Orban among others, relating neural activity to natural image statistics, by probabilistic inference in generative models. The major new contributions here are to 1) use generative models that go beyond the simplistic, hand-designed models used in the past; 2) demonstrate that it gives a unified explanation for complex selectivity in V2 and feedback modulations in V1. This is a significant and long-awaited contribution to the field, because it demonstrates ideas about feedback and hierarchical computation that have been around for a long time, but so far missed a convincing implementation and link to empirical data. The manuscript does a good job of contextualizing the work within the broader literature, with only a few exceptions noted below. Unfortunately however, the implementation and comparison with data appear to have important limitations, leaving some uncertainty about whether they actually support the claims. Additionally, there are some missing comparisons, and parts of the manuscript could be rewritten or restructured to improve clarity. Suggestions for improvement are detailed below.

Major Comments

The authors claim that their goal-general approach has advantages over feed-forward discriminative models, but these benefits are not convincingly demonstrated. A direct comparison with a discriminative model would strengthen the argument. In Methods, a feedforward discriminative model is described, but not clearly mentioned or compared against in the main text. Intuitively, one could expect that a discriminative model trained for texture discrimination, or even one pretrained on generic object recognition, may have intermediate layers that qualitatively match texture representations in V2; but it should not be able to reproduce illusory contour responses and modulations of noise correlation, to convincingly demonstrate the need for generative feedback. A related point is that incorporating top-down interactions within a goal-directed framework is possible, as done in some studies like <https://pubmed.ncbi.nlm.nih.gov/31036945/> ; <https://www.biorxiv.org/content/10.1101/408385v1> ; <https://pubmed.ncbi.nlm.nih.gov/31091234/> . This seems worth discussing.

Some issues follow from limiting the model to small input image patches. One is that selectivity for textures in the top layer might simply reflect that, in 40x40 patches cropped out of natural images, there may be not much more than a single texture and occasionally a single boundary between two objects. I understand the technical challenges of implementing the TDVAE with larger patch sizes, but this seems an important limitation. A second issue with input size is that some V1 filters extend across almost the entire image. This raises issues for interpreting the illusory contour results, because in cases like the example of Fig. 4a the inducer stimulus impinges on the neuron's receptive field. The text addressing this issue (Line 305 and following) seems a bit limited and unclear. The experimental effect is about neurons that are not driven by the inducers alone, whereas the model results are about the modulation on top of the linear response driven by the inducer impinging on the neuron's RF. In addition, a small minority of model neurons is modulated, and half of those are suppressed, not facilitated. So, I find it difficult to interpret the results. Is there any relation to surround suppression phenomena (which also involve feedback components)? Does suppression vs. facilitation depend on RF size or position? Overall I am not convinced that, with the current demonstration, this is a strong model of the illusory contour phenomenon.

A general issue is that while the findings are of potential significance, some effects sizes in the model are quite small (Fig. 2g and 5b,f). This should be explained. The conclusions drawn from the study are also somewhat limited due to the reliance on a specific subset of stimuli (Fig. 1), raising questions about the model's generalizability. A broader comparison involving all 15 texture families mentioned in the neural data could provide a more comprehensive evaluation. The inconsistency in the number of texture samples per family and the number of model neurons in various sections of the results is also a point of concern, with no clear explanation provided for these variations.

References

Line 51: "Importantly, learned invariances mean that high-level task variables do not necessarily carry the information required for other tasks and lower-level variables need to be queried for optimal inferences instead." Work by the DiCarlo lab showed that some low-level information (e.g. positional, viewpoint) is retained and even enhanced <https://pubmed.ncbi.nlm.nih.gov/26900926/>

Line 77: "...textures... critically contribute..." provide references?

Line 80: is ref 19 (Nurminen) about textures? A more relevant one seems Okazawa in area V4 <https://pubmed.ncbi.nlm.nih.gov/25535362/>

Line 402 etc, "with non-hierarchical and essentially linear models" is an ok description of some of the references cited, but not really for probabilistic mixtures of GSMs like ref. 49 and related prior publications: the inference of mixture component is hierarchical (in the sense of a hierarchical generative model, not of a multistage network) and the resulting inference of low-level variables and neural activity are profoundly non-linear.

Line 463 etc: I agree the SSN produces realistic response statistics and has been used to approximate probabilistic inference, but that literature does not include top-down feedback and, at face value, could be interpreted as evidence that top-down feedback is not really needed for probabilistic inference in a (much simpler) generative model. Seems worth discussing more accurately.

Clarity

The introduction focuses much more on the V2 texture selectivity, than on the V1 top-down modulations. I suggest changing this, because the V1 predictions are the most distinctive with respect to feedforward discriminative models. The strength of the normative proposal of this manuscript is that with a single computational goal, it reproduces different phenomena.

The TDVAE uses a complicated implementation, with the V1 feedforward activations being used twice: once to provide driving input to V2, and a second time to provide driving input to V1 while integrating the V2 feedback. I guess this is so there is a single step of FF-FB interaction, rather than a truly recurrent system. It seems important to motivate this more clearly, and discuss the biological plausibility.

The model predictions for neural activity are made under the sampling hypothesis, but this is not stated clearly enough, and a reader not familiar with sampling-based neural code might be confused as to how the responses are actually modeled. (eg. expanding text around line 121). A related point is that the manuscript points to refs 37,38 for how sampling-based representations capture trial-trial variability; that is fine, but those papers use the simpler GSM model, where the modulation of Fano can be understood (approximately) analytically, which is not the case for TDVAE; it would be important to show if stimulus dependent Fano factors resulting from the TDVAE are also consistent with V1 data.

The choice to report membrane potentials, which include negative values, as a measure of model neuron activity could be explained and motivated more. It would be beneficial to also consider the implications of applying non-linear transformations to better represent neuron firing rates and to justify the preference for membrane potentials over firing rates.

Line 192: "TDVAE has no free parameters." Do you mean to fit neural data? Because it does have parameters to fit image statistics obviously.

Line 322 etc: I am confused by the noise manipulation. The original, clean Kanitza stimulus should be maximally uninformative for the target neuron because there is no direct input to it (assuming the neuron is not driven by the inducers, of course). So the response of the neuron in that case should be entirely determined by the contextual prior. And then, adding noise to the entire stimulus still make the inducer noisy, and so should make the contextual prior broader, weaker. These points are worth clarifying.

Fig 4m, you could add horizontal and vertical lines at $x=0$, $y=0$ for clarity.

Fig. 5, the colors red and purple are used to contrast different conditions in panel c,e,f,g

Minor Comments, typos

The paper could offer a clearer explanation of the terms "deep" and "shallow" non-linearity as used in the Fig. 2a.

How results would change in a deeper TDVAE (>2 layers), for instance would the texture representation emerge in the second layer or in the deepest, or somewhere in between?

The clustering of V1 model neurons appears to present discrete values, in contrast to the broader range observed in V2 model neurons (Fig. 2e). Clarification on why V1 projections show discrete values would be informative.

The schematic representation of the model could benefit from refinement for better clarity and ease of understanding (Fig. 1F).

Fig. 5d and related text: is the number of predictors matched, when comparing decoding performance from V1 means, V2 means, and from correlations?

Line 299 "was similar" -> were similar

Line 799 "learned a complete linear basis" is it complete or overcomplete?

(Remarks on code availability)

(Remarks to the Author)

Overview:

This paper leverages recent advances in generative modeling in machine learning (the hierarchical VAE framework) to create a two-level image-computable model of hierarchical inference in the primate brain. This is a logical continuation of earlier work by the authors and by others which proposes that visual processing in the brain can be understood as inference in a task-independent generative model of natural images. Here, top-down or feedback connections from V2 to V1 communicate contextual prior information. These core ideas are not themselves new, but the authors have nonetheless made a significant contribution in terms of (i) technical developments in the model, (ii) performing experiments to probe the behavior of a hierarchical VAE, and (iii) linking that behavior qualitatively to relevant data from primate electrophysiology data. I am a fan of this line of work, but I have a number of concerns detailed below. I look forward to the authors' responses.

Comments on presentation:

- Referring to the model as "V1" and "V2" was very confusing as it leads to statements like "V1 had a similar effect to V1 in neural data." The beginning of the paper introduces model layers as L/M/H. I suggest sticking with this terminology so that "Layer L [in the model] had a similar effect to V1 [in neural data]."
- The term "goal-general" strikes me as inventing a new term for something that already exists in the literature as "task-independent" or "constitutive" representations of uncertainty (see Koblinger et al (2021) or Lange et al (2023))

Very high-level comments on the approach and conclusions:

- The most important thing I'd like to press the authors on is this: this paper uses a powerful computational model (the TDVAE), but analyzes its behavior qualitatively in terms of reproducing high-level patterns/trends in neural data. I have nothing against this qualitative approach -- indeed, it is impressive when a model with no free parameters "fits" neural data even if that fit is qualitative. However, qualitative trends still need some null model or basis for comparison. All of the results in Figs 3, 4, and 5 illustrate qualitative similarities between the TDVAE and actual neural data. These results are difficult to interpret without also seeing a *lack of* qualitative agreement in other models. Other models could include (i) an untrained TDVAE or a TDVAE trained on unnatural stimuli, (ii) the TDVAE with the top-down feedback ablated, (iii) the shallow VAE from Fig 2, etc.
- The framing of the results in terms of "top-down" effects is not entirely supported by the results. As the authors explain (starting on L132), there are many potential ways to instantiate a prior, and we have good reasons to believe that some long-term environment statistics are encoded in the brain implicitly such as in the density and sensitivity of different filters (cf. various works by Wei and Stocker). In the *model* presented here, the primary mechanism for encoding the prior $p(\text{layer L})$ (or $p(\text{V1})$) is using feedback from layer M (or "V2"), and otherwise layer L in the model is kept deliberately simplistic. Effects that appear *in the model* due to including a second "V2" layer could plausibly be implemented in the brain in a purely feedforward fashion. Strong evidence for feedback in the brain comes from inactivation, fast context switching, cross-modal effects, or time delays. (Notably, Lee and Mumford (2003) show time delays in illusory contour responses in V1, which is perhaps their strongest evidence for a feedback or recurrent component there). To summarize this point: effects that appear in the model to require feedback do not necessarily require feedback in the brain, especially given that we know V1 in the brain is more complex than layer L in the TDVAE. A more complex V1->Image generative model would (I conjecture) reproduce some of the effects highlighted here without the addition of the V2 layer.

Technical comments:

- On L152-155, the authors claim that the TDVAE architecture is "as general as possible," but the reality is that all of the results depend strongly on architectural details of the recognition model. This recognition model has quirks such as a deterministic image \rightarrow V1 \rightarrow V2 pathway which is neither very biologically plausible nor is it the only way to do inference in a hierarchical model. I see three feasible ways out of this:
 - (1) the authors could lean into the fact that their results do depend on recognition model details. The challenge is that this avenue reduces the generality of the results and invites further criticism that the recognition model itself appears biologically implausible.
 - (2) the authors could show that their main claims hold for other kinds of inference algorithms such as sampling from $p(V1, V2 | \text{image})$ or doing message passing. This would be much harder but would support the claim of generality.
 - (3) the authors could use importance sampling or self-normalised importance sampling (SNIS) to compute the actual moments (means, variances, and correlations) of the true posterior. This would be most consistent with what is said circa L196 which suggests an interpretation in terms of neural sampling from a true posterior. Doing this would allow the authors to make more precise claims about means, variances, correlations, etc. However, this would transform the results from a study of inference to a study of generative models, thus this option would also call into question any mechanistic "top down" interpretations.
- Addressing the previous bullet would also, I hope, alleviate some confusion with terms like "mean response" which ambiguously could mean an empirical mean of model responses, the posterior mean, or the " μ " part output by the recognition model, which are three related but distinct things.
- On a related note, the authors could use the same importance sampling trick to compute the model evidence directly rather than the ELBO, which would make it easier to compare model quality in Fig 2a. Comparing ELBO values is (unfortunately) standard, but ultimately hard to interpret because the ELBO may be low simply because the recognition model is poor, regardless of the quality of the generative model. Since both the recognition and generative components change across models, direct comparison of the ELBO is not a direct comparison of the quality of the fit to natural images.

- Fig 2b: it would be nice to see second-order receptive fields to show complex cells (i.e. response-weighted covariance in addition to response-weighted average). One cannot conclude much about the selectivity of the model's V2 cells from a uniform gray receptive field.

- L196-198 suggest a neural sampling framework where responses are samples from the posteriors. L198-203 then frames sampling as out of scope of this paper. I have two relatively minor issues with this: (1) I suspect including this response variance as part of the model (perhaps using the importance sampling trick described above) would improve the qualitative fit to some of the data, especially in the analysis of response variance in Fig 3c. Second (2), I am unconvinced that the statement in L201, "without loss of generality ...," is true. How could including a nonlinearity between posterior samples and neural responses have no effect on the results, which analyze neural responses?

- While the results in Fig 3 compare the model to neural data only qualitatively, I suspect that the qualitative fit would be even better if the V1 -> Image portion of the TDVAE generative model was more rich than linear+gaussian. A linear-gaussian V1 layer will develop simple cells but not complex cells, for instance. This likely impacts results in Fig 3e for V1 show very little ability to classify texture families. A moderately more complex, moderately nonlinear V1 -> Image generative model would likely further improve the qualitative fit to neural data here and elsewhere in the paper. This would be a substantial change for a minor improvement which I hope the authors will consider for future work if nothing else.

- Data shown in Fig 4c are not by themselves conclusive that top-down effects are present in the brain. They just show that context matters, and this could "in principle" be implemented in a feedforward or recurrent manner. Similarly, Fig 4b,f,g does not establish "top down" connections as the source of the effect in the model. If the authors want to emphasize "top down" effects, then wouldn't the relevant comparison be between TDVAE and TDVAE with the V2->V1 connection ablated? Or is the "linearized" model equivalent to ablating feedback? (Seems unlikely given the difference between projective and receptive fields).

- Also regarding Fig 4b-c, Lee and Mumford (2003) showed a time delay in the V1 response to illusory contours relative to the response to actual lines, which is a much stronger indication that this effect involves "top-down" effects compared to what is shown in Fig 4c. Could this time-delayed effect be simulated in the TDVAE, e.g. by comparing the image->V1 response to the response once the image -> v1 -> v2 -> v1 pathway executes?

- L322 and Figs 4k-m: while the statement "inference relies on the prior more when the evidence about observations is degraded" is true, it is not obvious that blurring the image constitutes "weak evidence" in the technical sense required here. A blurry image is nonetheless an image, albeit one that is perhaps away from the manifold of natural images. Then again, the Kanisza stimuli are also away from the manifold of natural images. Whether or not each of these constitutes "strong" or "weak" evidence -- in the sense of how it does or does not move the posterior away from the prior -- depends on other model details such as the functional form of $p(\text{image}|V1)$. In a linear Gaussian image model (as used for $p(\text{image}|V1)$ here), the likelihood function $p(V1 | \text{blurry image})$ does not broaden indefinitely as the image gets blurrier. Say, for instance, one of the columns of the $\$A\$$ matrix is constant (DC component of images). Then, the blurrier the image, the more it will shift posterior mass onto that component, which could feasibly *lower* the posterior entropy compared to a sharp natural image. Any image the model is conditioned on is "evidence" even if the image was constructed by a noisy or blurring process. Notably, a GSM would be different, since a dark image is explained by the scale term which removes any constraints on the other latents. The same property does not hold for a linear-gaussian image model and I don't see an obvious way to reliably drive this model towards its prior simply by manipulating the input. I therefore cannot interpret any of the results in Figs 4k-m.

- It is unclear when first reading the noise correlations section why there is any variability in the model's responses to identical stimuli given that it's established earlier that only the means produced by the recognition model are being analyzed, and the recognition model assumes a factorized posterior (zero correlations). It took some digging to uncover the source of variability and correlations in V1 in the model. (I also did not find this detail searching the methods.) I took a few minutes to navigate the code, and from what I understand, "noise correlations" in V1 here refer to the fact that the image -> V2 -> V1 pathway involves the reparameterization trick in V2, so V1 is conditioned on a single random sample from V2, but otherwise the recognition model is deterministic. This seems to conflate properties of this family of recognition models with properties of the actual posterior. If I take what is said in L196-198 seriously, then I should view both neural variability and noise correlations as reflective of the actual variance and correlations in the posterior distribution $p(V1|\text{image})$. This is in principle computable using SNIS as sketched above, but is not what the authors compute here. This again brings up my high-level point above: I think the authors need to commit either to analyzing moments of the actual posterior (e.g. using SNIS) or they need to commit to presenting their results as strongly dependent on the assumed functional form of the recognition model, which itself seems rather biologically implausible.

Miscellaneous minor comments:

- Why does Fig 2 only compare 5 synthetic textures if ref [18] used 15 textures? Would it be easy to add the other 10 texture families for a more direct comparison?

- Missing mean indicators in Fig 3h?

- The code does not run due to hard-coded paths in the environment dependencies. It also has hard-coded dependence on cuda and so couldn't be verified on my laptop.

(Remarks on code availability)

The code is technically available but very hard to read. I glanced at it briefly to clarify a question I had about the methods. I did not review the code thoroughly. I was unable to set up a local environment to test things because the requirements.txt file appears to include dependencies specific to the authors' systems. I did not troubleshoot it further.

Reviewer #4

(Remarks to the Author)

Overall, we found this paper to be well written, thorough, and interesting. It provides a goal-general learning framework by

inferring features of images at multiple levels in a hierarchy. The authors suggest an interesting Top-Down VAE (TDVAE) model for their hierarchical inference. The key aspect of their model is the use of top-down feedback from V2 to V1, such that neural activity in V1 depends not only on the stimulus but also on activity in V2. We support publication after addressing some relatively minor concerns.

Some of their methods need clarification. Their architecture figure, while connecting nicely to biology, is hard to follow computationally, and appears to hide a bunch of important computational structure. We attach a sketch of a proposed architecture diagram that makes things a lot clearer — though at the expense of revealing more clearly how the “top-down feedback” is in fact just a longer feedforward pathway.

Additionally, it’s unclear to me from their main text, methods, and supplement how the generative and recognition architectures are related, except for $p(\text{image}|V1)$. Is $p(V1|V2)$ part of the generative model? What is the recognition model? These components should be explained better.

This hierarchical structure is nice and relatively tractable. It seems that another way of interpreting their architecture is as a feedforward network with a long skip connection from V1-ff to V1-INT. But this hides an important problem: recurrence. The authors should spend some time considering the consequences of assuming essentially a one-pass loop. I don’t know that this is feasible through analysis, but it would be important to at least think about and discuss.

The authors only have one layer of top-down feedback. This seems reasonable at first glance, given that they only cover V1 and V2. But each of these brain areas are themselves multi-layer structures, with their own feedback loops. What are the consequences of this? These effects should be presumably at least as strong as feedback they address. The authors should address this issue.

We especially appreciated the distinction they made between different factorizations of a joint distribution over images and features that were identical when exact, but different when approximated. Since exact solutions are infeasible, this factorization provides potentially useful and biologically plausible structure. We think this nice theoretical aspect of their argument could be emphasized more prominently.

It’s too strong to say that “correlations in the posterior can be identified with noise correlations” (L198). There are other reasons one could obtain correlations in the posterior, related to other forms of information loss that are not manifested as stochastic stimulus-conditioned variability.

The authors “test the consequences of the proposal that noise correlations are shaped by contextual priors” (L368). What is special about contextual priors here? Their test consists of phase scrambling, but this is not a compelling test of top-down inference. Instead, it may be just a generic property of more feedforward nonlinearities. The authors observe that high variance in V2 response across textures helps discriminate between them, and Figure 5 identifies what happens as a consequence of top-down effects. But when combining feedforward nonlinearities with nuisance variables (e.g. different samples of a given texture), we expect to see the same effects (e.g. Yang, Cotton, Walker, Tolias, Pitkow 2021). What is it that makes the authors conclude that top-down feedback plays a special role in these properties, beyond generic properties of more feedforward computation?

Very minor:

Eq 4 — TDVAE should be superscript on q_{Φ} , not an exponent (or superscript) on its parentheses.

Their phase and filter scrambling could be related to the Scattering transform (Mallat 2012) and metamers (Freeman and Simoncelli 2011).

(Remarks on code availability)

Version 1:

Reviewer comments:

Reviewer #1

(Remarks to the Author)

The authors have provided a good and well-reasoned rebuttal. I wish to retract my previous comment about the model being a “hacked together VAE” - that is too strong. Indeed the authors have a clear theoretical motivation for their approach, and it is an interesting and novel contribution to the machine learning literature. The neurobiological motivations are also sound and clearly stated, and a model that can account for the role of feedback connections in the cortex is sorely needed.

However my reservations about this paper's contribution to neuroscience remain. As a neuroscience model one would hope that it could provide insight as to why neurons respond the way they do. Although the model replicates certain phenomena in visual cortex, the reason for these properties emerging is unclear, other than that they are the properties of a hierarchical VAE trained on natural images. So we are left with an opaque system that happens to mirror the behavior of another opaque system, with only superficial similarities in their architecture.

The model architecture as diagrammed in Figure 1f seems a far cry from the architecture of visual cortex. Given that orientation-selective RF's are found already in layer 4 of visual cortex, how are we to understand the many layers of processing - Z1-ff followed by Z1-INT - between the image and Z1? Why are all these layers needed and what are they doing? Moreover it appears that Z1 (putative V1) does not serve as input to Z2 (putative V2). Rather Z2 gets its input from a separate stream going through Z1-ff followed by Z2-ff. Z2 then conditions Z1 through another multilayered set of connections, Z1-TD followed by Z1-INT. But topologically there is nothing hierarchical about this - Z1 and Z2 are both driven by the image through separate networks, so I would put them essentially at the same level. That they share Z1-ff in common, with some additional processing for Z2, does not put Z2 above Z1. Z2 conditions Z1, but there is nothing "top-down" about this, it is really more of a "sideways" modulation through the mixing of Z2's contribution into the Z1-INT layers feeding into Z1.

Nevertheless, it is certainly true that the model exhibits interesting response properties, and one can see certain parallels to visual cortical response properties. At the end of the day however, it is unclear what to make of all this with respect to our understanding visual cortex.

(Remarks on code availability)

Reviewer #2

(Remarks to the Author)

The authors have addressed all my requests for clarification and for additional analysis. The revised manuscript, which includes many new simulations, controls, comparisons between models, and replication of additional V1 data, has convinced me of the soundness of the conclusions (fully addressing my main concern). After reading also the other reviewers comments and the authors responses, I am convinced that the manuscript provides a very important conceptual contribution to the field of probabilistic neural representations, and the model could be the basis for interesting follow-up work.

At this point I have only a few very minor suggestions that would not require a re-review from me.

- Line 97 is the first mention of 'correlations' and is not explained or contextualized at all in the Introduction. Because stimulus dependent correlations are a main part of the argument of the paper, it seems important to introduce them. The Introduction is already lengthy though, so an alternative would be to not mention correlations in the Introduction at all.

- Line 178 'Such choices... parameter sharing.' Could add a reference

- Line 182 mentions the 'computational graph' but then the pointer to Extended figure is given in the next sentence which makes it sound like the figure is about something else.

- Line 218 mentions image size (20-pixel). The different image sizes are important, as addressed in the rebuttal and in the Methods. But at this point of the text it is unclear why this number matters and even what it refers to (unless one has read Methods first, which not every reader will not do).

- Membrane potentials and firing rates, rectifying nonlinearity. I appreciate the authors motivation not to add a nonlinearity to their simulated activities, this is fine. But part of the justification given is that past work showed sampling explains both membrane potentials and spike counts. That work compared each version of the model to the corresponding data. Instead, simulated membrane potentials here are compared to empirical V1 spike count mean and noise correlations. I would try to tighten up the justification/discussion.

- Line 473 'the variational posterior is correlation-free'. This is an important point, but just mentioned casually and with jargon very deep in the paper. I suggest expanding and explaining in simpler terms. Also, I am not sure that even in Methods you explained that the Z1 variational posterior is factorized across Z1 neurons.

(Remarks on code availability)

Reviewer #3

(Remarks to the Author)

I reviewed this manuscript previously, and I thank the authors for making fairly substantial changes in response to my

comments and the comments of the other reviewers, as well as for providing in-depth replies to all concerns raised (even the rather minor ones!)

After reviewing the changes and the comments by the other reviewers, my original feeling remains that the model and analyses are interesting and thorough, but that claims about the modeling results' relevance to the brain are overstated. It's clear that feedback and recurrence in the brain make for richer information-processing. It's also clear that Z1 in the TDVAE model becomes richer by adding a Z2 → Z1 connection pathway. However, echoing my earlier concerns and the concerns of the other reviewers, I still do not think that any of the effects of Z2 on Z1 in the paper can be unambiguously mapped onto "top down" or "feedback" connections from V2 to V1 in the brain any more than they can be mapped onto local recurrence or even a more complex feedforward pathway in the brain. The authors' defense of the "top-down" terminology is essentially that (i) V2→V1 influences in the brain are fairly well documented, and thought to be involved in effects like time-delayed V1 responses to illusory contours; and (ii) the $q(Z2|im)q(Z1|Z2,im)$ factorization of the approximate posterior is one of many valid solutions for inference and implies some mechanistic information-flow from Z2 back to Z1.

Regarding (i), this would be a more compelling argument if Z1 in the model were closer to V1 in the brain. But as it stands, everywhere that the model is compared against V1 and V2 experimental data, it is clear that V1 in the brain is far more complex than Z1 in the model. Fig 2e vs 2f shows much more structure in V1 than Z1. Texture decoding performance from V1/V2 is not reported in Fig 2d, but could be another useful comparison (and I would wager that V1 in the brain performs better than Z1 in the model). Figs 3d/3f/3h show much less drastic of a difference between V1 and V2 than Figs 3c/3e/3g show for model Z1 and Z2. Perhaps the exchange of information between Z1 and Z2 in the model is a good model for the relationship between simple cells (Z1) and other more complex cell types (Z2) all within V1, and perhaps the Z2→Z1 influence in the model nicely captures the effects of local recurrence in the brain.

While I do think that V2→V1 feedback in the brain is likely the primary source for some of the effects claimed in this paper (like illusory contour responses), I don't think that we actually learn that from the TDVAE model. New controls in Fig 4 are useful, and I thank the authors for going through the efforts to include them. Unfortunately, I think they end up undermining the story: the fact that the ShallowVAE and the TDVAE with Z2 clamped still show an enhanced response to illusory contours calls into question whether "top down" effects are really the key to the story, even in the model. While inference of $q(Z2|im)$ produces *more* of an effect, it's not clear that more is better here; after all, all model fits here are qualitative and the experimental data in Fig 4c also suggest to me that the illusory contour response in the brain is not all that vigorous either.

Regarding point (ii) above, I do agree with the authors that the $q(Z2|im)q(Z1|Z2,im)$ factorization is nice in the sense that it is computationally convenient. In fact, this is essentially one step away from rederiving message-passing algorithms, since both $q(Z1|im) = \int q(Z2|im)q(Z1|Z2,im) dZ2$ and $q(Z2|im) = \int q(Z1|im)q(Z2|Z1) dZ1$, implying, mechanistically, recurrent information flow between Z1 and Z2 and that the true posterior is a fixed point of these bi-directional dynamics. In this work, the authors have borrowed a useful trick from machine learning (cf. LVAE, VLAE, NVAE architectures) where amortized inference is applied to hierarchical models using an initial deterministic feedforward pass from inputs to both Z1 and Z2, which is then combined with a reparameterized sample of Z2 fed back to Z1, inducing some stochasticity. However, the ELBO (or $KL(q||p)$ if you prefer) is technically defined in terms of an expectation over q . At training time, it is sufficient to estimate this expectation using a single reparameterized sample from Z2, since this gives unbiased (but high variance) estimates of the ELBO gradient. This is sufficient for learning as long as the learning rate is small, averaging away that stochasticity and effectively maximizing the ELBO. But using the reparameterization trick as a mechanistic source of stochasticity at inference time abuses the definition of the ELBO and confuses a trick used for training (single reparameterized samples drawn from q) with a trick for inference (recurrent message passing between Z2 and Z1). True variational inference converges to a single best-estimate for $q(Z1)$ and $q(Z2)$ for each input and is "deterministic" in the sense that the same "best" $q(Z2)q(Z1)$ approximation will be reached for the same input every time (local optima notwithstanding). The new text L224–L233 is misleading in this regard. A deterministic algorithm for inference is not necessarily a MAP algorithm; variational inference taken to convergence is also nearly deterministic in the sense that it gives the same q for the same input every time. Likewise, a stochastic inference algorithm like sampling is not "probabilistic" in virtue of its stochasticity, it is probabilistic in virtue of the fixed point of its dynamics being a representation of the posterior.

All that being said, the TDVAE is an interesting model and it's interesting that applying the reparameterization trick in Z2 at test-time reproduces some properties of V1 in the brain that the authors report. I would support the authors in publishing this work as an interesting case-study in recurrent (but not necessarily "top-down") dynamics of a particular (novel?) inference scheme where reparameterized samples are used at test-time, making for a sort of hybrid between variational and monte carlo methods. However, the claims of the normativity and the generality of the model remain overstated, as do the claims that Z2→Z1 influences in the model should be interpreted as "top-down" effects.

A few more minor nits:

- L178: should include references for the claim that the architectural motif is "widely used in ML". (Ladder VAE and related methods, perhaps?)
- L218, Fig 2a, and extended data table 4: was only one model of each type trained? The ELBO (like EM) is known to be prone to getting stuck in local maxima. Unclear whether ELBO differences like -400 vs -407 are meaningful. If not already doing so, better practice would be to train multiple instances of each model and take the best-performing model in each class.
- Fig 2b: I appreciate the addition of STC for Z2, but at the given resolution it looks like noise. The frequency domain visualization helps a bit, but it is still rather hard to make sense of.

(Remarks on code availability)

Reviewer #4

(Remarks to the Author)

Overall, I still find this to be an interesting study, and I support its publication with a bit of clarifications in discussion. I have two main comments, and a bunch of minor ones. I don't feel I need to see a further revision.

First: I think it is important to clarify the distinctions between top-down and recurrent. Conventionally top-down connections are viewed as feedback, and thus are recurrent. In contrast, their model is purely feedforward, because they are only unrolling one loop of recurrence. This is helpful to many readers to say up front.

Second: I'm glad the authors incorporated a computational diagram to explain their architecture. Done well, this will be very helpful. I think the Extended Data Figure 1 is more important than a supplementary figure, and could go well with Figure 1f. That said, I find the current version quite confusing. The superposition of models are very hard to parse. For instance, there are two streams involving Z2: the FF pathway and the generative pathway. Are they both used?

I also should mention I have several disagreements with Reviewer 1. The architecture is reasonably principled. And VAEs should not be judged on giving mechanistic insights. There are many many ways of achieving the same representations and computations. For some purposes we really care about the nuances here. But for computational insight about the kinds of things that are represented, much progress has been made without a one-to-one correspondence between artificial neurons and biological neurons.

MINOR:

Fig 1f: aesthetic suggestion: black arrows confusingly bend unnecessarily. Reorganize feature maps so flow is more easily seen.

L165: some conflict here between "TDVAE was left as general as possible" and "Z1 is linearly related to image"

L175: "sharing the entry layers of the two networks... is referred to as parameter sharing." This doesn't sound right, it's instead an architectural choice. Parameter sharing would instead be duplicated layers at distinct sites whose parameters are tied. You could say instead that this is "equivalent" to parameter sharing in a different architecture with two distinct branches, but I'm not sure what you gain from saying that.

L220, comma should be colon

Fig 2a caption: Shallow-VAE: non-hierarchical VAE constrained to Z1. Isn't this just linear, since Z1 is linear in pixels?

L272: A 0.2 improvement is considered LOW for Z1, but a smaller 0.1 improvement by their preferred model is considered outperforming? Seems like changing judgments.

L272: Text needs labels to reference bars in Fig 2d so the reader can more easily compare claims to results.

L276 "endow the Z2 to build" — misused grammar. Should be "endow Z2 *with* capacity to build..." or similar

L278: Mean over what? Neurons encoding means of Variational Approximation? See Reviewer's previous comment on being careful about the use of the term "mean", which can mean many things here.

L297: Here .08 is "markedly lower" while 0.28 is only "slightly higher"?

L968: ELBO = *Evidence* Lower Bound, not *Expectation* Lower Bound.

Many places report absurdly small p values. We cannot have such confidence in the model, so just report something like $p \ll 10^{-3}$.

(Remarks on code availability)

Version 2:

Reviewer comments:

Reviewer #1

(Remarks to the Author)

The paper has undergone another revision, mostly to the narrative. The fundamental structure of the model remains unchanged.

My main concern, as stated previously, is that the topology of this model does not match that of the visual cortical areas it seeks to shed light on. To be clear, the signal flow in this model is

image  Z1

image  Z2  Z1

whereas in visual cortex we have

image  V1  V2  V1  V2 . . .

In other words, in visual cortex we see a recurrent loop: V1 feeds to V2; V2 feeds back to V1, presumably changing its state; the modified V1 feeds to V2 again, and so forth. It is a loop that repeats. That is what makes it mysterious, and difficult to model.

The model in this paper has no such loop, it is more of a sideways modulation in one step with no recurrence. This is even emphasized by the authors now in the discussion:

"This single-step top-down pass avoids highly recurrent refinement of the responses between V1 and V2."

which is then justified by saying

"Experiments relying on time delays between neural responses in V1 and V2 are compatible with our account but more precise experimental data can identify more elaborate recurrent computations."

It is not clear which experiments are being referred to. According to Bullier the signal delay between V1 and V2 is 2-4 ms due to traveling via myelinated fibers over a short distance. So there is ample time for signals to flow back and forth.

Again, I wish to emphasize that the model has many interesting properties and attributes in its own right - it is an impressive piece of work, very extensive analyses and interesting results - so there is no question in my mind that this work should ultimately be published. It could even be seen as loosely capturing "higher-level" modulation of V1 since the stream to Z2 goes through more processing than the stream to Z1. But the fundamental gap between the model and visual cortex, as pointed out above, should be stated forthrightly in the introduction, and perhaps the title modified accordingly to qualify what the model is actually showing.

(Remarks on code availability)

Reviewer #3

(Remarks to the Author)

This is my third review of this manuscript after some back and forth with the authors. In my last review, I raised two primary lingering concerns:

1. My first and biggest concern had been whether or not it was appropriate to analogize Z1/V1 and Z2/V2. I had expressed concerns about simplicity and complexity of the model and brain respectively. The authors' most recent reply was very thorough on this point, and they have convinced me that the Z2->Z1 pathway is, in fact, comparable to V2->V1 feedback effects in the brain, at least enough to be of genuine interest and use to the neuroscience community.

2. My second concern from the prior round of reviews had to do with the nature of variational inference and neural variability. This is a minor technical point and in no way calls into question the validity of the model or the results. The way that the TDVAE model explains correlated neural variability -- indeed the way it explains any neural variability -- is that samples of Z2 are drawn from the (fixed) $q(Z2|input)$ distribution, and $q(Z1|input,Z2)$ is conditioned on the sampled value from Z2. If the model were doing "pure sampling", it would draw samples for both Z1 and Z2 and there would be a lot more total variability. If the model were doing "pure variational inference", it would seek a steady-state representation of Z2 by fully marginalizing over Z1, and there would be essentially no variability remaining. My main point before was that what the TDVAE does is somewhere in between these. This is interesting, possibly even novel from an algorithmic perspective! But, I concede that it is not really wrong to call the TDVAE algorithm "variational inference" even if it includes a sampling component. I don't quite rescind this point, but I will demote it to the category of minor nit-picking.

I am happy to consider all issues resolved.

(Remarks on code availability)

Response to Reviewers

Please note that all the line numbers mentioned below refer to the updated manuscript and that response is in **bold**.

Reviewer #1

The paper attempts to explain certain phenomena in cortical areas V1 and V2 via a "top-down VAE (variational auto encoder)" model. In contrast to the normal VAE that has a fixed prior, the prior here in V1 is modulated by a higher area V2. The authors show that when the model is trained on 'natural images' (dataset not specified),

The data set is specified and referenced in the Methods, under the section 'Natural image datasets'.

a number of phenomena emerge such as the ability to classify textures and completion of Kaniza figures, which bear resemblance to response properties of V2 neurons.

Overall I don't think we gain much insight about V1 or V2 from this paper.

End-to-end trained models provide an opportunity to investigate the properties of sensory cortices under the assumption that these are shaped by natural image statistics. Further, contrasting different end-to-end trained architectures makes it possible to characterize the computational principles underlying processing sensory stimuli. End-to-end trained deep learning models have achieved remarkable success in predicting visual cortical neural responses to natural and synthesized images. These models were exclusively trained to perform discrimination on naturalistic images. Although the computational principle for training these discriminative models – supervised learning – is not a relevant paradigm in biological systems, non-supervised deep learning models have been less explored in this context. In particular, VAEs have achieved great successes in machine vision. Despite their widespread use in computer vision and despite their theoretical appeal, we are not aware of other studies that provided such detailed study of applicability of VAEs to the hierarchy of the visual cortex. We introduced a stringent hierarchical version of VAEs that we explored under end-to-end training. We chose to rely on a very conservative and highly principled training and showed that without any parameter tuning – beyond what natural image statistics dictates – a range of phenomena characteristic of the early visual system is born for free. In summary, the study offers an integration of two lucrative lines of research of the visual cortex: generative models and deep neural networks. We believe that the deep generative modeling framework presented here is a valuable contribution that will be of high interest for the wider community. To make the comparison with deep

discriminative models more explicit, we have added a standard feed-forward model to contrast with our proposed model (pgs 8-9, lines 310-318).

We agree with the comment of the reviewer that there is a trade-off between end-to-end training and interpretability. Please see our arguments about these issues later. We have extended the discussion of the paper to reflect this.

Further, when considering the contributions of this paper, beyond the considerations presented above, as the reviewer later pointed out, evaluation of a V2-dependent prior in V1 is an exciting perspective to interpret top-down processing in the visual cortex. The proposed VAE formalism provides a potent framework for these investigations. Further, we would like to highlight that the probabilistic nature of our model is crucial for interpreting noise correlations in a normative setting, which we explore in detail in the manuscript.

Overall, we employed a conceptually simple unsupervised learning approach using a hierarchical variational autoencoder (VAE). After end-to-end training, the model's learned representations were found to be aligned with a broad spectrum of experimental findings, including phenomena such as illusory contours and the structure of noise correlations. To better convey these messages, we have rewritten the introduction of the paper, as well as we have updated the first paragraph of the discussion (pg 17 line 526).

It strikes me as a hacked together VAE with some intriguing emergent properties, but how exactly these arise mechanistically from the top-down flow in the model is entirely unclear.

We respectfully disagree with the reviewer's characterization of our computational approach as 'hacked together', instead we argue that the construction of the model is normative. Our model is grounded in a principled framework, where the loss function is based on simple probability theory. The use of amortized inference and the Variational Autoencoder (VAE) framework are both well-established methods in the field of unsupervised learning. In fact, we believe our approach is less "engineered" than many unsupervised deep learning models in the machine learning literature, which often involve assembling disparate modules and introducing custom loss functions.

While we incorporate a few inductive biases, some of which have previously been applied in linear generative models of V1, the system is largely allowed to minimize its general loss function autonomously. Even within the VAE domain, we have made a conscious effort to avoid some common industry-standard practices (such as beta-VAE and gradient clipping during training), which we believe are less principled.

We address these points in the Results (pg 5 line 162-165, 171-185).

We agree with the reviewer on a limitation of deep learning models, ie. that interpretability of the models is limited by the choice of using a flexible function class to implement the (nonlinear) computations. This approach presents us with a trade-off: we can investigate the computations necessary to process natural images without specific

commitment to a particular choice, but leaves the mechanistic interpretation of the learned model open. We agree with the reviewer that mechanistic insights are ultimately important, and should be the subject of research. We also believe that the open and data (image statistics) driven approach is important as it shows what are the properties of the nervous system that are consequences of the natural pressures coming from the environment. (Mechanistic insights provided by deep discriminative models (diCarlo et al) are also limited, still their contribution is not disputed). As noted above, the discussion has been updated with this extension (pgs 18-19 lines 590-601).

Moreover, there are now numerous models of texture classification and of kaniza figure completion - what does this model achieve that other models do not? Does it perform better? Does it give a better account of V1 and V2? These things are not clear.

We thank the reviewer for highlighting this point. It is indeed the case that earlier studies have discussed illusory contour responses (Dura Bernal et al (2012); George and Hawkins (2009); Kogo et al (2009); Lotter et al (2020); Pang et al (2021)), some of them actually relying on Bayesian computations. We now clarify our contribution more clearly in the discussion (pg 20 lines 674-685).

I am sympathetic to the authors point of view regarding top-down feedback - it is obviously an essential aspect of inference in visual cortex - but the missing things we would like to know are, what do V2 neurons compute from the outputs of V1? and what signals is V2 sending back to V1, and how exactly do these modulate V1 neurons? In other words, how mechanistically do the feedforward and feedback flows work?

We thank the reviewer for highlighting this point. While the mechanistic insights are not available in this end-to-end trained model, the hierarchical deep generative model permits the characterization of the properties of top-down feedback. We argue that the contextual prior is a particularly instructive view on how V2 shapes V1 activity: the larger spatial scale and more complex features of V2 can inform V1 about high-level content of the visual environment. The consequences of this form of feedback are twofold. First, considering texture-like features, inference about these implies specific forms of contextual effects in V1. This has been explored through the correlations present in model V1 responses. We now expand on this by investigating how the mean of the model V1 activity is shaped by feedback: by performing decoding on an ablated model we show that feedback contributes to enhanced decoding of texture information in V1 (Fig. 5a). Experimental insights about texture statistics-related information in V1 is obtained from the investigations of Ziemba (2019) who, based on temporal evolution of the differences between texture responses and phase scrambled version of texture responses, argued that this information reaches V1 through feedback. We extended our analysis to investigate this effect and showed that the top-down feedback could explain this difference (Fig. 5c). Second, other high-level features represented in V2 and V4, not investigated in our modeling study, will imply other forms of contextual priors that can be directly investigated.

These computational considerations are not providing direct mechanistic insights about how V2 features are combined and are the subject to further investigations. We have now updated the discussion to reflect on these issues (pgs 18-19 lines 590-601).

The problem with the VAE approach is that it is highly opaque with regard to these issues.

Please see our arguments about the interpretability of deep learning models above.

The encoder and decoder are composed of multilayer perceptrons, and the paper gives little justification for the number of neurons in each layer and how these might map on to neural substrates in V1 or V2.

We acknowledge that, similar to other end-to-end deep learning models, we did not attempt to map each individual MLP layer to a specific neuronal layer. Instead, we treated the layers as general computational blocks. As a consequence, anatomical justification was not sought for network sizes. The size of these networks were solely justified by ensuring the best performance on a test set of natural images. We chose network sizes where the ELBO was highest but we found limited sensitivity to the network sizes. Instead of network size, we investigated the effect of architectural choices in detail. We reported experiments with changing the complexity of the integration component of $q(Z_1 | x, Z_2)$ (based on reviewer suggestion, V1 and V2 of the model were renamed); the complexity of $p(Z_1 | Z_2)$. A new experiment was performed to test the effect of not sharing initial stages of $q(Z_2 | x)$ and $q(Z_1 | x, Z_2)$ (pg 7, line 212-220)

In the Supplementary material, in Extended Data Table 3 readers can find the hyperparameters of alternative models. Furthermore, Extended Data Tables 4 and 5 summarize relevant properties (ELBOs, Z1 and Z2 representation quantities) of these models. Namely, we found the most relevant to test linear and non-linear fusion models (Z1-INT) which combine bottom-up and top-down signals.

In summary, the computational graph of the recognition model builds on anatomical data (which we expand on in the results (pg 5, lines 171-185)) but the computational blocks were considered to be general components without explicit consideration of the architecture. We highlight this point in the Discussion (pg 19, lines 620-632).

This also makes it problematic to show what V2 learns, as illustrated in Figure 2b. By contrast, other models such as Hosoya & Hyvarinen (2015), which is not cited, do allow one to visualize how the learned V2 neurons aggregate signals from V1.

We thank the reviewer for directing us to this relevant paper. A shared focus of that paper and our study is the hierarchical computations of the visual cortex. That paper indeed offers a systematic exploration of the types of statistics the network needs to calculate in order to obtain several important characteristics of neural representation in V2. Notably, texture sensitivity was not among these. Our choice to employ end-to-end training makes

it possible to demonstrate the sufficiency of natural image statistics to shape visual cortical representations. The end-to-end training is in contrast with the gradual introduction of additional computational layers offered in that paper, in which training of subsequent computational layers occurs with learned parameters of the previous layer.

A more important distinction is the normative approach taken in our study: based solely on the principles of probability theory and through the insights of state-of-the-art machine learning we propose a computational architecture that is capable of addressing top-down influences in the visual cortical hierarchy. While hierarchical inference has been proposed before as a normative motivation for top-down influences (Lee, 2003), and inspired experiments (Lee & Nguyen, 2001), it were the breakthroughs in machine learning that permitted training such a hierarchical generative model on natural images.

We have now rewritten the introduction to better emphasize the contribution of the normative approach to characterize the top-down computations and we now also elaborate in the discussion on the contributions of the approach taken by the Hosoya & Hyvarinen (2015) and Schwartz & Simoncelli (2001) papers (pgs 18-19 lines 590-601)

The authors should also consider the work of Karklin & Lewicki (2003) who formed a two-layer sparse coding/ICA model. It bears similarities to their approach, as it is also a model where the second layer modulates the prior of the first layer by adjusting its variance.

We thank the reviewer for highlighting this paper, which presents a two-layer model of natural images considering computational perspectives (and not biological ones). We now integrate its contributions with Karklin & Lewicki (2009) by the same authors (pg. 18, lines 551-553). One key distinction between the suggested paper and our work is that in the former, the higher-level layer modulates only the variance, not the correlation, of the lower-level features.

Finally, I found the paper long-winded and hard to read - extremely long and complicated figure captions, complicated figures,

We thank the reviewer for drawing our attention to this. We have now rewritten the introduction to enhance its focus, we have streamlined the narrative of the paper by removing the experiment related to panels Fig. 4k-m and adding experiments that better integrate with the main messages of the paper (Fig. 5). We strived to make the captions more compact and to avoid overlaps with the main text.

and most of the interesting and important bits of the model are shoe-horned into the methods section at the end, or even into the 'extended data' section.

We strived to include everything that is important for the general readership of Nature Communications available in the main text of the paper but we are sorry if any particular element of interest was relegated to the Extended Data section. We are not entirely sure

about the relevant components, but here we highlight several edits that have an extended explanation in the main text. We have extended the theoretical background provided directly in the Results (pgs 3-5, lines 132-151), including a clearer description on the distinction between alternative partitionings of the joint variational posterior.

If had to summarize this model to someone else, to describe what the authors did, it would be extremely hard to explain.

In addition to rewriting the introduction we have also updated the first paragraph of the discussion (pg 17 line 526). We also believe that the streamlining of the narrative together with the new experiments shown on Figure 5 make the message on how hierarchical inference shapes top-down interactions more eloquent.

Reviewer #2

This manuscript addresses a long-standing challenge in models of the visual cortex: explaining the computational role of top-down feedback and capturing the resulting modulations of neural activity. The authors set the problem up by contrasting discriminative feedforward models against generative models. The first class of models has been extremely successful in capturing response properties of neurons in the visual cortex, but require supervision for training and generally lack feedback at inference time. Generative models on the other hand are trained without supervision, to build general-purpose representations, but applications to cortical data remain limited to simplistic generative models, and often do not model feedback signals explicitly in multistage systems. The authors propose to use more powerful deep generative models, specifically VAE, that are common in machine learning but have not been used widely to model visual cortex. They formulate variants of a two-layer (deep?) VAE as a model of areas V1 and V2, including one that explicitly incorporates feedback as part of the inference (top-down VAE, TDVAE). With this model, and assuming a neural code based on probabilistic sampling, they show that the V2 layer learns a representation of textures that reproduces properties measured previously in V2, and that the effects of feedback in the V1 layer reproduce two prior empirical observations, i.e. illusory contour responses and modulation of noise correlations (published by this lab in 2019).

We thank the reviewer for their precise summary

This manuscript builds on past work pioneered by Orban among others, relating neural activity to natural image statistics, by probabilistic inference in generative models. The major new contributions here are to 1) use generative models that go beyond the simplistic, hand-designed models used in the past; 2) demonstrate that it gives a unified explanation for complex selectivity in V2 and feedback modulations in V1. This is a significant and long-awaited contribution to the field, because it demonstrates ideas about feedback and hierarchical computation that have been around for a long time, but so far missed a convincing implementation and link to empirical data. The manuscript does a good job of contextualizing the work within the broader literature, with only a few exceptions noted below.

We thank the reviewer for the positive words

Unfortunately however, the implementation and comparison with data appear to have important limitations, leaving some uncertainty about whether they actually support the claims. Additionally, there are some missing comparisons, and parts of the manuscript could be rewritten or restructured to improve clarity. Suggestions for improvement are detailed below.

We thank the reviewer for raising these points and we answer them in detail below.

Major Comments

The authors claim that their goal-general approach has advantages over feed-forward discriminative models, but these benefits are not convincingly demonstrated. A direct comparison with a discriminative model would strengthen the argument. In Methods, a feedforward discriminative model is described, but not clearly mentioned or compared against in the main text. Intuitively, one could expect that a discriminative model trained for texture discrimination, or even one pretrained on generic object recognition, may have intermediate layers that qualitatively match texture representations in V2; but it should not be able to reproduce illusory contour responses and modulations of noise correlation, to convincingly demonstrate the need for generative feedback. A related point is that incorporating top-down interactions within a goal-directed framework is possible, as done in some studies like <https://pubmed.ncbi.nlm.nih.gov/31036945/> ; <https://www.biorxiv.org/content/10.1101/408385v1> ; <https://pubmed.ncbi.nlm.nih.gov/31091234/> . This seems worth discussing.

Thank you for raising this issue. We have now added an analysis of a feed-forward discriminative model (CORnet) (Kubilius 2018) as well as an untrained hierarchical VAE as baselines. We conducted experiments analogous to those conducted with the TDVAE model to contrast the representations learned by the two models. We have integrated these models with the investigation of the representation learned through training on natural images (Figure 2). We have also extended the analysis of illusory contours with the feed forward model and included the results (Fig. 4g). The Results section has been extended accordingly (pg 8-9, lines 291-304, 310-318, pg 13 lines 416-420). In the Methods section, we have added a new subsection titled "Discriminative model benchmark," which provides general information about the discriminative model. In addition, we have updated all relevant sections with specific details pertaining to CORnet.

We thank the reviewer for pointing out an opportunity to contrast our model with recurrent alternatives. We now include a discussion of these recurrent models in the Discussion (pg 19-20, lines 633-652).

Some issues follow from limiting the model to small input image patches. One is that selectivity for textures in the top layer might simply reflect that, in 40x40 patches cropped out of natural images, there may be not much more than a single texture and occasionally a single boundary

between two objects. I understand the technical challenges of implementing the TDVAE with larger patch sizes, but this seems an important limitation. A second issue with input size is that some V1 filters extend across almost the entire image. This raises issues for interpreting the illusory contour results, because in cases like the example of Fig. 4a the inducer stimulus impinges on the neuron's receptive field. The text addressing this issue (Line 305 and following) seems a bit limited and unclear. The experimental effect is about neurons that are not driven by the inducers alone, whereas the model results are about the modulation on top of the linear response driven by the inducer impinging on the neuron's RF. In addition, a small minority of model neurons is modulated, and half of those are suppressed, not facilitated. So, I find it difficult to interpret the results. Is there any relation to surround suppression phenomena (which also involve feedback components)? Does suppression vs. facilitation depend on RF size or position? Overall I am not convinced that, with the current demonstration, this is a strong model of the illusory contour phenomenon.

The Reviewer accurately points out the multi-level consequences of 'limited field of view': it affects the analysis of the immediate surround of a Z₁/V1 receptive field but also affects the access to other components of natural images statistics. We addressed the former with new analyses by increasing patch size to 50 px in order to strengthen the results of the paper, most importantly the illusory contour study. This permits to limit the overlap between the receptive field of the model neurons and the stimulus, thus reducing the linear response considerably. However, the increase in patch size was limited by the extra compute requirements of the larger model. We now perform additional experiments with an extended patch size and address the issue at relevant points in the manuscript (pg 11, lines 369-372). We have also performed a more careful analysis of the direction of top-down modulation. The larger patch size made it possible to analyze the top-down modulation in more detail. Because of limited patch size, a rigorous control over the receptive field properties of the neurons included in the analysis was not possible in the 40-px model. We found that a rigorous control of the experimental settings (1, limited overlap between the Kanizsa stim and the RF; 2, Kanizsa stimulus is required to fit onto the image patch for all shift values; 3, peaks of contour/Kanizsa and TDVAE/linear responses are required to be aligned) eliminated all neurons that displayed inhibitory effects for the Kanizsa stimulus. The updated panels are now included in the manuscript (Fig. 4).

A general issue is that while the findings are of potential significance, some effects sizes in the model are quite small (Fig. 2g and 5b,f). This should be explained.

We have revisited Fig 3g using the extended set of texture families (proposed by this Reviewer) and this resulted in a more pronounced difference between the texture and phase scrambled settings. Further, we clarified the standardization of texture images, which also contributed to a cleaner analysis. Specifically, whitening was applied to input images. However, this whitening was not consistent for natural, texture, and phase scrambled images, which inconsistency was eliminated. The updated statistics can be found on the new Fig. 3g panel. We have updated the Methods accordingly. Similar to

Fig 3g, consistent whitening at Fig 6f also increased effect size. We have also reevaluated the narrative and in order to streamline it, we have replaced the original Fig 6b (difference between within-family and across-family correlation matrix dissimilarities) with the panel originally shown under Extended Data Figure 9b (difference between within-stimulus and across-stimulus similarities). Please also note that inspired by the question about effect sizes, we revisited the question of the mismatch between the *magnitudes* of dissimilarities in the model and in the experiments. The magnitudes in the model appear to be larger than those in the experiments. We have clarified the reason for that in the text but here we also expand on this issue. In order to focus on the covariability resulting from top-down interactions, we used the mean of $q(Z_1 | Z_2, x)$ when sampling $q(Z_2 | x)$. This ensured that private variability of the variational posterior (as it is an uncorrelated Normal) is not interfering with the covariability resulting from sampling Z_2 . While we believe that such a treatment is an advantage of the theoretical approach, we extended the analysis with sampling not only Z_2 but Z_1 as well. Three combinations of Z_1 and Z_2 samples are now shown on Extended Data Figs. 10c,d, which are compatible with the time window used in the Bányai et al (2019) experiment and correspond to changes in the details of sampling strategies.

The conclusions drawn from the study are also somewhat limited due to the reliance on a specific subset of stimuli (Fig. 1), raising questions about the model's generalizability. A broader comparison involving all 15 texture families mentioned in the neural data could provide a more comprehensive evaluation.

We agree with the reviewer that the paper does benefit from a better alignment of the experiments and modeling. We extended our analyses to a larger set of texture families, matching the size used in the experiments. We have updated all the relevant figures and modified the text accordingly (Figs. 2, 3, 5, & 6).

The inconsistency in the number of texture samples per family and the number of model neurons in various sections of the results is also a point of concern, with no clear explanation provided for these variations.

We thank the reviewer for drawing our attention to this issue. In our experiments with the TDVAE we strived to reproduce the conditions of the experiments. We clarify this in the relevant Methods sections ('Comparison texture image dataset', 't-SNE visualization', 'Classification decoding').

References

Line 51: "Importantly, learned invariances mean that high-level task variables do not necessarily carry the information required for other tasks and lower-level variables need to be queried for optimal inferences instead." Work by the DiCarlo lab showed that some low-level information (e.g. positional, viewpoint) is retained and even enhanced
<https://pubmed.ncbi.nlm.nih.gov/26900926/>

We thank the reviewer for drawing attention to this. As a result of a rewrite of the introduction, this point was eliminated from the text.

Line 77: "...textures... critically contribute..." provide references?

References added

Line 80: is ref 19 (Nurminen) about textures? A more relevant one seems Okazawa in area V4 <https://pubmed.ncbi.nlm.nih.gov/25535362/>

We thank the reviewer for pointing this out and bringing our attention to the Okazawa paper. We replaced the citation as suggested.

Line 402 etc, "with non-hierarchical and essentially linear models" is an ok description of some of the references cited, but not really for probabilistic mixtures of GSMs like ref. 49 and related prior publications: the inference of mixture component is hierarchical (in the sense of a hierarchical generative model, not of a multistage network) and the resulting inference of low-level variables and neural activity are profoundly non-linear.

We clarified this part of the text about the different approaches including the GSM model. (pg 18, lines 547-553)

Line 463 etc: I agree the SSN produces realistic response statistics and has been used to approximate probabilistic inference, but that literature does not include top-down feedback and, at face value, could be interpreted as evidence that top-down feedback is not really needed for probabilistic inference in a (much simpler) generative model. Seems worth discussing more accurately.

We are sorry for this confusion: SSNs are not available for hierarchical computations, the text has been clarified (pg 18, lines 585-586). We did not mean to imply that top-down is required for probabilistic inference, instead we argue that flexible inference requires hierarchical generative models, and hierarchical generative models naturally accommodate top-down computations.

Clarity

The introduction focuses much more on the V2 texture selectivity, than on the V1 top-down modulations. I suggest changing this, because the V1 predictions are the most distinctive with respect to feedforward discriminative models. The strength of the normative proposal of this manuscript is that with a single computational goal, it reproduces different phenomena.

We thank the reviewer for this suggestion. We have rewritten the introduction inspired by this comment, as well as we have rewritten the first paragraph of the discussion (pg 17 line 526).

Further, to strengthen the emphasis on top-down processes, we now extend the paper with additional analyses that establish links with additional experimental data. First, we show that texture decodability in Z_1 depends on feedback from Z_2 (Fig. 5a). Further, inspired by results from Corey Ziemba (Ziemba et al (2019) J Neurosci), where the authors demonstrate that the temporal dynamics of sensitivity to texture statistics in V1 suggests a top-down component, we analyzed sensitivity to texture statistics in V1 responses with and without texture-specific feedback from V2 (Fig. 5c). This new analysis emphasizes that texture-level statistics emerge in V2 and is then fed back to V1. In an additional experiment, we reproduced the conditions of the contour integration scheme of Chen et al (2014), Neuron, and showed that TDVAE displays similar top-down modulation effects in model V1 to those found in V1 of non human primates (Fig. 5d, Extended Data Figs. 9c,d).

The TDVAE uses a complicated implementation, with the V1 feedforward activations being used twice: once to provide driving input to V2, and a second time to provide driving input to V1 while integrating the V2 feedback. I guess this is so there is a single step of FF-FB interaction, rather than a truly recurrent system. It seems important to motivate this more clearly, and discuss the biological plausibility.

The motivation for the specific architecture is dual. From a machine learning point of view, the joint component of the two functions of the recognition model ($q(Z_2 | x)$; $q(Z_1|x,Z_2)$) (V1 and V2 in the model are renamed to Z_1 and Z_2, respectively, in the updated manuscript) is a principled modification of the architecture, and is under the umbrella of ‘parameter sharing’, a standard method in machine learning (Sonderby et al (2016) NeurIPS). From a biological point of view, this choice is motivated by anatomical constraints: relying on a direct pathway to Z_2 is not biologically justified but a feed-forward pathway relaid in V1 has been established (Callaway & Wiser (1996) Vis Neurosci). We speculate that the layer Z_1, which integrates feed-forward and feedback pathways, is in layer 2/3. We expand on the motivation in the Results section (pg 5, lines 171-185) and include a discussion of this point in the Discussion section (pg 19, lines 620-632).

Beyond this motivation, in the updated manuscript we present the results obtained with an architecture that features disjunct functions for the recognition model without parameter sharing. These results confirm that key properties of the recognition model are not affected by this choice. We have updated the manuscript accordingly (lines 212-220).

The model predictions for neural activity are made under the sampling hypothesis, but this is not stated clearly enough, and a reader not familiar with sampling-based neural code might be

confused as to how the responses are actually modeled. (eg. expanding text around line 121). A related point is that the manuscript points to refs 37,38 for how sampling-based representations capture trial-trial variability; that is fine, but those papers use the simpler GSM model, where the modulation of Fano can be understood (approximately) analytically, which is not the case for TDVAE; it would be important to show if stimulus dependent Fano factors resulting from the TDVAE are also consistent with V1 data.

We thank the reviewer for pushing this point. We agree with the reviewer that its relationship to response variance needs to be explained, which could provide extra intuition on how the posterior in the model is related to measurable quantities of response statistics. We have updated the text to make the sampling hypothesis more clear and its role better explained both in the Results (pg 7 lines 222-233) and in the Discussion (pg 18 lines 567-584). We also included a more detailed explanation in the Methods under subsection ‘Relating inference in the generative model to neural activity’.

With respect to the stimulus statistics dependence of response variance (referred here as variability in spike count, Fano factor), the Reviewer precisely points to the use of an important generative model of natural image statistics, GSM. The Reviewer refers to it as a simpler model that is limited to a largely linear relationship between latent variables and image pixels, reminiscent of the lower hierarchical layer in TDVAE, Z_1. While that model lacks the nonlinearity implemented in the higher hierarchical level, Z_2, it actually implements a very mild, very simple nonlinearity, the global scaling in the generative model. In former accounts of stimulus-dependent variability of responses this scaling had a central role (Orbán et al (2016) Neuron, Festa et al (2021) Nature Comm, Echeveste (2020) Nat Neurosci). This nonlinearity could be integrated in the lower component of the generative model, which is both feasible and would also be a promising extension of the Variational Autoencoder framework. We strive to streamline the argument of this paper to deliver more efficiently the contribution of the higher hierarchical layer. This nonlinear extension of the lower layer of the generative model is discussed in a separate study (Catoni et al (2025), arXiv). Nevertheless, we take this opportunity to extend the Discussion to cover this topic (pg 19, lines 633-640).

The choice to report membrane potentials, which include negative values, as a measure of model neuron activity could be explained and motivated more. It would be beneficial to also consider the implications of applying non-linear transformations to better represent neuron firing rates and to justify the preference for membrane potentials over firing rates.

We thank the reviewer for highlighting this point. We have now clarified this choice at the beginning of results (pg 7 line 245-249). Here we describe our motivation so that the discussion can be openly performed. We agree that a firing rate and/or spiking account is a relevant aspect of modeling cortical responses. The extension of VAEs we developed is new to the field (both in ML and computational neuroscience), which was built on a number of important insights. These components are both principled from a mathematical point of view and stringent in the sense that we attempted to limit the ‘bag

of (industry standard) tricks'. This resulted in a model that is readily interpretable from a machine learning point of view, which ensures a high level of transparency of assumptions. In this sense, the introduction of a firing rate-based or spiking based deep generative model would have added a layer that is not common in the field and the reader would have a harder time to disambiguate if the results are consequences of this specific choice or more general results coming from standard VAE/probabilistic generative modeling principles. In summary, rate and spiking based extensions establish a definitely exciting avenue, which is also supported by existing tools for Poisson-latent VAEs.

We now summarize these points in the text.

Line 192: "TDVAE has no free parameters." Do you mean to fit neural data? Because it does have parameters to fit image statistics obviously.

We thank the reviewer for highlighting this. The assumption is correct, we have clarified this in the text (pg 7, line 219-220).

Line 322 etc: I am confused by the noise manipulation. The original, clean Kanitza stimulus should be maximally uninformative for the target neuron because there is no direct input to it (assuming the neuron is not driven by the inducers, of course). So the response of the neuron in that case should be entirely determined by the contextual prior. And then, adding noise to the entire stimulus will make the inducer noisy, and so should make the contextual prior broader, weaker. These points are worth clarifying.

Thank you for raising this issue. We expected that the *relative* contribution of top-down signals becomes stronger as the blur in the image is increased. We expected this to happen because the feed-forward contribution to Z_1 weakens. Nevertheless, we acknowledge that this experiment depends on a number of factors and ultimately we realized that the arguments of the paper are not directly served by this analysis. As Figure 4 is now extended with a set of new experiments (see the new Fig. 5), we decided to discard the three panels investigating the effects of blur in the Kanizsa stimulus.

Fig 4m, you could add horizontal and vertical lines at $x=0$, $y=0$ for clarity.

According to other suggestions, this panel was removed from the text.

Fig. 5, the colors red and purple are used to contrast different conditions in panel c,e,f,g

We thank the reviewer for pointing this out. Colors have been revised on this figure.

Minor Comments, typos

The paper could offer a clearer explanation of the terms "deep" and "shallow" non-linearity as used in the Fig. 2a.

We thank the reviewer for highlighting this issue, we have clarified the caption of the figure, including references to the table containing the parameters (Fig. 2, pg 6).

How results would change in a deeper TDVAE (>2 layers), for instance would the texture representation emerge in the second layer or in the deepest, or somewhere in between?

This is an interesting question that deserves investigation on its own. As our study highlights, inductive biases that are realized through limited computational complexity (ie. linear generative model in Z_1) shape the representations at intermediate layers of a hierarchical VAE. We assume that similar limits determine the presence of texture representations. These inductive biases can actually encompass other forms of anatomical constraints, eg. the ‘field of view’ of a neuron at a given level of the hierarchy (ie. receptive field size), too. This question is realistically a subject of upcoming studies but we expand on them in the discussion (pgs 20-21, lines 694-709, 714-720).

The clustering of V1 model neurons appears to present discrete values, in contrast to the broader range observed in V2 model neurons (Fig. 2e). Clarification on why V1 projections show discrete values would be informative.

With the extended set of texture families this apparent regularity is not present (see new Fig. 2e).

The schematic representation of the model could benefit from refinement for better clarity and ease of understanding (Fig. 1F).

We thank the Reviewer for highlighting this. We now include a new figure panel that complements the existing one such that a cleaner computational graph is accessible (Extended Data Fig. 1).

Fig. 5d and related text: is the number of predictors matched, when comparing decoding performance from V1 means, V2 means, and from correlations?

We thank the reviewer for pointing this out. We now match the number of Z_1 neurons* for mean decoding performance and correlation decoding performances. As the population is smaller for Z_2 , the predictors are not matched between Z_1 and Z_2 . We make this clear in the caption.

Line 299 “was similar” -> were similar

Text corrected.

Line 799 “learned a complete linear basis” is it complete or overcomplete?

The architecture permits slight overcompleteness, however the learned model only uses a complete set. We now clarify this in the Methods (under section Architectural details).

Reviewer #3

Overview:

This paper leverages recent advances in generative modeling in machine learning (the hierarchical VAE framework) to create a two-level image-computable model of hierarchical inference in the primate brain. This is a logical continuation of earlier work by the authors and by others which proposes that visual processing in the brain can be understood as inference in a task-independent generative model of natural images. Here, top-down or feedback connections from V2 to V1 communicate contextual prior information. These core ideas are not themselves new, but the authors have nonetheless made a significant contribution in terms of (i) technical developments in the model, (ii) performing experiments to probe the behavior of a hierarchical VAE, and (iii) linking that behavior qualitatively to relevant data from primate electrophysiology data. I am a fan of this line of work, but I have a number of concerns detailed below. I look forward to the authors' responses.

We thank the reviewer for the summary and for their positive words.

Comments on presentation:

- Referring to the model as "V1" and "V2" was very confusing as it leads to statements like "V1 had a similar effect to V1 in neural data." The beginning of the paper introduces model layers as L/M/H. I suggest sticking with this terminology so that "Layer L [in the model] had a similar effect to V1 [in neural data]."

We thank the reviewer for the suggestion. We see the point that our choice easily leads to ambiguous terminology. We switch a terminology that resolves ambiguity while also preserving the analogy between the biological and modeling layers. We chose to use Z_1 and Z_2, which is also aligned with the machine learning terminology.

- The term "goal-general" strikes me as inventing a new term for something that already exists in the literature as "task-independent" or "constitutive" representations of uncertainty (see Koblinger et al (2021) or Lange et al (2023))

We thank the reviewer for the suggestion. We saw some appeal in contrasting naming for the two paradigms, but ultimately agree that widespread terminology has more advantages. We have updated the text accordingly.

Very high-level comments on the approach and conclusions:

- The most important thing I'd like to press the authors on is this: this paper uses a powerful computational model (the TDVAE), but analyzes its behavior qualitatively in terms of reproducing high-level patterns/trends in neural data. I have nothing against this qualitative approach -- indeed, it is impressive when a model with no free parameters "fits" neural data even if that fit is qualitative. However, qualitative trends still need some null model or basis for comparison.

We thank the reviewer for making this point. We appreciate this concern and adopt/propose additional analyses, which we detail below.

All of the results in Figs 3, 4, and 5 illustrate qualitative similarities between the TDVAE and actual neural data. These results are difficult to interpret without also seeing a *lack of* qualitative agreement in other models. Other models could include (i) an untrained TDVAE or a TDVAE trained on unnatural stimuli, (ii) the TDVAE with the top-down feedback ablated, (iii) the shallow VAE from Fig 2, etc.

We have repeated the experiments with the relevant controls. We appreciate the suggestion to expand the results by using ablation experiments. Here we briefly clarify the ablation idea in the context of TDVAE. In Figure 4g we introduce an analysis where the Kanizsa experiment is performed with sampling the prior instead of the variational posterior. This experiment can be considered as a version of ablation as it corresponds to severing the connection $q(Z_2|x)$, such that no image information is available at Z_2 , which formally results in sampling the prior. Inspired by the Reviewer's comment we now perform an additional experiment, in which we specifically reduce information from Z_2 . As there is a single function that integrates x and Z_2 inputs to Z_1 , $q(Z_1|x, Z_2)$, removing this completely is not useful. However, we propose to clamp Z_2 to zero. We repeat the Kanizsa experiment with this clamped version too, which is now included in Figure 4g. With this, we expand on the contribution of an intact Z_2 to the code in Z_1 : we extend the analysis of the Z_2 prior-sampled model to show that elimination of stimulus-related information from Z_2 reduces texture family decodability in Z_1 (Fig. 5a, pg 13, lines 421-429). According to the Reviewer's suggestion, we extended the paper by including a feed-forward goal-directed model and an untrained VAE, which we contrast with results of TDVAE. This is now shown on Fig. 2a,d and Extended Data Fig. 9a and discussed in the Results (pg 6 lines 208-209, pg 8, lines 291-304, pg 12 lines 391-394, pg 21, lines 697-700).

- The framing of the results in terms of "top-down" effects is not entirely supported by the results.

As the authors explain (starting on L132), there are many potential ways to instantiate a prior, and we have good reasons to believe that some long-term environment statistics are encoded in the brain implicitly such as in the density and sensitivity of different filters (cf. various works by Wei and Stocker). In the *model* presented here, the primary mechanism for encoding the prior $p(\text{layer } L)$ (or $p("V1")$) is using feedback from layer M (or " $V2$ "), and otherwise layer L in the model is kept deliberately simplistic. Effects that appear *in the model* due to including a

second "V2" layer could plausibly be implemented in the brain in a purely feedforward fashion. Strong evidence for feedback in the brain comes from inactivation, fast context switching, cross-modal effects, or time delays. (Notably, Lee and Mumford (2003) show time delays in illusory contour responses in V1, which is perhaps their strongest evidence for a feedback or recurrent component there). To summarize this point: effects that appear in the model to require feedback do not necessarily require feedback in the brain, especially given that we know V1 in the brain is more complex than layer L in the TDVAE. A more complex V1->Image generative model would (I conjecture) reproduce some of the effects highlighted here without the addition of the V2 layer.

We agree with the reviewer that priors can come in many forms as exemplified by the works cited by the reviewer or others (Ganguli & Simoncelli (2014) *Neur Comp*; Vacher et al (2022) *Neural Networks*). In our paper, we are arguing for a specific form of prior, contextual prior, a flexible form of prior that is capable of changing with stimulus content. With respect to the illusory contour experiment presented in the paper, we agree with the reviewer that in the *neural data* the time delay seen in the evolution of the illusory contour effect is a critical component for claiming top-down contributions. Once the experiment has established that the contextual prior required for the illusory contour emerges with the contribution of top-down influence, our paper aims to 1, establish a normative model that learns a top-down inference model based solely on natural images; 2, demonstrate that this provides a parsimonious model of the illusory contour experiment; 3, the effect is indeed relying on top-down influences. While the time delay has explanatory power only in the electrophysiology experiments, we believe that performing an analogous experiment in the model provides additional insights into the working of the model. We include a figure panel showing the temporal evolution and related text in the manuscript (Fig. 5b, page 13, lines 429-440). In addition, we are now elaborating on another aspect of the effect of top-down feedback by investigating texture decoding performance in Z_1 under two conditions: using the TDVAE model, and the TDVAE with the feed-forward input to Z_2 severed, ie. sampling Z_2 prior instead of the Z_2 posterior (Fig. 5a, page 13, lines 421-429). This experiment extends the investigations that assessed mean responses relying on Kanizsa stimuli. Further, we extend the results to cover additional evidence from non-human primate recordings on delayed discrimination of texture/phase scrambled stimuli (Ziemba, 2019 *J Neurosci*, Fig. 5c) and delayed effect of contour integration (Chen et al, 2014, *Neuron*, Fig. 5d). We agree with the reviewer that elaborating on the various forms of priors has clear benefits for the readers, and therefore we extended the Discussion with a paragraph reflecting this (page 20, lines 654-657).

Technical comments:

- On L152-155, the authors claim that the TDVAE architecture is "as general as possible," but the reality is that all of the results depend strongly on architectural details of the recognition model. This recognition model has quirks such as a deterministic image -> V1 -> V2 pathway

which is neither very biologically plausible nor is it the only way to do inference in a hierarchical model. I see three feasible ways out of this:

(1) the authors could lean into the fact that their results do depend on recognition model details. The challenge is that this avenue reduces the generality of the results and invites further criticism that the recognition model itself appears biologically implausible.

(2) the authors could show that their main claims hold for other kinds of inference algorithms such as sampling from $p(V_1, V_2 | \text{image})$ or doing message passing. This would be much harder but would support the claim of generality.

(3) the authors could use importance sampling or self-normalised importance sampling (SNIS) to compute the actual moments (means, variances, and correlations) of the true posterior. This would be most consistent with what is said circa L196 which suggests an interpretation in terms of neural sampling from a true posterior. Doing this would allow the authors to make more precise claims about means, variances, correlations, etc. However, this would transform the results from a study of inference to a study of generative models, thus this option would also call into question any mechanistic "top down" interpretations.

We appreciate this insight of the Reviewer. We attempt to break down this comment and provide specific answers to specific parts of it.

First, thank you for pointing to our claim on the generality of the architecture. We clarify this statement to avoid confusion (pg 5, lines 162-165, 171-185). Briefly, what we meant by this claim was that apart from the approximations of the VAE framework (optimizing the ELBO instead of directly learning the log marginal likelihood; using variational inference and more specifically an amortized version of it) there are no further approximations, similar to early attempts for hierarchical VAEs (e.g. ladder VAE), approximations for inference (e.g. beta VAE) or approximations in the training (gradient clipping), and avoided a convolutional architecture, which biases the learned representation.

We also point out that in our paper we thoroughly investigated the effect of factorization of the variational posterior. As the reviewer correctly pointed out $p(V_1, V_2 | \text{image})$ is the most general form of the posterior. In an amortized variational inference setting we believe that it is a natural assumption that the variational posterior factorizes according to one of the factorizations described in the paper. We trained models for both scenarios and found that the top-down model is characterized by a higher ELBO. Nevertheless we pointed out what is qualitatively common for the two models and which properties are distinct. Namely, both models give rise to V1-like filters in the lower level and texture selective units in the higher level. However, only TDVAE is capable of capturing meaningful noise correlations at Z_1 . There is a specific choice of the recognition model, which is motivated by industry standards and biological arguments, but not scrutinized thoroughly: the shared parameters of $q(Z_2 | x)$ and $q(Z_1 | x, Z_2)$ at the initial stage of computation. We now include the architecture which does not include parameter sharing, and thus features disjoint neural networks for the two functions. The updated text describes this comparison (pg 5 line 178-181, pg 7, line 212-219).

Second, we are not entirely sure about the meaning of the description ‘deterministic image -> V1 -> V2 pathway’. Variational autoencoders implement probabilistic inference and yield a probability distribution (variational posterior). Here we expand on the deterministic/stochastic distinction in the context of learning and in the context of inference. In the context of learning, training of this model takes place through stochastic sampling of this posterior. Although in general a single sample is taken for a given image, it is still a stochastic process. There are extensions, in which multiple samples are taken (Importance-weighted Variational Autoencoders, IWAE, Burda et al (2015) arXiv), which ensure a tighter ELBO as well as a more expressive posterior (evidently) (Cremer et al (2017) arXiv). We have actually extended IWAE to accommodate TDVAE in a separate study, but did not include this exploration in the current paper as we did not find qualitative advantages of this more intricate approach over the current model and we believe that the reader would benefit little from this addition while making the storyline more convoluted. In the context of inference, the question is more directly related to the message of the paper as it concerns how we map TDVAE to neural activity. In the sampling interpretation in order to relate the posterior yielded by TDVAE to neural activity, we argue that stochastic sampling is performed. In the electrophysiology results that serve as a comparison to modeling results it is the trial-averaged mean that is reported, and therefore we interpret those results in terms of posterior mean. While we could perform the sampling-averaging process to make this evident, we bypass sampling and rely on the simple posterior mean. So, indeed, in this context we are relying on characteristics of the TDVAE that is deterministically computed from images, but variances are omitted only because we are contrasting modeling results to trial-averages of neural data. Note, that in machine learning applications sampling the variational posterior is rarely performed but doing sampling is perfectly justified, even if the posterior is an approximate one. As presented above, sampling the posterior is not necessarily a critical factor to interpret neural data. There is one spot, however, in which actual sampling of the posterior is core to the results of the paper: the correlation experiments rely on the covariances obtained through sampling the Z₂ posterior (earlier denoted as V2 posterior). We are now elaborating on this further as this issue is related to another question of this Reviewer. Based on this interpretation of the Reviewer’s comment, we have put extra effort in clarifying this concern and, indeed, to address the next comment of the Reviewer on the ambiguous terminology (Methods: ‘Relating inference in the generative model to neural activity’).

Alternatively, another interpretation of the Reviewers comment on deterministic image-> v1-> v2 mapping can be that despite the neural network-parametrized $q(Z_2 | x)$ variational posterior naturally accommodates response variability at the level of Z₂, this model does not address variability in the hierarchy of processing layers leading to Z₂. We relate this issue to the diversity of roles different cortical layers play in processing visual input. The presented theory assumes auxiliary variables that are not directly representing a probability distribution and their role does not identify stochastic elements. This is indeed an assumption that is worth emphasising. Variability in these layers can still be inherited from earlier processing stages that are not covered in this

paper. A more interesting insight is that parameter uncertainty in the model of the environment was proposed to be reflected in stochasticity in synaptic transmission (Aitchison (2021) Nat Neuro), which is a source of variability that can explain excess variability not addressed by the hierarchical inference framework presented here.

Third, the Reviewer proposes alternative ways to obtain samples from the neural data, which is actually closer to sampling directly the generative model rather than the approximate recognition model. Based on point two above, we believe that we provided convincing evidence that sampling the recognition model provides valuable insights into response statistics. Nevertheless, we reflect on the valuable insights of the Reviewer in the Discussion (pg 18, line 567-584).

- Addressing the previous bullet would also, I hope, alleviate some confusion with terms like "mean response" which ambiguously could mean an empirical mean of model responses, the posterior mean, or the "mu" part output by the recognition model, which are three related but distinct things.

We have extended the description of how model responses correspond to experimentally measured quantities (pg. 7 line 223-249). We have carefully reviewed the text to resolve these ambiguities.

- On a related note, the authors could use the same importance sampling trick to compute the model evidence directly rather than the ELBO, which would make it easier to compare model quality in Fig 2a. Comparing ELBO values is (unfortunately) standard, but ultimately hard to interpret because the ELBO may be low simply because the recognition model is poor, regardless of the quality of the generative model. Since both the recognition and generative components change across models, direct comparison of the ELBO is not a direct comparison of the quality of the fit to natural images.

We highly appreciate this effort to push for a more principled approach when performing probabilistic computations. We agree that evaluation of the model through sampling the learned generative model is more principled as the ELBO calculation ignores the mismatch between the true and variational posteriors and agree that the machine learning field would benefit from developing such stronger benchmarks. We would be enthusiastic about introducing this problem in a machine learning publication, which could explore the conditions under which the ELBO and sampling-based evaluation yield contrasting results. However, we believe that relying on industry standards has benefits for the readership of this journal, as contrasting ELBOs is a standard procedure.

- Fig 2b: it would be nice to see second-order receptive fields to show complex cells (i.e. response-weighted covariance in addition to response-weighted average). One cannot conclude much about the selectivity of the model's V2 cells from a uniform gray receptive field.

We have now updated the relevant panel with second order RF analysis.

- L196-198 suggest a neural sampling framework where responses are samples from the posteriors. L198-203 then frames sampling as out of scope of this paper. I have two relatively minor issues with this: (1) I suspect including this response variance as part of the model (perhaps using the importance sampling trick described above) would improve the qualitative fit to some of the data, especially in the analysis of response variance in Fig 3c.

We thank the reviewer for pushing this point. We agree with the reviewer that the model's relationship to response variance needs to be explained, which could provide extra intuition on how the posterior in the model is related to measurable quantities of response statistics. We have updated the text to make the sampling hypothesis more clear and its role better explained both in the Results (pg 7 line 223) and in the Discussion (pg 18 line 567). We also included a more detailed explanation in the Methods under subsection 'Relating inference in the generative model to neural activity'.

The actual figures referred to the Reviewer (Fig 3c) is analyzing variance of responses across images. As these responses are calculated in experiments as mean spike counts in a finite time window, variance jointly comes from the variance of the mean across images and from sampling variance. We strived to reproduce the conditions of the experiments by matching the number of samples for a given image with the length of the trials (assuming 20-ms sampling time).

Second (2), I am unconvinced that the statement in L201, "without loss of generality ...," is true. How could including a nonlinearity between posterior samples and neural responses have no effect on the results, which analyze neural responses?

We thank the reviewer for giving us a chance to revisit this point. We have now clarified this choice at the beginning of results (pg 7 lines 245-249) and expand on this in the Methods (section 'Relating inference in the generative model to neural activity'). Here we describe our motivation so that the discussion can be openly conducted. We agree that a firing rate and/or spiking account is a relevant aspect of modeling cortical responses. The extension of VAEs we developed is new to the field (both in ML and computational neuroscience), which was built on a number of important insights. These components are both principled from a mathematical point of view and stringent in the sense that we attempted to limit the 'bag of (industry standard) tricks'. This resulted in a model that is readily interpretable from a machine learning point of view, which ensures a high level of transparency of assumptions. In this sense, the introduction of a firing rate-based or spiking based deep generative model would have added a layer that is not common in the field and the reader would have a harder time to disambiguate if the results are consequences of this specific choice or more general results coming from standard VAE/probabilistic generative modeling principles. In summary, rate and spiking based extensions establish a definitely exciting avenue, which is also supported by existing tools for Poisson-latent VAEs.

The results in Fig 3 compare the model to neural data only qualitatively, I suspect that the qualitative fit would be even better if the V1 -> Image portion of the TDVAE generative model was more rich than linear+gaussian. A linear-gaussian V1 layer will develop simple cells but not complex cells, for instance. This likely impacts results in Fig 3e for V1 show very little ability to classify texture families. A moderately more complex, moderately nonlinear V1 -> Image generative model would likely further improve the qualitative fit to neural data here and elsewhere in the paper. This would be a substantial change for a minor improvement which I hope the authors will consider for future work if nothing else.

We agree with the reviewer that the generative model of Z_1 is simpler than V1. We strived to keep the narrative simple but we acknowledge that a better fit to the texture representation at V1/V2 could be even more improved. We will take the Reviewer's advice and will explore this avenue in the future. We have recently made strides towards mild nonlinear extension of the generative model by implementing a scale-mixtures variant of the VAE which provides promising improvements both in terms of efficiency of inference and relation to neural response statistics (Catoni et al (2025) arXiv, pg 19 line 635-640).

- Data shown in Fig 4c are not by themselves conclusive that top-down effects are present in the brain. They just show that context matters, and this could "in principle" be implemented in a feedforward or recurrent manner. Similarly, Fig 4b,f,g does not establish "top down" connections as the source of the effect in the model. If the authors want to emphasize "top down" effects, then wouldn't the relevant comparison be between TDVAE and TDVAE with the V2->V1 connection ablated? Or is the "linearized" model equivalent to ablating feedback? (Seems unlikely given the difference between projective and receptive fields).

This question is related to questions addressed above, which we expand here with relevant remarks. The analysis presented on 4c is demonstrating that the effects that were demonstrated in primate and mouse experiments and which have been attributed to top-down influences can be explained based on *hierarchical inference* with a model *end-to-end trained on natural images*, thus giving a normative framework to interpret and predict top-down influences. We agree that being able to reproduce illusory contour responses alone would not be sufficient to argue for the necessity of top-down computations. Along with the extensions of Fig 4, we have rewritten the first paragraph of the discussion, which shall provide a clear scope of the paper (pg 17, line 526). With respect to Fig 4b,f, g, we refer to the earlier response we gave to the same Reviewer on the two forms of ablation, one that is present in the current manuscript and another one inspired by the remark of this Reviewer where feedback information is reduced.

- Also regarding Fig 4b-c, Lee and Mumford (2003) showed a time delay in the V1 response to illusory contours relative to the response to actual lines, which is a much stronger indication that this effect involves "top-down" effects compared to what is shown in Fig 4c. Could this time-delayed effect be simulated in the TDVAE, e.g. by comparing the image->V1 response to the response once the image -> v1 -> v2 -> v1 pathway executes?

We now include a simulation of this effect in the updated Figure 4 (Fig 5b) and expand on modeling the time-delay experiments (pg 13, lines 429-440).

Further, to strengthen the emphasis on top-down processes, we now extend the paper with additional analyses that establish links with additional experimental data. Inspired by results from Corey Ziemba (Ziemba et al (2019) J Neurosci), where the authors demonstrate that the temporal dynamics of sensitivity to texture statistics in V1 suggests a top-down component, we analyzed sensitivity to texture statistics in V1 responses with and without texture-specific feedback from V2 (Fig. 5c). This new analysis emphasizes that texture-level statistics emerge in V2 and is then fed back to V1. In an additional experiment, we reproduced the conditions of the contour integration scheme of Chen et al (2014), Neuron, and showed that TDVAE displays similar top-down modulation effects in model V1 to those found in V1 of non human primates (Fig. 5d, Extended Data Figs. 9c,d).

- L322 and Figs 4k–m: while the statement "inference relies on the prior more when the evidence about observations is degraded" is true, it is not obvious that blurring the image constitutes "weak evidence" in the technical sense required here. A blurry image is nonetheless an image, albeit one that is perhaps away from the manifold of natural images. Then again, the Kanizsa stimuli are also away from the manifold of natural images. Whether or not each of these constitutes "strong" or "weak" evidence -- in the sense of how it does or does not move the posterior away from the prior -- depends on other model details such as the functional form of $p(\text{image}|v1)$. In a linear Gaussian image model (as used for $p(\text{image}|V1)$ here), the likelihood function $p(V1 | \text{blurry image})$ does not broaden indefinitely as the image gets blurrier. Say, for instance, one of the columns of the A matrix is constant (DC component of images). Then, the blurrier the image, the more it will shift posterior mass onto that component, which could feasibly *lower* the posterior entropy compared to a sharp natural image. Any image the model is conditioned on is "evidence" even if the image was constructed by a noisy or blurring process. Notably, a GSM would be different, since a dark image is explained by the scale term which removes any constraints on the other latents. The same property does not hold for a linear-gaussian image model and I don't see an obvious way to reliably drive this model towards its prior simply by manipulating the input. I therefore cannot interpret any of the results in Figs 4k-m.

Thank you for raising this issue. We expected that the *relative* contribution of top-down signals becomes stronger as the blur in the image is increased. We expected this to happen because the feed-forward contribution to Z_1 weakens. Nevertheless, we acknowledge that this experiment depends on a number of factors and ultimately we realized that the arguments of the paper are not directly served by this analysis. As Figure 4 is now extended with a set of new experiments, we decided to discard the three panels investigating the effects of blur in the Kanizsa stimulus.

- It is unclear when first reading the noise correlations section why there is any variability in the model's responses to identical stimuli given that it's established earlier that only the means

produced by the recognition model are being analyzed, and the recognition model assumes a factorized posterior (zero correlations). It took some digging to uncover the source of variability and correlations in V1 in the model. (I also did not find this detail searching the methods.) I took a few minutes to navigate the code, and from what I understand, "noise correlations" in V1 here refer to the fact that the image \rightarrow V2 \rightarrow V1 pathway involves the reparameterization trick in V2, so V1 is conditioned on a single random sample from V2, but otherwise the recognition model is deterministic. This seems to conflate properties of this family of recognition models with properties of the actual posterior. If I take what is said in L196-198 seriously, then I should view both neural variability and noise correlations as reflective of the actual variance and correlations in the posterior distribution $p(V1|image)$. This is in principle computable using SNIS as sketched above, but is not what the authors compute here. This again brings up my high-level point above: I think the authors need to commit either to analyzing moments of the actual posterior (e.g. using SNIS) or they need to commit to presenting their results as strongly dependent on the assumed functional form of the recognition model, which itself seems rather biologically implausible.

We are sorry for this confusion. Here we provide a precise explanation and in the text we make updates to make sure that this question is thoroughly addressed (section ‘Relating inference in the generative model to neural activity’). As the Reviewer precisely points out, any given image deterministically produces a posterior $p(Z_1, Z_2 | x)$ (note, that Reviewer 2 suggested a shift in naming the variables to avoid confusion), which practically breaks down into uncorrelated Normal distributions, $p(Z_2 | x)$ and $p(Z_1 | x, Z_2)$. These variational posteriors do not display any form of correlation, although the formalism itself would permit learning covariances. However, the quantity we are evaluating is $p(Z_1 | x)$, which requires marginalization over Z_2 . We practically perform sampling from the variational posterior from Z_2 , and perform a Monte Carlo integral through the samples.

Let us make two notes here. First, an interesting insight for hierarchical variational inference is that the partitioning of the joint posterior we use permits a posterior that is more expressive than the variational posterior for Z_1 , while the other ‘chain’ partitioning of the posterior permits more expressive posterior for Z_2 . Second, the standard practice in machine learning does not rely on extensive sampling of the variational posterior, as in a non-hierarchical version the variational posterior constrains the class of the posterior anyway. However, in hierarchical VAEs sampling is actually computationally advantageous to evaluate the posterior. Even in machine learning practice sampling is performed during training though, albeit usually having a single sample. There is an intuitive extension of the standard VAE formalism, Importance Weighted Variational Autoencoders (IWAEs), which explicitly relies on multiple samples during training. IWAEs both provide a tighter bound to the marginal likelihood than the ELBO of standard VAEs, as well as it implicitly encodes a more expressive posterior than the class of the variational posterior itself. Interestingly, we have extended the standard IWAE formalism to feature a latent hierarchy, and tested it with natural images. We obtained qualitatively similar results as the presented TDVAE model. We have decided to omit this model from

the paper as it did not bring any additional insight but made the narrative more difficult for a non-machine learning readership.

Returning to the original question on how noise correlations emerge, we argue that marginalization over Z_2 induces variance in the mean of $p(Z_1 | x, Z_2)$, which – when considering multiple samples from Z_2 – results in a mixture of Gaussians. This mixture is flexible enough to display a rich correlation structure. Thus, while the image is constant, changes in Z_2 induce correlations. Importantly, this correlation is then a top-down influence-dependent form of correlated activity. We have now expanded on this in the section ‘Relating inference in the generative model to neural activity’ in the Methods.

Miscellaneous minor comments:

- Why does Fig 2 only compare 5 synthetic textures if ref [18] used 15 textures? Would it be easy to add the other 10 texture families for a more direct comparison?

We thank the reviewer for drawing our attention to the benefits of stronger alignment with the experimental paradigm. We have extended the analysis with a 15-family texture set.

- Missing mean indicators in Fig 3h?

Figure corrected.

- The code does not run due to hard-coded paths in the environment dependencies. It also has hard-coded dependence on cuda and so couldn't be verified on my laptop.

We thank the reviewer for taking the time to look at the code and apologize for the problems encountered. We included an updated source code package that contains detailed instructions on how to run the code with and without an Nvidia GPU.

Reviewer #3 (Remarks on code availability):

The code is technically available but very hard to read. I glanced at it briefly to clarify a question I had about the methods. I did not review the code thoroughly. I was unable to set up a local environment to test things because the requirements.txt file appears to include dependencies specific to the authors' systems. I did not troubleshoot it further.

We believe that the new code package makes the code testable both on laptops without an Nvidia GPU and on workstations with an Nvidia GPU. The code is derived from the code of [Rao et al., Continual unsupervised representation learning, NeurIPS, 2019] and inherits its structure and 2019-era infrastructure. We agree that a rewrite with modern technologies would make the code more accessible.

Reviewer #4

Overall, we found this paper to be well written, thorough, and interesting. It provides a goal-general learning framework by inferring features of images at multiple levels in a hierarchy. The authors suggest an interesting Top-Down VAE (TDVAE) model for their hierarchical inference. The key aspect of their model is the use of top-down feedback from V2 to V1, such that neural activity in V1 depends not only on the stimulus but also on activity in V2. We support publication after addressing some relatively minor concerns.

Some of their methods need clarification. Their architecture figure, while connecting nicely to biology, is hard to follow computationally, and appears to hide a bunch of important computational structure. We attach a sketch of a proposed architecture diagram that makes things a lot clearer — though at the expense of revealing more cleanly how the “top-down feedback” is in fact just a longer feedforward pathway.

We are grateful to the reviewers for taking the time and effort to sketch this diagram. We strongly agree that it provides a clean overview of the computational graph that complements the existing cartoon. We have included an updated version of this in the manuscript (Extended Data Fig. 1).

Additionally, it's unclear to me from their main text, methods, and supplement how the generative and recognition architectures are related, except for $p(\text{image}|\text{V1})$. Is $p(\text{V1}|\text{V2})$ part of the generative model? What is the recognition model? These components should be explained better.

We have updated the text, Methods, and Extended Data to clarify the structure of generative and recognition components, as well as their relationship to empirical data (pg 5 lines 171-185, pg 7 lines 221-249, Methods / Generative models, Extended Data Fig. 1).

This hierarchical structure is nice and relatively tractable. It seems that another way of interpreting their architecture is as a feedforward network with a long skip connection from V1-ff to V1-INT. But this hides an important problem: recurrence. The authors should spend some time considering the consequences of assuming essentially a one-pass loop. I don't know that this is feasible through analysis, but it would be important to at least think about and discuss.

We thank the reviewers for highlighting this point. The mathematical reason for having a single pass comes from the combination of the top-down partitioning of the posterior and the variational approximation itself: as the parameters of the variational posteriors are ultimately passed forward through the parallel pathways including the Z_2 AND the skip connection to reach Z_1, and then Z_2 is not updated anymore based on Z_1, there is no recurrence needed at this point. Theoretically this is a highly relevant question, however, and an experimental paradigm for investigating how the representations in V1 and V2 become gradually refined would be highly welcome. We briefly address this argument in the Discussion (pg 19 lines 597-601).

The authors only have one layer of top-down feedback. This seems reasonable at first glance, given that they only cover V1 and V2. But each of these brain areas are themselves multi-layer structures, with their own feedback loops. What are the consequences of this? These effects should be presumably at least as strong as feedback they address. The authors should address this issue.

We thank the reviewers for highlighting this point. We have extended the discussion to highlight recent research that explicitly addresses recurrence within V1 (pg 19-20 lines 633-652).

We especially appreciated the distinction they made between different factorizations of a joint distribution over images and features that were identical when exact, but different when approximated. Since exact solutions are infeasible, this factorization provides potentially useful and biologically plausible structure. We think this nice theoretical aspect of their argument could be emphasized more prominently.

We thank the Reviewers for highlighting this point. We have now updated the text to reflect the Reviewers' comment (pg 3-4, line 132-146).

It's too strong to say that "correlations in the posterior can be identified with noise correlations" (L198). There are other reasons one could obtain correlations in the posterior, related to other forms of information loss that are not manifested as stochastic stimulus-conditioned variability.

We agree with the Reviewers that the current phrasing does not reflect the richness of potential sources of noise correlations. We have rectified this in the updated manuscript (pg 7, line 240-242).

The authors "test the consequences of the proposal that noise correlations are shaped by contextual priors" (L368). What is special about contextual priors here? Their test consists of phase scrambling, but this is not a compelling test of top-down inference. Instead, it may be just a generic property of more feedforward nonlinearities. The authors observe that high variance in V2 response across textures helps discriminate between them, and Figure 5 identifies what happens as a consequence of top-down effects. But when combining feedforward nonlinearities with nuisance variables (e.g. different samples of a given texture), we expect to same effects can be seen (e.g. Yang, Cotton, Walker, Tolias, Pitkow 2021). What is it that makes the authors conclude that top-down feedback plays a special role in these properties, beyond generic properties of more feedforward computation?

We thank the Reviewers for highlighting their paper: it provides some truly great insights about the interplay between nuisance variables, such as phase, translation, etc, and nonlinear representations. Without limiting the merits of the cited paper, we would like to note that we are intimately familiar with this issue: our Gaspar et al (2019) eLife paper investigated the negative effect of nuisance variables on linear codes and concluded that the elementary but ubiquitous point-nonlinearity, the firing rate nonlinearity can help establish linearly readable codes in the presence of nuisance variables. This point wise nonlinearity is similar to the example put forward in the paper, where a quadratic nonlinearity is proposed.

The non-human primate physiology experiment we are investigating in our paper is under the category that is referred to in the paper as stimulus-dependent noise correlations measured under invariant stimulus presentation. The paper introduces another form of correlation, nuisance correlations. In terms of our experiments, these 'nuisances' would be the changes in images that leave the overall statistics intact (ie. the texture family is identical) but changes occur in the identity of images (ie. different samples are considered). Such nuisance variables might come into play with uncontrolled eye movements too. However, the electrophysiology experiments were controlled for such eye movements, therefore the effects seen there are not expected to be 'nuisance correlations'. Getting back to the issue of nuisance correlations: we believe that this notion is highly relevant to the ideas presented in our paper. In fact, the analysis presented on Fig. 6e displays noise correlations (correlations in the posterior for one particular texture image) against correlations in the mean of the responses across multiple samples from the same texture family. While we call the latter signal correlation, it is actually the same notion as the one termed nuisance correlation in the referred

paper. As such, we demonstrate a link between noise correlations and nuisance correlations. This link is established through contextual priors: the shape of the posterior, as reflected by the way noise correlation is measured in Fig. 6c, is shaped by the contextual prior, and this contextual prior is argued to be specific to a texture family.

The reviewers propose that similar effects as those shown in Figure 6f could be obtained through ‘combining feedforward nonlinearities with nuisance variables’. Because of the relationship between nuisance correlations and noise correlations, we agree that similar effects could be obtained in a feed forward network for nuisance correlations. This feed forward explanation, however, does not seem to apply to noise correlations, which we investigate here (Fig. 6f,g).

To highlight how V2 representations affect V1 activity through top-down connections, we have extended the results with a pair of analyses. Importantly, these analyses establish further links with neurophysiology data from V1. We have explored removal of the texture statistics-related information from V2 by sampling the V2 prior instead of the posterior. We found that most of the linear decodability of texture information in V1 is eliminated by this procedure (Fig. 5a, pg 13 lines 421-429). This gives a chance to explore that V2-level sensitivity to texture-like statistics, as shown by Freeman et al 2013, is carried back to V1. We found that texture decoding from phase-scrambled versions of textures was smaller than for intact textures. Ziemba et al (2019) J Neurosci investigated the emergence of selectivity to phase scrambling and found that the difference in sensitivity to this statistics emerges later than in V2, indicating that the sensitivity to texture-level statistics is carried by top-down interactions. Similar results were obtained in the TDVAE framework: we assumed that initial responses in V1 rely on sampling the V2 prior rather than the posterior. Under such conditions, sensitivity to phase scrambling was limited (Fig. 5c, pg 13-14, lines 441-446). While this study is for mean responses instead of correlations, it suggests that top-down connections contribute strongly to texture-related statistics. Thus, we argue that changes in the patterns of noise correlations are compatible with texture-related information in mean responses and can be parsimoniously explained by top-down influences. We further expand on this at pgs 15-16 lines 470-482, 500-503.

Very minor:

Eq 4 — TDVAE should be superscript on q_{Φ} , not an exponent (or superscript) on its parentheses.

We thank the Reviewers for highlighting this problem. We addressed it in Eqs. (4) and (6).

Their phase and filter scrambling could be related to the Scattering transform (Mallat 2012) and metamers (Freeman and Simoncelli 2011).

We thank the Reviewers for highlighting this question. We followed the procedure outlined in Freeman and Simoncelli (2013) to generate the synthetic images, a method closely related to that of Portilla and Simoncelli (2000). Notably, we agree that the philosophy behind the image manipulation in Freeman (2011) (and other papers) aligns

with our approach, as it involves altering certain image structures while preserving specific statistical properties. However, we chose to cite the 2013 paper in the relevant Methods section as the details of our manipulation closely align with its approach.

Response to Reviewers

Please note that all the line numbers mentioned below refer to the updated manuscript and that response is in **bold**. *Colored text* refers to updated text in the manuscript.

Reviewer #1

The authors have provided a good and well-reasoned rebuttal. I wish to retract my previous comment about the model being a "hacked together VAE" - that is too strong. Indeed the authors have a clear theoretical motivation for their approach, and it is an interesting and novel contribution to the machine learning literature. The neurobiological motivations are also sound and clearly stated, and a model that can account for the role of feedback connections in the cortex is sorely needed.

We thank the Reviewer for appreciating our effort and are glad that we could convincingly argue for the soundness of our approach.

However my reservations about this paper's contribution to neuroscience remain. As a neuroscience model one would hope that it could provide insight as to why neurons respond the way they do. Although the model replicates certain phenomena in visual cortex, the reason for these properties emerging is unclear, other than that they are the properties of a hierarchical VAE trained on natural images. So we are left with an opaque system that happens to mirror the behavior of another opaque system, with only superficial similarities in their architecture.

Indeed, our approach puts emphasis on what insights the computational principles deliver and leaves circuit-level processes that implement the computations to a later work. However, we believe this is an important remark and clarify the contribution in the Discussion (line 610). We definitely see the need to address these processes and we regard such investigations as important future contributions.

The presented model provides a computational / algorithmic level account of the processes taking place in the early visual cortex. The approach is computational in the sense that it provides a normative account of how the connections from V2 to V1 need to be organized in order to perform hierarchical inference. It is algorithmic in the sense that sampling is assumed to perform inference. The choice of Variational Autoencoders to approximate inference is not considered to be essential but we propose it to be a convenient platform for the investigation of complex computations [57,56]. However, the approach hinders some circuit-level motivations, which can be addressed in an implementation-level model [66,64]. Such an approach has demonstrated that the combination of amortized inference and sampling to perform inference can provide insights into the recurrent interactions within the lateral structure of V1 [67,64] but a version that could perform hierarchical computations has not been explored so far.

The model architecture as diagrammed in Figure 1f seems a far cry from the architecture of visual cortex. Given that orientation-selective RF's are found already in layer 4 of visual cortex, how are we to understand the many layers of processing - Z1-ff followed by Z1-INT - between the image and Z1? Why are all these layers needed and what are they doing? Moreover it appears that Z1 (putative V1) does not serve as input to Z2 (putative V2). Rather Z2 gets its input from a separate stream going through Z1-ff followed by Z2-ff. Z2 then conditions Z1 through another multilayered set of connections, Z1-TD followed by Z1-INT. But topologically there is nothing hierarchical about this - Z1 and Z2 are both driven by the image through separate networks, so I would put them essentially at the same level. That they share Z1-ff in common, with some additional processing for Z2, does not put Z2 above Z1. Z2 conditions Z1, but there is nothing "top-down" about this, it is really more of a "sideways" modulation through the mixing of Z2's contribution into the Z1-INT layers feeding into Z1.

Our analysis focuses on one layer of V1, layer 2/3. We think that it would be a stretch to address layer by layer analysis, although we believe that this will be an important avenue for future research. However, inspired by the Reviewer's comment, we have analyzed the responses of other components of the circuitry. Orientation sensitivity can be identified in the neural network comprising V1-ff (see the receptive fields of a few selected dimensions in V1-ff below). However, inclusion of this would detract the reader from the main focus of the paper, top-down interactions between V2 and V1.

Nevertheless, it is certainly true that the model exhibits interesting response properties, and one can see certain parallels to visual cortical response properties. At the end of the day however, it is unclear what to make of all this with respect to our understanding of visual cortex.

Our paper demonstrates that hierarchical inference has strong predictive power with respect to the organization of top-down influences in V1, irrespective of the way the computations are implemented. The insight delivered by hierarchical inference contrasts with an attention-related interpretation of top-down influences. The paper explicitly includes a discriminative model as a control, which was shown to be a powerful tool to interpret hierarchical computations. The hierarchical probabilistic inference framework can be directly contrasted with the feed-forward architecture of discriminative models. Our neurally inspired extension of the Variational Autoencoder framework will have further impact on interpretation of top-down influences at other levels of sensory processing.

Reviewer #2

The authors have addressed all my requests for clarification and for additional analysis. The revised manuscript, which includes many new simulations, controls, comparisons between models, and replication of additional V1 data, has convinced me of the soundness of the conclusions (fully addressing my main concern). After reading also the other reviewers comments and the authors responses, I am convinced that the manuscript provides a very important conceptual contribution to the field of probabilistic neural representations, and the model could be the basis for interesting follow-up work.

We thank the Reviewer for the constructive comments.

At this point I have only a few very minor suggestions that would not require a re-review from me.

- Line 97 is the first mention of 'correlations' and is not explained or contextualized at all in the Introduction. Because stimulus dependent correlations are a main part of the argument of the paper, it seems important to introduce them. The Introduction is already lengthy though, so an alternative would be to not mention correlations in the Introduction at all.

Thank you for highlighting this. We have accepted the Reviewer's suggestion (line 94).

Through the development of a Variational Autoencoder model, our work provides a normative interpretation of the extensive top-down network present in the visual cortical hierarchy and demonstrates signatures of top-down interactions in neural response patterns.

- Line 178 'Such choices... parameter sharing.' Could add a reference

References added.

- Line 182 mentions the 'computational graph' but then the pointer to Extended figure is given in the next sentence which makes it sound like the figure is about something else.

Thank you for highlighting this issue. We have moved the reference to the figure to the previous sentence.

- Line 218 mentions image size (20-pixel). The different image sizes are important, as addressed in the rebuttal and in the Methods. But at this point of the text it is unclear why this number matters and even what it refers to (unless one has read Methods first, which not every reader will not do).

We have clarified the text (line 222).

We performed the comparison on a smaller model that used 20×20-pixel image patches instead of the default 40×40-pixel patches.

- Membrane potentials and firing rates, rectifying nonlinearity. I appreciate the authors motivation not to add a nonlinearity to their simulated activities, this is fine. But part of the justification given is that past work showed sampling explains both membrane potentials and spike counts. That work compared each version of the model to the corresponding data. Instead, simulated membrane potentials here are compared to empirical V1 spike count mean and noise correlations. I would try to tighten up the justification/discussion.

We have extended the discussion according to this suggestion (line 603).

In our study, TDVAE was directly used to model neural responses. Earlier studies using generative models of natural images have distinguished real-valued responses to model membrane potentials or firing-rate transformed outputs to model spiking activity [37,38]. A Poisson-distributed version of VAE is an appealing variant to model spiking activity [56] but modeling stimulus-dependent changes in noise correlations needs careful evaluation of private variability of model neurons as this affects quantitative fits to neural data [65].

- Line 473 'the variational posterior is correlation-free'. This is an important point, but just mentioned casually and with jargon very deep in the paper. I suggest expanding and explaining in simpler terms. Also, I am not sure that even in Methods you explained that the Z_1 variational posterior is factorized across Z_1 neurons.

We thank the Reviewer for pointing this out. We have clarified this in the Results.

In order to demonstrate the contribution of top-down interactions, we capitalize on a specific property of TDVAE. At a given stimulus and a specific level of Z_2 response, the variational posterior is chosen to be a Laplace distribution, and specifically a variant that is free of correlations. However, as Z_2 response is not a given number but a distribution over possible values as defined by the posterior, variations in Z_2 might induce covariations in Z_1 . Consequently, in TDVAE, it is the top-down influence that is exclusively responsible for the covariability of Z_1 neurons.

Reviewer #3

I reviewed this manuscript previously, and I thank the authors for making fairly substantial changes in response to my comments and the comments of the other reviewers, as well as for providing in-depth replies to all concerns raised (even the rather minor ones!)

We thank the reviewer for this feedback.

After reviewing the changes and the comments by the other reviewers, my original feeling remains that the model and analyses are interesting and thorough, but that claims about the modeling results' relevance to the brain are overstated.

We argue below that the technical concerns of the Reviewer are actually not issues. Further, we argue that the neuroscience concerns of the Reviewer are addressed by considering the full spectrum of analyses and neuroscience evidence.

Please see our point-by-point replies to the specific issues below.

It's clear that feedback and recurrence in the brain make for richer information-processing. It's also clear that Z1 in the TDVAE model becomes richer by adding a Z2 -> Z1 connection pathway. However, echoing my earlier concerns and the concerns of the other reviewers, I still do not think that any of the effects of Z2 on Z1 in the paper can be unambiguously mapped onto "top down" or "feedback" connections from V2 to V1 in the brain any more than they can be mapped onto local recurrence or even a more complex feedforward pathway in the brain.

Challenging the proposal that Z₂ of the model corresponds to V2 is justified in principle and we are more than happy to clarify. The point of the Reviewer that Z₁ modulation by Z₂ activity can be just as well identified with local recurrence in the brain contradicts both the literature and our analyses:

- 1. Illusory contour responses recorded in V1 have been widely reported to originate at V2 in non-human primates (NHPs) and at LM, a homologous region of the ventral stream of rodents, in mice. As the Reviewer pointed out in the previous round, those were time delay experiments that provided evidence for this. We made this time delay effect explicit in our newly introduced figure (Fig. 5b). This argues for a similar effect of Z₂ on Z₁ in the model as that observed in V2 and V1 in macaques. We agree that more direct evidence than timing is favoured. Not in NHPs but in mice such more direct evidence has recently been delivered (Pak et al, 2020): optogenetic inhibition of LM was shown to eliminate illusory contour responses. Intriguingly, this paper also confirmed that layer 2/3 has much stronger illusory contour response than layer 4, reinforcing our argument about the laminar organisation of the top-down feedback (the paper also highlights strong illusory responses in the superficial layers, the layer that is anatomically identified with the convergence zone of top-down feedback -- we speculate that this can be Z₁-int on Fig. 1f). Thus, these evidences point to V2 involvement in illusory contour responses in V1.**
- 2. The Reviewer suggests that Z₂ modulation of Z₁ in the model is not mapped to V2 to V1 connections. Please note that identification of Z₂ with V2 was not done**

through the role of Z_2 in the illusory contour modulation of Z_1 alone. Instead, we started with a thorough characterization of the Z_2 representation and a thorough reproduction of the battery of analyses of the original publications performed on V2 data. We are not aware of a comparably precise comparison. Our confidence is boosted by direct feedback from the authors on those papers on this.

3. The importance of this is further emphasized by the analysis inspired by the previous round of reviews: texture statistics related information, represented in Z_2 , is carried back to Z_1 in the model, a result confirming electrophysiology data from NHPs (Ziamba et al, 2019, Fig. 5c). That paper also shows a laminar profile consistent with the Pak et al mouse paper, again arguing for the involvement of superficial layers and L2/3.
4. Finally, the last timing experiment, contour completion (Fig 5d) is also showing results consistent with an experiment that provided evidence for V2 involvement.

Taken together, these various analyses cover experimental data that span highly out of distribution stimuli and more naturalistic ones, all pointing to the conclusion that the modulation of Z_1 delivered by Z_2 is highly aligned with modulations of V1 attributed to V2. We have updated the discussion to reflect these points (lines 551 and 663).

The hierarchical organization of TDVAE showed strong alignment with the early hierarchy of the ventral stream of the visual cortex: linear features of Z_1 reflected simple cell characteristics of V1, while the texture representation in Z_2 closely matched texture sensitivities of V2 neurons in primates and rodents. The match between the hierarchy of representations made it possible to investigate top-down computations.

As texture was shown to be represented in V2, and evidence supports that texture-related information reaches V1 with delay [45], this result also confirms that the feedback from Z_2 to Z_1 in TDVAE corresponds to across-area rather than within-area computations.

The authors' defense of the "top-down" terminology is essentially that (i) V2->V1 influences in the brain are fairly well documented, and thought to be involved in effects like time-delayed V1 responses to illusory contours;

And as pointed out above, further support comes from that Z2 representation faithfully reproduces characteristics of the representation found in V2.

and (ii) the $q(Z_2|im)q(Z_1|Z_2,im)$ factorization of the approximate posterior is one of many valid solutions for inference and implies some mechanistic information-flow from Z2 back to Z1.

Regarding (i), this would be a more compelling argument if Z1 in the model were closer to V1 in the brain. But as it stands, everywhere that the model is compared against V1 and V2 experimental data, it is clear that V1 in the brain is far more complex than Z1 in the model.

We acknowledge that there are simplifications in the model. Please note however, that some of the apparent simplification comes from a simplification of the terminology. We refer to Z1 as V1 in the paper for the ease of comprehension: as Fig. 1f indicates, the circuitry of the box accommodating Z1 is far more complex, and has the potential to elaborate on additional circuit elements of V1. Our discussion of this issue in the Discussion made an attempt to clarify the laminar contributors to TDVAE computations. We acknowledge that the paper would benefit from a brief clarification at the introduction of the theory (eg. by indicating that the boxes of Fig. 1f allude to the circuitry hosted by V1 and V2, and Z1 corresponds to a specific layer of the V1 circuitry). We believe that it has merits to investigate the auxiliary circuitry of Fig. 1f further but we believe that it is far beyond the scope of this paper. An actual simplifying assumption of the model is the linear generative model of V1 -> image, which follows a wide lineup of earlier studies and reflects that simple cell-like receptive fields emerge in a linear generative model. We argue that the model delivers an array of phenomena found in V1 without further assumptions about the composition of V1. Thus, simplification serves transparency. To clarify the paper, we add extra background to the Results, when the model is introduced (line 151).

Specifically, we identify Z₁ responses to layer 2/3 responses in V1. Although the circuitry in the recognition model is more complex than a single layer, for the sake of simplicity, we refer to the Z₁ neurons as V1 throughout the paper.

Fig 2e vs 2f shows much more structure in V1 than Z1.

Indeed, we agree with the reviewer. It is important though to note that this 'more structure' is way less than V2/Z₂. We hope that our model can inspire more fine grained analyses of the neural data on how different layers at different times contribute to this structure. We believe that the model provides the right tool to formulate testable hypotheses about such questions.

Texture decoding performance from V1/V2 is not reported in Fig 2d, but could be another useful comparison (and I would wager that V1 in the brain performs better than Z1 in the model).

Fig. 3f delivers the information on texture decoding that the Reviewer requests. We indeed did our best to be thorough. And no, Z₁ is not worse than V1.

Let us highlight one piece of the result here which is actually very illuminating. The neuroscience experiments provide little control over where the information captured by a decoder (or alternatively by t-SNE) originates. Our experiment shown on Fig. 5a indicates that the majority of texture decodability measured in Z₁ is inherited from Z₂. Optogenetic interventions might give us a chance to investigate this directly.

Figs 3d/3f/3h show much less drastic of a difference between V1 and V2 than Figs 3c/3e/3g show for model Z1 and Z2. Perhaps the exchange of information between Z1 and Z2 in the model is a good model for the relationship between simple cells (Z1) and other more complex cell types (Z2) all within V1, and perhaps the Z2->Z1 influence in the model nicely captures the effects of local recurrence in the brain.

Please see our arguments on how Z_2 is well justified to be identified with V2. We do not see compelling evidence that a similar texture representation was present in V1 as that reported in Freeman 2013 Nat Neurosci, Ziemba 2016 PNAS and consistent findings from LM of mice.

While I do think that V2->V1 feedback in the brain is likely the primary source for some of the effects claimed in this paper (like illusory contour responses), I don't think that we actually learn that from the TDVAE model.

We do not intend to state that we learn these from the model. Instead, we state that a spectrum of effects attributed to top-down connections to V1 is a consequence of hierarchical perceptual inference. The contribution of this paper is to construct a nonlinear generative model and show that irrespective of whether top-down influences are tested on very artificial or more naturalistic stimuli, the TDVAE provides a synthesizing, and notably parameter-free account. In this respect the computational framework (variational autoencoder) is a vehicle to (end-to-end) train and make inference in such a hierarchical model. These top-down effects are distinctive as discriminative models cannot capture these.

New controls in Fig 4 are useful, and I thank the authors for going through the efforts to include them. Unfortunately, I think they end up undermining the story: the fact that the ShallowVAE and the TDVAE with Z2 clamped still show an enhanced response to illusory contours calls into question whether "top down" effects are really the key to the story, even in the model. While inference of $q(Z_2|i)$ produces more of an effect, it's not clear that more is better here; after all, all model fits here are qualitative and the experimental data in Fig 4c also suggest to me that the illusory contour response in the brain is not all that vigorous either.

Indeed, the substantial contribution comes from inference of $q(Z_2|i)$ and a marginal effect is still present in the clamped Z_2 (not in the one relying on the prior). Critically, the residual difference between linear response and model response for clamped Z2 does not provide direct account of boosted illusory contour response in V1 *after* V2 displays signatures of inference.

Regarding point (ii) above, I do agree with the authors that the $q(Z_2|i)q(Z_1|Z_2,i)$ factorization is nice in the sense that it is computationally convenient. In fact, this is essentially one step away from rederiving message-passing algorithms, since both $\int q(Z_1|i) = \int q(Z_2|i)q(Z_1|Z_2,i) dZ_2$ and $\int q(Z_2|i) = \int q(Z_1|i)q(Z_2|Z_1) dZ_1$, implying, mechanistically, recurrent information flow between Z1 and Z2 and that the true posterior is a fixed point of these bi-directional dynamics.

Thank you for pointing this out.

In this work, the authors have borrowed a useful trick from machine learning (cf. LVAE, VLAE, NVAE architectures) where amortized inference is applied to hierarchical models using an initial deterministic feedforward pass from inputs to both Z_1 and Z_2 , which is then combined with a reparameterized sample of Z_2 fed back to Z_1 , inducing some stochasticity. However, the ELBO (or $KL(q||p)$ if you prefer) is technically defined in terms of an expectation over q . At training time, it is sufficient to estimate this expectation using a single reparameterized sample from Z_2 , since this gives unbiased (but high variance) estimates of the ELBO gradient. This is sufficient for learning as long as the learning rate is small, averaging away that stochasticity and effectively maximizing the ELBO.

We thank the Reviewer for the summary.

But using the reparameterization trick as a mechanistic source of stochasticity at inference time abuses the definition of the ELBO and confuses a trick used for training (single reparameterized samples drawn from q) with a trick for inference (recurrent message passing between Z_2 and Z_1).

We agree that the 'reparameterization trick' permits the calculation of the gradient, critical for learning. However, please note that the reparameterization trick has nothing to do with how inference is performed in the learned model. We perform marginalization over Z_2 through a Monte Carlo integral: the ELBO is written up to approximate the joint posterior $p(Z_1, Z_2|x)$ with $q(Z_1, Z_2|x)$. This latter is then partitioned into a product and we get a pair of variational posteriors, one actually conditioned on the other. When we marginalize the joint posterior with Monte Carlo integral, we are routinely sampling this approximate posterior of Z_2 . Undeniably, the inference will be approximate. As a benefit, this marginal posterior will be more expressive than the basic variational distribution. We actually demonstrate that this additional structure in the variational posterior reflects the fundamental requirement that the correlation of the posterior (referred to as noise correlation in the paper) reflects the structure in the data (captured by signal correlations) (Fig. 6e, Extended Data Fig. 10b). In summary, we respectfully disagree that 'abuse' or 'confuse' are adequate terms to describe the approach we take.

The Reviewer later describes our approach as an 'interesting' and 'novel' inference scheme. We did not originally consider it novel, as we were just consistently applying classical tools from probability theory to a rather novel tool (hierarchical variational autoencoder). To clarify this, we have added a paragraph to the Methods (line 1073).

Inference in the hierarchical generative model.

The recognition model directly provides the essential tools to perform inference in the model. The amortized variational inference inherent in the VAE framework yields a normal-distributed approximation of the true posterior distribution $p_\theta(Z_1, Z_2 | image) \approx q_\phi(Z_1, Z_2 | image)$. To relate the inference of the model to neural responses in V1 and V2, marginal distributions are required. Marginal posterior for Z_2 is directly provided by our choice of partitioning, as the TDVAE learns the function $q_\phi(Z_2 | image)$. For the marginal posterior of Z_1 , however, only $q_\phi(Z_1 | Z_2, image)$ is available, necessitating marginalization over Z_2 :

$$q_\phi(Z_1 | image) = \int dZ_2 q_\phi(Z_1 | Z_2, image) q_\phi(Z_2 | image)$$

Instead of performing the integral, we obtain a Monte Carlo approximation by sampling $q_\phi(Z_2 | \text{image})$. Consequently, the marginal posterior for Z_1 combines variational inference with sampling.

Two notes on the above argument.

First, we are aware that a multi-sample ELBO would yield a variational posterior that is inherently more expressive. This Importance Weighted Variational Autoencoder (IWAE) is a great alternative to obtain a tighter ELBO (Burda et al, 2015). We chose the standard single-sample formulation of the learning because it yields clearer insights for the readers as the posterior in IWAE is highly implicit (Cremer et al, 2017).

Nevertheless, we have done the homework: we extended the IWAE to a hierarchical setting, implemented it and trained it. As expected, better ELBOs were obtained. The qualitative properties of the posteriors did not differ from those of the TDVAE. While we considered this a useful exercise, we deemed this sort of analysis way beyond the scope of this submission and not necessarily relevant for the readership of this paper.

Second, as the Reviewer previously suggested, constructing a sampler for the generative model is also an option to obtain guarantees that the way we obtain the approximate posterior, $p(Z_1 | x)$, is adequate. This sampler can provide a chance to further validate the approximate posterior we obtain through Monte Carlo integral on variational posteriors. We have implemented and completed this analysis. We designed a sampler that could evaluate the moments of the posterior distribution of the generative model. We focused on the moments relevant for the paper, the posterior mean at Z_1 and Z_2 , as well as posterior correlation at Z_1 .

Our analysis confirmed that the estimated means and correlations faithfully match those obtained from the generative model. We have added a supplementary figure to the paper (Extended Data Fig. 5) and updated the results (line 217) and discussion (line 593) to refer to this additional layer of validation.

Inference in the model was validated by contrasting the match between inference in the recognition and in the generative models (see Methods for details and Extended Data Fig. 5).

Alternatively, inference can be performed by directly sampling the learned generative model. We performed such analysis to validate the recognition model but relied on the recognition model throughout the paper. We motivate the chosen approach by three arguments. First, while evaluation of the recognition model through sampling-based variational inference and direct evaluation of the generative model through sampling yields the same posterior in a well-calibrated model, the variational posterior provides a useful tool for disentangling the contribution of top-down influences to both response means and response correlations.

Extended Data Figure 5. Validation of inference. Comparison of posterior means (a, b) and posterior correlation (c) obtained from the recognition model (denoted with q) and from the generative model (through performing self-normalizing importance sampling, SNIS). Moments are calculated for Z_1 and Z_2 for images sampled from the 15 texture families. Individual dots represent means of latent posteriors (a, b) or posterior correlations of latent pairs (c). Colors represent different texture families. Moments are calculated for 80 (a, b) or three (c) images from every texture family.

We have added the description of this validation to the Methods (line 1084).

Validation of the learned deep generative model.

Optimizing the ELBO promises tightening approximation of the true posterior with the variational posterior, $q_\phi(Z_1, Z_2 | \text{image})$. As it is critical to assess if this approximation is sufficiently tight, we have validated the model by calculating the moments of the posterior obtained defined by the generative model, $p_\theta(Z_1, Z_2 | \text{image})$, and that provided by the recognition model, $q_\phi(Z_1, Z_2 | \text{image})$. We used self-normalized importance sampling (SNIS) to calculate the means of Z_1 and Z_2 of $p_\theta(Z_1, Z_2 | \text{image})$ and correlations between Z_1 's. SNIS provides a convenient opportunity to calculate these by simply obtaining samples from the joint normal distribution of $q_\phi(Z_1, Z_2 | \text{image})$ and then reweighing these samples by the fraction of the generative and recognition posteriors.

We performed SNIS for images from the 15 texture families, using 80 or three instances from each family for the means and correlations, respectively. Means and correlations were calculated by collecting a large number of samples ($4 \cdot 10^3$ for means and $38.4 \cdot 10^6$ for correlations) discarding the sample with the largest weight. SNIS-estimated moments were contrasted with the moments estimated from the recognition model (Extended Data Fig.~5).

True variational inference converges to a single best-estimate for $q(Z_1)$ and $q(Z_2)$ for each input and is "deterministic" in the sense that the same "best" $q(Z_2)q(Z_1)$ approximation will be reached for the same input every time (local optima notwithstanding).

We thank the Reviewer for expanding on this. We agree that our approach generalizes variational inference. In fact the inference in VAEs generalizes the 'true' variational inference by default as it is using an amortized version. Our approach yields a more expressive marginal posterior from a joint than the class of distributions the basic variational approximation yields.

The new text L224-L233 is misleading in this regard. A deterministic algorithm for inference is not necessarily a MAP algorithm; variational inference taken to convergence is also nearly deterministic in the sense that it gives the same q for the same input every time.

We apologize if it was interpretable as if probabilistic inference was only possible with a stochastic solution: we certainly did not mean to imply this, we fully agree with the Reviewer. We have clarified the text (line 232).

*A **non-probabilistic** approach focuses on the best interpretation of the image, thus the neural response would be identified with the maximum of the distribution. In neural data, this maximum a posteriori response is identified with the trial-averaged response and variance is interpreted as mere noise in the circuitry. In contrast, optimal computations require a probabilistic interpretation. For this, we adopt the sampling hypothesis [34]. The sampling hypothesis takes a stochastic approximation to represent a probability distribution. According to this, neural activity at any given time is a stochastic sample taken from the probability distribution and the sequence of samples constitutes the distribution*

Likewise, a stochastic inference algorithm like sampling is not "probabilistic" in virtue of its stochasticity, it is probabilistic in virtue of the fixed point of its dynamics being a representation of the posterior.

We strongly agree with this statement.

All that being said, the TDVAE is an interesting model and it's interesting that applying the reparameterization trick in Z2 at test-time reproduces some properties of V1 in the brain that the authors report. I would support the authors in publishing this work as an interesting case-study in recurrent (but not necessarily "top-down") dynamics of a particular (novel?) inference scheme where reparameterized samples are used at test-time, making for a sort of hybrid between variational and monte carlo methods.

We have strived to address all the concerns and clarify all the remaining issues of the Reviewer above. Consequently, we argue that the conclusions of the paper hold: both the technical details of the model are sound (and the Reviewer's interpretation about a specific role of reparameterization trick in the results is a misunderstanding); as well as our claims about the computations in the ventral stream are supported by extensive analyses of the analogy of the Z_2/V_2 representations, and the contributions of Z_2 connections to Z_1 activations reminiscent of V2 influences on V1.

However, the claims of the normativity and the generality of the model remain overstated, as do the claims that $Z_2 \rightarrow Z_1$ influences in the model should be interpreted as "top-down" effects.

It is normative as it is optimized to learn a generative model of natural images and does this in an unsupervised manner, albeit having algorithmic constraints on learning (optimising the ELBO instead of learning $P(x)$) and inference (approximating the posterior with a combination of amortised variational inference and Monte Carlo). Or in other words, as normative as a deep generative model gets.

A few more minor nits:

- L178: should include references for the claim that the architectural motif is "widely used in ML". (Ladder VAE and related methods, perhaps?)

Thank you for this remark. References added.

- L218, Fig 2a, and extended data table 4: was only one model of each type trained? The ELBO (like EM) is known to be prone to getting stuck in local maxima. Unclear whether ELBO differences like -400 vs -407 are meaningful. If not already doing so, better practice would be to train multiple instances of each model and take the best-performing model in each class.

We agree with the Reviewer that repeating model training from different random seeds is a useful empirical way for checking the stability of results wrt the different local extrema found with stochastic gradient methods. Due to compute constraints, we only repeated the training of our main model "TDVAE deep nonlinearity" and incorporated it into the "Model training details" section of Methods (line 1160) and into Extended Data Table 4.

We found that the ELBO difference between the two "TDVAE deep nonlinearity" model instances is 1. In light of this, the ELBO difference between the 20x20-pixel TDVAE models with shared and non-shared feedforward pathways (-400 versus -407) is probably significant. It is more important, however, that these two models show similar qualitative behavior (not discussed in the paper in detail, but see Extended Data Table 4).

Line 1178: *In the paper, we present one instance of each model type, except our main model 'TDVAE deep nonlinearity' the training of which we repeated starting from a different random seed. We found that the validation ELBO values (original: -1699, repeated: -1700) are well separated from those of our other 40x40-pixel models (next largest validation ELBO: -1705) and that the dimensions used by the learned representations (active Z_1 units: original: 1256, repeated: 1256; active Z_2 units: original: 6, repeated: 5) are consistent (see Extended Data Table 4). These results signal that our main model 'TDVAE deep nonlinearity' is well defined.*

Line 224: *Simulations did not identify major performance or qualitative differences with constraining the model to shared pathways (validation ELBOs: shared pathway: -400, distinct pathways: -407; see also Extended Data Table 4; see also Methods).*

- Fig 2b: I appreciate the addition of STC for Z2, but at the given resolution it looks like noise. The frequency domain visualization helps a bit, but it is still rather hard to make sense of.

Thank you for highlighting this. The frequency-domain response provides insight about the potentially nonlinear response profile of the neuron. The structure there indicates orientation sensitivity, but not necessarily one that can be read out linearly. The noisy STA confirms that no linear structure is available. However, in order to make the message of STC easier to interpret, we now include orientation tuning profiles for Z_2 , a standard method in neuroscience. Orientation tuning is obtained with moving grating stimuli in experiments, i.e. through marginalizing through the phase of periodic stimulus. We have updated the text accordingly.

Caption of Fig. 2: *The first-order and an example second-order receptive field, as well as the orientation tuning (STA, STC, and ori., respectively, see Methods) of a selected active Z_2 neuron*

Line 269: *The second-order, nonlinear receptive fields and the orientation tuning curves of Z_2 units, however, reveal orientation and wavelength selectivity (Fig. 2b STC and ori., Methods), offering a first glimpse into the representation learned by the Z_2 layer.*

Line 1275: *The orientation tuning of a Z_1 or Z_2 unit was calculated from the posterior mean responses of the unit to sinusoidal gratings with different orientation, phase, wavelength, and contrast values. Response amplitude was defined as the difference between the maximum and minimum responses while sweeping over all phase values at fixed stimulus orientation, wavelength, and contrast. Fig. 2b shows the response amplitude of the selected Z_2 unit at different stimulus orientations with stimulus wavelength 1/8 of the patch edge and stimulus contrast equal to that of the training set.*

Reviewer #4

Overall, I still find this to be an interesting study, and I support its publication with a bit of clarifications in discussion. I have two main comments, and a bunch of minor ones. I don't feel I need to see a further revision.

We thank the Reviewer for the encouraging words.

First: I think it is important to clarify the distinctions between top-down and recurrent. Conventionally top-down connections are viewed as feedback, and thus are recurrent. In contrast, their model is purely feedforward, because they are only unrolling one loop of recurrence. This is helpful to many readers to say up front.

We thank the reviewer for this remark. We have clarified this in the Methods (line 1017) as well as in the Discussion (line 629).

The computational graph shows that the Z_1 layer is reached through Z_1 -ff, Z_2 -ff, Z_1 -TD, and Z_1 -INT by crossing the Z_2 layer. As such this is a top-down architecture. While parts of this circuitry, notably Z_1 -ff, involve neurons that might reside in V_1 , the architecture is not recurrent as those are not the same neurons that participate in the feed forward and feedback information flows.

Note that the information that reaches Z_1 through the top-down influences of Z_2 are not implemented in a recurrent loop, instead, the feed-forward pathway that is shared between Z_1 and Z_2 is relaying information to Z_2 before feeding it to Z_1 . This single-step top-down pass avoids highly recurrent refinement of the responses between V_1 and V_2 .

Second: I'm glad the authors incorporated a computational diagram to explain their architecture. Done well, this will be very helpful. I think the Extended Data Figure 1 is more important than a supplementary figure, and could go well with Figure 1f.

We thank the reviewer for highlighting this possibility. As the current extended figure contains three alternative architectures we argue that it is easier for the reader to keep these at one place and therefore we prefer to leave this in the supplementary material. We also think that a single depiction of the model structure is more straightforward for the readers, thus also avoiding an overcrowded figure. However, if the Editor thinks it would be more useful to integrate these figures, we are happy to do so.

That said, I find the current version quite confusing. The superposition of models are very hard to parse. For instance, there are two streams involving Z2: the FF pathway and the generative pathway. Are they both used?

Thank you for this note. We have clarified the figure with added shading to the neural networks belonging to the generative model. We have added explanation to the caption:

Yellow corresponds to the recognition part of the model, blue corresponds to the generative model, MLPs of the generative model have blue shading.

Further, we have added a clarification to the Methods (line 1011):

The resulting computational graph is depicted in Extended Data Fig. 1. Note that the generative model component was used for training but our analyses exclusively focused on the recognition model. This choice is motivated by the fact that the recognition model is responsible for performing inference in the model, that is the recognition model delivers the (approximate) posteriors. Those are the posteriors that V1 and V2 neurons are assumed to represent during perceptual computations.

I also should mention I have several disagreements with Reviewer 1. The architecture is reasonably principled. And VAEs should not be judged on giving mechanistic insights. There are many many ways of achieving the same representations and computations. For some purposes we really care about the nuances here. But for computational insight about the kinds of things that are represented, much progress has been made without a one-to-one correspondence between artificial neurons and biological neurons.

We thank the reviewer for reinforcing this point.

MINOR:

Fig 1f: aesthetic suggestion: black arrows confusingly bend unnecessarily. Reorganize feature maps so flow is more easily seen.

We thank the reviewer for the suggestion. We have updated the figure to eliminate several twists and turns.

L165: some conflict here between “TDVAE was left as general as possible” and “Z1 is linearly related to image”

We have clarified this description (line 162). Briefly, the general VAE framework provides a platform for acquiring a generative model, and this platform is operated in a principled manner. Inductive biases and stimulus statistics jointly shape the learned generative model and the choice of linear lower component reflects one particular inductive bias.

First, the computational framework is determined by a highly principled approximation in probabilistic inference but apart from the well-motivated restrictions (optimizing a lower bound to the true objective, performing variational amortized inference, deterministic propagation of signals across layers of the hierarchy), the TDVAE was left as general as possible such that its properties were determined solely by natural image statistics. Second, we capitalized on neuroscience intuitions about V1 and V2 to constrain the architecture for learning about low- and mid-level features, Z_1 and Z_2 . Along this line...

L175: "sharing the entry layers of the two networks... is referred to as parameter sharing." This doesn't sound right, it's instead an architectural choice. Parameter sharing would instead be duplicated layers at distinct sites whose parameters are tied. You could say instead that this is "equivalent" to parameter sharing in a different architecture with two distinct branches, but I'm not sure what you gain from saying that.

Thank you for highlighting this. The use of the term parameter sharing was motivated by the fact that the ELBO defines two distinct neural networks to perform two independent steps of inference, and we assumed that part of the neural networks is characterized by the same set of parameters. In our specific case this choice ultimately leads to a simplified architecture. We have updated the text to convey this intuition better (line 180).

This is analogous to parameter sharing, a widely used approach in machine learning, where two components of the computational graph (parts of $p(z_1 | z_2, x)$ and $p(z_2 | x)$ in our case) are tied. This results in an architecture where a single feed forward pathway reaches V1 and then V2.

L220, comma should be colon

Thank you for noting this.

Fig 2a caption: Shallow-VAE: non-hierarchical VAE constrained to Z1. Isn't this just linear, since Z1 is linear in pixels?

Yes, this insight is right. We added this to the caption.

Shallow-VAE: non-hierarchical VAE constrained to z_1 , yielding a linear model of V1.

L272: A 0.2 improvement is considered LOW for Z1, but a smaller 0.1 improvement by their preferred model is considered outperforming? Seems like changing judgments.

We apologize for the confusing phrasing. The reported numbers were absolute decoding performances and not differences. We have clarified the phrasing (line 284).

The non-hierarchical version of VAE that completely lacked a Z_2 was slightly outperformed by TDVAE (compare 0.1073 ± 0.0020 with 0.1943 ± 0.0012 for ' Z_1 , shallow VAE' and ' Z_1 , TDVAE', respectively, Fig.2d).

L272: Text needs labels to reference bars in Fig 2d so the reader can more easily compare claims to results.

Thank you for this suggestion, we have updated the text accordingly (line 281)

Decoders that queried the texture family that a particular image was sampled from but ignored the identity of the image showed low performance on model Z_1 neurons, which was still distinct from chance (0.1943 ± 0.0012 , error denotes standard deviation over five repeats of the fit, Fig. 2d ' Z_1 , TDVAE').

L276 “endow the Z_2 to build” — misused grammar. Should be “endow Z_2 *with* capacity to build...” or similar

Text updated.

L278: Mean over what? Neurons encoding means of Variational Approximation? See Reviewer’s previous comment on being careful about the use of the term “mean”, which can mean many things here.

Thank you for highlighting this. We have clarified the text (line 289)

Texture was reliably decodable from the response *intensities* of Z_2 neurons of TDVAE.

L297: Here .08 is “markedly lower” while 0.28 is only “slightly higher”?

Again, we apologize for the confusing phrasing. The reported numbers were absolute decoding performances and not differences. We have clarified the phrasing (line 306).

The untrained model showed slightly higher decoding performance on texture families in Z_1 (compare 0.2850 ± 0.0078 with 0.1943 ± 0.0012 for ' Z_1 , untrained' and ' Z_1 , TDVAE', respectively, Fig. 2d}) but, in contrast with TDVAE, markedly lower performance in Z_2 than the TDVAE (compare 0.0820 ± 0.0076 with 0.8761 ± 0.0001 for ' Z_2 , untrained' and ' Z_2 , TDVAE', respectively, Fig. 2d).

L968: ELBO = *Evidence* Lower Bound, not *Expectation* Lower Bound.

That's absolutely right, thank you for spotting.

Many places report absurdly small p values. We cannot have such confidence in the model, so just report something like $p \ll 10^{-3}$.

We see the Reviewer's point. We have abided by the standards of Nature Communications when reporting these numbers.

Response to Reviewers

Please note that all the line numbers mentioned below refer to the LaTeX source of the updated manuscript and that response is in **bold**. *Colored text* refers to updated text in the manuscript.

Reviewer #1

The paper has undergone another revision, mostly to the narrative. The fundamental structure of the model remains unchanged.

My main concern, as stated previously, is that the topology of this model does not match that of the visual cortical areas it seeks to shed light on. To be clear, the signal flow in this model is

image  Z1
image  Z2  Z1

whereas in visual cortex we have

image  V1  V2  V1  V2 . . .

In other words, in visual cortex we see a recurrent loop: V1 feeds to V2; V2 feeds back to V1, presumably changing its state; the modified V1 feeds to V2 again, and so forth. It is a loop that repeats. That is what makes it mysterious, and difficult to model.

The model in this paper has no such loop, it is more of a sideways modulation in one step with no recurrence. This is even emphasized by the authors now in the discussion:

"This single-step top-down pass avoids highly recurrent refinement of the responses between V1 and V2."

which is then justified by saying

"Experiments relying on time delays between neural responses in V1 and V2 are compatible with our account but more precise experimental data can identify more elaborate recurrent computations."

It is not clear which experiments are being referred to. According to Bullier the signal delay between V1 and V2 is 2-4 ms due to traveling via myelinated fibers over a short distance. So there is ample time for signals to flow back and forth.

We thank the reviewer for making their point very clear. We would like to clarify at this point that image information does not directly reach Z_2 . Instead, a neural network that precedes Z_1 transmits the information towards Z_2 . Although a detailed study of that neural network is beyond the scope of this paper, we assume that it can be associated with a subnetwork of a cortical column. This way, a feed-forward pathway reaches Z_2 and Z_2 projects back to Z_1 via intermediate neural networks. It is indeed an important

observation that this architecture avoids full recurrent processing such as that in the V1 layer 2/3 – V2 layer 4 – V2 layer 2/3 – V1 layer 2/3. Note, that the circuitry is certainly more complex than an A – B – A recurrence, as the above loop might also include additional elements like V1 layer 5. Still, it is important to acknowledge that layer 2/3 forward connections are highly characteristic and dominant but are not part of the circuitry we discuss in the paper. Reading the posterior ($p(Z_1|\text{image})$) for downstream computations has not been discussed in the paper but these layer 2/3 projections will certainly be critical in those.

To make sure these considerations are faithfully delivered to the reader, we added a paragraph, '*Neural circuitry for hierarchical inference.*' (line 120(LaTeX)/225(PDF)) where we provide these details.

Again, I wish to emphasize that the model has many interesting properties and attributes in its own right - it is an impressive piece of work, very extensive analyses and interesting results - so there is no question in my mind that this work should ultimately be published. It could even be seen as loosely capturing "higher-level" modulation of V1 since the stream to Z2 goes through more processing than the stream to Z1. But the fundamental gap between the model and visual cortex, as pointed out above, should be stated forthrightly in the introduction, and perhaps the title modified accordingly to qualify what the model is actually showing.

We have updated the text at three points:

1) The most comprehensive update concerned a restructuring of the first section of the Results. This restructuring is meant to help the reader to go through different considerations underlying the model in a structured manner. We have introduced three subsections: '*Computational principles of hierarchical inference*' (line 103(LaTeX)/123(PDF)), '*Algorithmic solution to hierarchical inference*' (line 113(LaTeX)/157(PDF)), '*Neural circuitry for hierarchical inference*' (line 120(LaTeX)/225(PDF)). (Due to their lengths, we do not insert the contents of these subsections here.)

2) Second, the last paragraph of the introduction features a clear reference to the architectural detail that the model is not built on a recurrent network (line 94(LaTeX)/93(PDF)).

The proposed framework relies on a feed-forward V1 – V2 pathway and a top-down pathway that avoids direct feedback to the feed-forward components, but integrates feed-forward and top-down components in a distinct population.

3) Third, in the Discussion, we make clear that the top-down structure we consider is not a recurrent structure, and that the model architecture does not consider all known anatomical pathways (line 195(LaTeX)/631(PDF)).

It is important to note that feed-forward information flow towards Z_2 is not implemented through Z_1 . While feed-forward projections from V1 have been identified from layer 4 neurons [41], layer 2/3 provides a strong source of feed-forward

information flow [40]. In our paper we did not consider how the V1 code is read by downstream areas for computational purposes, in which layer 2/3 projections certainly have a central role. Interestingly, such a feed-forward component would contribute to a more direct recurrent loop between V1 and V2, which could enrich computations.

Reviewer #3

This is my third review of this manuscript after some back and forth with the authors. In my last review, I raised two primary lingering concerns:

1. My first and biggest concern had been whether or not it was appropriate to analogize Z1/V1 and Z2/V2. I had expressed concerns about simplicity and complexity of the model and brain respectively. The authors' most recent reply was very thorough on this point, and they have convinced me that the Z2->Z1 pathway is, in fact, comparable to V2->V1 feedback effects in the brain, at least enough to be of genuine interest and use to the neuroscience community.

We thank the reviewer for the valuable and extensive discussion on this point, we believe that it contributed to the clarification of the text.

2. My second concern from the prior round of reviews had to do with the nature of variational inference and neural variability. This is a minor technical point and in no way calls into question the validity of the model or the results. The way that the TDVAE model explains correlated neural variability -- indeed the way it explains any neural variability -- is that samples of Z2 are drawn from the (fixed) $q(Z2|input)$ distribution, and $q(Z1|input,Z2)$ is conditioned on the sampled value from Z2. If the model were doing "pure sampling", it would draw samples for both Z1 and Z2 and there would be a lot more total variability. If the model were doing "pure variational inference", it would seek a steady-state representation of Z2 by fully marginalizing over Z1, and there would be essentially no variability remaining. My main point before was that what the TDVAE does is somewhere in between these. This is interesting, possibly even novel from an algorithmic perspective! But, I concede that it is not really wrong to call the TDVAE algorithm "variational inference" even if it includes a sampling component. I don't quite rescind this point, but I will demote it to the category of minor nit-picking.

Thank you for expanding on this topic. We certainly find your comment on extra variability very insightful. We need to add though that we attempted to be consistent on this issue and therefore the concern raised above does not apply. We approached the problem by assuming that neural responses constitute a sampling-based representation of the posterior. Thus, we assumed that the VAE produces a variational approximation to the true posterior but neural responses correspond to samples from this posterior. We relied on the specifics of the experimental paradigms to specify the number of samples we collected. In the case of the Z1 posterior, sampling concerned both sampling Z2 to approximate the Monte Carlo integral and then sampling the individual variational posteriors ($p(Z1|Z2,image)$). This way, the variability of responses consistently approximates the variability of the true posterior, as well as the marginalized variational posterior approximates the true posterior.

We have clarified this choice in the paper (line 118(LaTeX)/216(PDF)):

In practice, we collect samples from the variational posteriors of Z_1 and Z_2 , which is then compared to responses of V1 and V2 neurons in electrophysiology experiments. The number of these samples is set to approximate the specific experimental conditions TDVAE responses are tested on (see Methods).

I am happy to consider all issues resolved.

We thank the reviewer for all the constructive comments we received.